# Adam Can Converge Without Any Modification On Update Rules

**Yushun Zhang**[13], **Congliang Chen**[1], **Naichen Shi**[2], **Ruoyu Sun**[13]*, **Zhi-Quan Luo**[13]
[1]The Chinese University of Hong Kong, Shenzhen, China [2]University of Michigan, US
[3]Shenzhen Research Institute of Big Data
{yushunzhang,congliangchen}@link.cuhk.edu.cn, naichens@umich.edu
{sunruoyu,luozq}@cuhk.edu.cn

## Abstract

Ever since Reddi et al. (2018) pointed out the divergence issue of Adam, many new variants have been designed to obtain convergence. However, vanilla Adam remains exceptionally popular and it works well in practice. Why is there a gap between theory and practice? We point out there is a mismatch between the settings of theory and practice: Reddi et al. (2018) pick the problem after picking the hyperparameters of Adam, i.e., $(\beta_1, \beta_2)$; while practical applications often fix the problem first and then tune $(\beta_1, \beta_2)$. Due to this observation, we conjecture that the empirical convergence can be theoretically justified, only if we change the order of picking the problem and hyperparameter. In this work, we confirm this conjecture. We prove that, when the 2nd-order momentum parameter $\beta_2$ is large and 1st-order momentum parameter $\beta_1 < \sqrt{\beta_2} < 1$, Adam converges to the neighborhood of critical points. The size of the neighborhood is propositional to the variance of stochastic gradients. Under an extra condition (strong growth condition), Adam converges to critical points. It is worth mentioning that our results cover a wide range of hyperparameters: as $\beta_2$ increases, our convergence result can cover any $\beta_1 \in [0, 1)$ including $\beta_1 = 0.9$, which is the default setting in deep learning libraries. To our knowledge, this is the first result showing that Adam can converge *without any modification* on its update rules. Further, our analysis does not require assumptions of bounded gradients or bounded 2nd-order momentum. When $\beta_2$ is small, we further point out a large region of $(\beta_1, \beta_2)$ combinations where Adam can diverge to infinity. Our divergence result considers the same setting (fixing the optimization problem ahead) as our convergence result, indicating that there is a phase transition from divergence to convergence when increasing $\beta_2$. These positive and negative results provide suggestions on how to tune Adam hyperparameters: for instance, when Adam does not work well, we suggest tuning up $\beta_2$ and trying $\beta_1 < \sqrt{\beta_2}$.

## 1 Introduction

Modern machine learning tasks often aim to solve the following finite-sum problem.

$$\min_{x \in \mathbb{R}^d} f(x) = \sum_{i=0}^{n-1} f_i(x), \tag{1}$$

where $n$ is the number of samples or mini-batches and $x$ denotes the trainable parameters. In deep learning, Adam (Kingma & Ba, 2014) is one of the most popular algorithms for solving (1). It has been applied to various machine learning domains such as natural language processing (NLP) (Vaswani et al., 2017; Brown et al., 2020; Devlin et al., 2018), generative adversarial networks (GANs) (Radford et al., 2015; Isola et al., 2017; Zhu et al., 2017) and computer vision (CV) (Dosovitskiy et al., 2021). Despite its prevalence, Reddi et al. (2018) point out that Adam can diverge with a wide range of hyperparameters. A main result in (Reddi et al., 2018) states that [2]:

---

*Correspondence author
[2]We formally re-state their results in Appendix D.2.

36th Conference on Neural Information Processing Systems (NeurIPS 2022).

*For any $\beta_1, \beta_2$ s.t. $0 \le \beta_1 < \sqrt{\beta_2} < 1$, there exists a problem such that Adam diverges.*

Here, $\beta_1$ and $\beta_2$ are the hyperparameter to control Adam's 1st-order and 2nd-order momentum. More description of Adam can be seen in Algorithm 1 (presented later in Section 2.1). Ever since (Reddi et al., 2018) pointed out the divergence issue, many new variants have been designed. For instance, AMSGrad (Reddi et al., 2018) enforced the adaptor $v_t$ (defined later in Algorithm 1) to be non-decreasing; AdaBound (Luo et al., 2019) imposed constraint $v_t \in [C_l, C_u]$ to ensure the boundedness on effective stepsize. We introduce more variants in Appendix D.1.

On the other hand, counter-intuitively, vanilla Adam remains exceptionally popular (see evidence at (Scholar)). Without any modification on its update rules, Adam works well in practice. Even more mysteriously, we find that the commonly reported hyperparameters actually satisfy the divergence condition stated earlier. For instance, Kingma & Ba (2014) claimed that $(\beta_1, \beta_2) = (0.9, 0.999)$ is a "good choice for the tested machine learning problems" and it is indeed the default setting in deep learning libraries. In super-large models GPT-3 and Megatron (Brown et al., 2020; Smith et al., 2022), $(\beta_1, \beta_2)$ is chosen to be $(0.9, 0.95)$. GAN researchers (e.g. Radford et al. (2015); Isola et al. (2017)) use $(\beta_1, \beta_2) = (0.5, 0.999)$. All these hyperparameters live in the divergence region $\beta_1 < \sqrt{\beta_2}$. Surprisingly, instead of observing the divergence issue, these hyperparameters achieve good performances and they actually show the sign of convergence.

Why does Adam work well despite its theoretical divergence issue? Is there any mismatch between deep learning problems and the divergent example? We take a closer look into the divergence example and find out the mismatch *does* exist. In particular, we notice an important (but often ignored) characteristic of the divergence example: (Reddi et al., 2018) picks $(\beta_1, \beta_2)$ *before* picking the sample size $n$. Put in another way, to construct the divergence example, they change $n$ for different $(\beta_1, \beta_2)$. For instance, for $(\beta_1, \beta_2) = (0, 0.99)$, they use one $n$ to construct the divergent example; for $(\beta_1, \beta_2) = (0, 0.9999)$, they use another $n$ to construct another divergent example. On the other hand, in practical applications of Adam listed above, practitioners tune the hyperparameters $(\beta_1, \beta_2)$ *after* the sample size $n$ is fixed. So there is a gap between the setting of theory and practice: the order of picking $n$ and $(\beta_1, \beta_2)$ is different.

Considering the good performance of Adam under fixed $n$, we conjecture that Adam can converge in this setting. Unfortunately, the behavior of vanilla Adam is far less studied than its variants (perhaps due to the criticism of divergence). To verify this conjecture, we run experiments for different choices of $(\beta_1, \beta_2)$ on a few tasks. First, we run Adam for a convex function (2) with fixed $n$ (see the definition in Section 3.2). Second, we run Adam for the classification problem on data MNIST and CIFAR-10 with fixed batchsize. We observe some interesting phenomena in Figure 1 (a), (b) and (c).

First, when $\beta_2$ is large, the optimization error is small for almost all values of $\beta_1$. Second, when $\beta_1$, $\beta_2$ are both small, there is a red region with relatively large error. On MNIST, CIFAR-10, the error in the red region is increased by 1.4 times than that in the blue region. The situation is a lot worse on function (2) (defined later in Section 3.2): the error in the red region is 70 times higher.

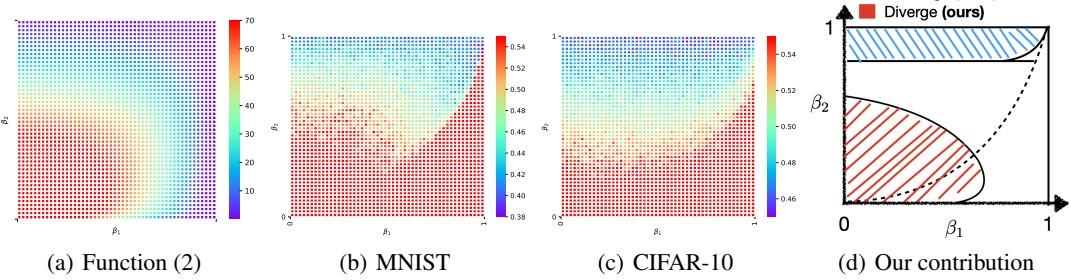

(a) Function (2)  (b) MNIST  (c) CIFAR-10  (d) Our contribution

Figure 1: **(a), (b), (c)**: The performance of Adam on different tasks. For each task, we show the results with $\beta_1$ and $\beta_2$ in grids $\{(k_1/50, k_2/50) | k_1 = 0, \cdots, 49, k_2 = 0, \cdots, 49\}$. **(a)**: the optimality gap $x - x^*$ on the convex function (2). **(b) (c)**: The training loss on MNIST and CIFAR-10. **(d)**: An illustration of our contribution in $(\beta_1, \beta_2)$ phase diagram. The shape of the region follows the solution to our analytic conditions. The size of the region depends on $n$. The dotted curve satisfies $\beta_1 = \sqrt{\beta_2}$. In all figures, $n$ is fixed before picking $(\beta_1, \beta_2)$.

While Adam's performances seem unstable in the red region, we find that *Adam always performs well in the top blue region in Figure 1.* This seems to suggest that Adam can converge without any algorithmic modification, as long as $\beta_1$ and $\beta_2$ are chosen properly. We ask the following question:

*Can Adam provably converge without any modification on its update rules?*

In this work, we theoretically explore this question. Our contributions are visualized in Figure 1 (d). We prove the following results when $n$ is fixed (or more rigorously, when the function class is fixed):

- We prove that when $\beta_2$ is large enough and $\beta_1 < \sqrt{\beta_2}$, Adam converges to the neighborhood of critical points. The size of the neighborhood is propisitional to the variance of stochastic gradients. With an extra condition (so-called strong growth condition), we prove that Adam can converge to critical points. As $\beta_2$ increases, these results can cover any momentum parameter $\beta_1 \in [0, 1)$ including the default setting $\beta_1 = 0.9$. In particular, our analysis does *not* require bounded gradient assumption.

- We study the divergence issue of small-$\beta_2$ Adam. We prove that: for any fixed $n$ (or more rigorously, for any fixed function class), there exists a function such that, Adam diverges to infinity when $(\beta_1, \beta_2)$ is picked in the red region in Figure 1 (d). The size of the red region increases with $n$. The shape of the region follows the solution to our analytic conditions.

- We emphasize a few characteristics of our results. **(1) phase transition**. The divergence result considers the same setting as our convergence result, indicating that there is a phase transition from divergence to convergence when changing $\beta_2$. **(2) problem-dependent bounds**. Our convergence and divergence regions of $(\beta_1, \beta_2)$ are problem-dependent, which is drastically different from (Reddi et al., 2018) which established the problem-independent worst-case choice of $(\beta_1, \beta_2)$. **(3) non-asymptotic characterization**. the "divergence region" of $(\beta_1, \beta_2)$ expands as $n$ increases and converges to the whole region $[0, 1)^2$ as $n$ goes to infinity, which recovers (actually stronger than) the problem-independent divergence result of (Reddi et al., 2018) that requires $\beta_1 < \sqrt{\beta_2}$. In this sense, we can view the divergence result of (Reddi et al., 2018) as an asymptotic characterization of the divergence region (as $n \to \infty$) and our divergence result as a non-asymptotic characterization (for any fixed $n$). We provide more discussion in Section 4.

- Our positive and negative results can provide suggestions for tuning $\beta_1$ and $\beta_2$: for instance, when Adam does not work well, we suggest tuning up $\beta_2$ and trying $\beta_1 < \sqrt{\beta_2}$. We provide more tuning suggestions in Appendix C.

We believe our results can boost new understandings for Adam. While Reddi et al. (2018) reveal that "Adam can diverge", our results show the other side of the coin: when $n$ is fixed (or when function class is fixed), Adam can still converge without any modification on its update rules. Our results suggest that Adam is still a theoretically justified algorithm and practitioners can use it confidently.

We further emphasize that our convergence results can cover any $\beta_1 \in [0, 1)$, which allows the algorithm to bring arbitrarily heavy momentum signals. It turns out that large-momentum Adam is not easy to analyze. Even with stronger assumptions like bounded gradient ($\|\nabla f(x)\| < C, \forall x$), its convergence is not well understood (see related works in Section 2.2). To our best knowledge, this is the first result that proves vanilla Adam with any $\beta_1$ can converge *without* any assumption of bounded gradient or bounded 2nd-order momentum. The proof contains a new method to handle unbounded momentum in the stochastic non-linear dynamics system. We will highlight our technical novelties in Section 5.

## 2 Preliminaries

### 2.1 Review of Adam

We consider finite-sum problem (1). We use $x$ to denote the optimization variable. We denote $\nabla f_j$ as the gradient of $f_j$ and let $\circ$ be the component-wise product. The division and square-root operator are component-wise as well. We present randomly shuffled Adam in Algorithm 1.

In Algorithm 1, $m$ denotes the 1st-order momentum and $v$ denotes the 2nd-order momentum. they are weighted averaged by hyperparameter $\beta_1, \beta_2$, respectively. Larger $\beta_1, \beta_2$ will adopt more history information. We denote $x_{k,i}, m_{k,i}, v_{k,i} \in \mathbb{R}^d$ as the value of $x, m, v$ at the $k$-th outer loop (epoch) and $i$-th inner loop (batch), respectively. We choose $\eta_k = \frac{\eta_1}{\sqrt{nk}}$ as the stepsize. In practice, $\epsilon$ is adopted for numerical stability and it is often chosen to be $10^{-8}$. In our theory, we allow $\epsilon$ to be an arbitrary non-negative constant including 0.

In the original version of Adam in (Kingma & Ba, 2014), it has an additional "bias correction" step. This "bias correction" step can be implemented by changing the stepsize $\eta_k$ into $\hat{\eta}_k = \frac{\sqrt{1-\beta_2^k}}{1-\beta_1^k}\eta_k$

**Algorithm 1** Adam

> Initialize $x_{1,0} = x_0$, $m_{1,-1} = \nabla f(x_0)$ and $v_{1,-1} = \max_i \nabla f_i(x_0) \circ \nabla f_i(x_0)$.
> **for** $k = 1 \to \infty$ **do**
>     Sample $\{\tau_{k,0}, \tau_{k,1}, \cdots, \tau_{k,n-1}\}$ as a random permutation of $\{0, 1, 2, \cdots, n-1\}$
>     **for** $i = 0 \to n-1$ **do**
>         $m_{k,i} = \beta_1 m_{k,i-1} + (1 - \beta_1) \nabla f_{\tau_{k,i}}(x_{k,i})$
>         $v_{k,i} = \beta_2 v_{k,i-1} + (1 - \beta_2) \nabla f_{\tau_{k,i}}(x_{k,i}) \circ \nabla f_{\tau_{k,i}}(x_{k,i})$
>         $x_{k,i+1} = x_{k,i} - \frac{\eta_k}{\sqrt{v_{k,i}} + \epsilon} \circ m_{k,i}$
>     **end for**
>     $x_{k+1,0} = x_{k,n}$; $v_{k+1,-1} = v_{k,n-1}$; $m_{k+1,-1} = m_{k,n-1}$
> **end for**

and using zero initialization. In Algorithm 1, the "bias correction" step is replaced by a special initialization, which corrects the bias as well. Note that $\hat{\eta}_k \in [\sqrt{1 - \beta_2}\eta_k, \frac{1}{1-\beta_1}\eta_k]$ is well-bounded near $\eta_k$, so $\eta_k$ and $\hat{\eta}_k$ brings the same convergence rate. In addition, as the effect of initialization becomes negligible when the training progresses, Adam with zero & our initialization will have the same asymptotic behavior. In the main body of our proof, we follow the form of Algorithm 1, which makes results cleaner. For completeness, we add the proof on the convergence of Adam with "bias correction" steps in Appendix G.11.

In our analysis, we make the assumptions below.

**Assumption 2.1.** we consider $x \in \mathbb{R}^d$ and $f_i(x)$ satisfies gradient Lipschitz continuous with constant $L$. We assume $f(x)$ is lower bounded by a finite constant $f^*$.

**Assumption 2.2.** $f_i(x)$ and $f(x)$ satisfy: $\sum_{i=0}^{n-1} \|\nabla f_i(x)\|_2^2 \leq D_1 \|\nabla f(x)\|_2^2 + D_0, \forall x \in \mathbb{R}^d$.

Assumption 2.2 is quite general. When $D_1 = 1/n$, it becomes the "constant variance" with constant $D_0/n$. "constant variance" condition is commonly used in both SGD and Adam analysis (e.g. (Ghadimi et al., 2016; Zaheer et al., 2018; Huang et al., 2021)). Assumption 2.2 allows more flexible choices of $D_1 \neq n$ and thus it is weaker than "constant variance".

When $D_0 > 0$, the problem instance is sometimes called "non-realizable" (Shi et al., 2020). In this case, adaptive gradient methods are not guaranteed to reach the exact critical points. Instead, they only converge to a bounded region (near critical points) (Zaheer et al., 2018; Shi et al., 2020). This phenomenon indeed occurs for Adam in experiments, even with diminishing stepsize (see Figure 4 (a)). The behavior of SGD is similar: constant stepsize SGD converges to a bounded region with its size propositional to the noise level $D_0$ (Yan et al., 2018; Yu et al., 2019; Liu et al., 2020b).

When $D_0 = 0$, Assumption 2.2 is often called "strong growth condition" (SGC) (Vaswani et al., 2019). When $\|\nabla f(x)\| = 0$, under SGC we have $\|\nabla f_j(x)\| = 0$ for all $j$. SGC is increasingly popular recently e.g.(Schmidt & Roux, 2013; Vaswani et al., 2019). This condition is known to be reasonable in the overparameterized regime where neural networks can interpolate all data points (Vaswani et al., 2019). We will show that Adam can converge to critical points if SGC holds.

When $n, f^*, L, D_0, D_1$ are fixed a priori, we use $\mathcal{F}_{L,D_0,D_1}^{n,f^*}(\mathbb{R}^d)$ to denote the function class containing $f(x)$ satisfying Assumption 2.1 and 2.2 with constant $n, f^*$, etc.. Since $n$ is fixed when the function class $\mathcal{F}_{L,D_0,D_1}^{n,f^*}(\mathbb{R}^d)$, we introduce this notation to clearly present the divergence result in Proposition 3.3. Without this pre-defined function class, the claim of divergence might be confusing.

## 2.2 Related Works

Ever since Reddi et al. (2018) pointed out the divergence issue, there are many attempts on designing new variants of Adam. Since we focus on understanding Adam *without modification* on its update rules, we introduce more variants later in Appendix D.1.

Compared with proposing new variants, the convergence of vanilla Adam is far less studied than its variants (perhaps due to the criticism of divergence). There are only a few works analyzing vanilla Adam and they require extra assumptions. Zhou et al. (2018b) analyze the counter-example in (Reddi et al., 2018) and find certain hyperparameter can work. However, their analysis is restricted to the counter-example. Zaheer et al. (2018) study the relation between mini-batch sizes and (non-)convergence of Adam. However, this work require $\beta_1 = 0$ and Adam is reduced to RMSProp

(Hinton et al., 2012). De et al. (2018) analyze RMSProp and non-zero-$\beta_1$ Adam, but they assume the sign of all stochastic gradients to keep the same. It seems unclear how to check this condition a priori. Additionally, they require $\beta_1$ to be inversely related to the upper bound of gradient, which forces $\beta_1$ to be small (as a side note, this result only applies to full-batch Adam). Défossez et al. (2020) analyze Adam with $\beta_1 < \beta_2$ and provide some insights on the momentum mechanisms. However, their bound is inversely proportional to $\epsilon$ (the hyperparameter for numerical stability) and the bound goes to infinity when $\epsilon$ goes to 0. This is different from practical application since small $\epsilon$ such as $10^{-8}$ often works well. Further, using large $\epsilon$ is against the nature of adaptive gradient methods because $\sqrt{v}$ no longer dominates in the choice of stepsize. In this case, Adam is essentially transformed back to SGD. Two recent works (Huang et al., 2021) and (Guo et al., 2021) propose novel and simple frameworks to analyze Adam-family with large $\beta_1$. Yet, they require the effective stepsize of Adam to be bounded in certain interval, i.e., $\frac{1}{\sqrt{v_t}+\epsilon} \in [C_l, C_u]$ [3]. This boundedness condition changes Adam into AdaBound (Luo et al., 2019) and thus they cannot explain the observations on original Adam in Section 1. To summarize, all these works require at least one strong assumption (e.g. large $\epsilon$). Additionally, they all (including those for new variants) require bounded gradient assumptions.

A recent work (Shi et al., 2020) takes the first attempt to analyze RMSProp without bounded gradient assumption. They show that RMSProp can converge to the neighborhood of critical points. [4] We believe it is important to study Adam rather than RMSProp: Numerically, Adam often outperforms RMSProp on complicated tasks (e.g. on Atari games, the mean reward is improved from 88% to 110% (Agarwal et al., 2020)). Theoretically, literature on RMSProp cannot reveal the interaction between $\beta_1$ and $\beta_2$; or how these hyperparameters jointly affect (or jeopardize) the convergence of Adam. However, it is non-trivial to jointly analyze the effect of $\beta_1$ and $\beta_2$. We point out there are at least three challenges. First, it seems unclear how to control the massive momentum $m_t$ of Adam. Second, $m_t$ is multiplied by $1/\sqrt{v_t}$, causing non-linear perturbation. Third, $m_t$ and $1/\sqrt{v_t}$ are statistically dependent and cannot be decoupled. We propose new methods to resolve these issues. We highlight our technical novelties in Section 5.

## 2.3 The Importance and Difficulties of Removing Bounded Gradient Assumptions

Here, we emphasize the importance to remove bounded gradient assumption. First, unlike the assumptions in Section 2.1, bounded gradient is *not* common in SGD analysis. So it is of theoretical interests to remove this condition for Adam. Second, bounded gradient condition rules out the chances of gradient divergence a priori. However, there are numerical evidences showing that Adam's gradient can diverge (see Section 6 and Appendix B). Removing the boundedness assumption helps us point out the divergence and convergence phase transition in the $(\beta_1, \beta_2)$ diagram.

However, it is often difficult to analyze convergence without bounded gradient assumption. First, it is non-trivial to control stochastic momentum. Even for SGD, this task is challenging. For instance, An early paper Bertsekas & Tsitsiklis (2000) analyzed SGD-type methods without any boundedness condition. But it is not until recently that Yu et al. (2019); Liu et al. (2020b); Jin et al. (2022) prove SGDM (SGD with momentum) converges without bounded gradient assumption. Such attempts of removing boundedness assumptions are often appreciated for general optimization problems where "bounded-assumption-free" is considered as a major contribution.

Secondly, for Adam, the role of momentum $m_t$ is even more intricate since it is multiplied by $1/\sqrt{v_t}$. Combined with $v_t$, the impact of previous signals not only affect the update direction, but also change the stepsize for each component. Further, both momentum $m_t$ and stepsize $1/\sqrt{v_t}$ are random variables and they are highly correlated. Such statistical dependency causes trouble for analysis. In summary, the role of momentum in Adam could be much different from that in SGDM or GDM. Even with boundedness conditions, the convergence of large-$\beta_1$ Adam is still not well understood (see related works in Section 2.2). In this work, we propose new techniques to handle Adam's momentum under any large $\beta_1$, regardless of the gradient magnitude. These techniques are not revealed in any existing works. We introduce our technical contribution in Section 5.

---

[3] For completeness, we explain why they require this condition in Appendix D.1.

[4] We notice that they also provide a convergence result for Adam with $\beta_1$ close enough to 0. However, a simple calculation by Zhang et al. (2022) shows that they require $\beta_1 < 10^{-7}$. Thus their result does not provide much extra information other than RMSProp.

## 3 Main Results

### 3.1 Convergence Results

Here, we give the convergence results under large $\beta_2$.

**Theorem 3.1.** *For any $f(x) \in \mathcal{F}_{L,D_0,D_1}^{n,f^*}(\mathbb{R}^d)$, we assume the hyperparameters in Algorithm 1 satisfy: $\beta_1 < \sqrt{\beta_2} < 1$; $\beta_2$ is greater or equal to a threshold $\gamma_1(n)$; and $\eta_k = \frac{\eta_1}{\sqrt{nk}}$. Let $k_m \in \mathbb{N}$ satisfies $k_m \geq 4$ and $\beta_1^{(k_m-1)n} \leq \frac{\beta_1^n}{\sqrt{k_m-1}}$,[5] we have the following results for any $T > k_m$:*

$$\min_{k\in[k_m,T]} \mathbb{E}\left\{ \min\left[ \sqrt{\frac{2D_1 d}{D_0}}\|\nabla f(x_{k,0})\|_2^2, \|\nabla f(x_{k,0})\|_2 \right] \right\} = \mathcal{O}\left(\frac{\log T}{\sqrt{T}}\right) + \mathcal{O}(\sqrt{D_0}).$$

**Remark 1: the choice of $\beta_2$.** Our theory suggests that large $\beta_2$ should be used to ensure convergence. This message matches our experimental findings in Figure 1. We would like to point out that the requirement of "large $\beta_2$" is neccessary, because small $\beta_2$ will indeed lead to divergence (shown later in Section 3.2). We here comment a bit on the the threshold $\gamma_1(n)$. $\gamma_1(n)$ satisfies $\beta_2 \geq 1 - \mathcal{O}\left(\frac{1-\beta_1^n}{n^2 \rho}\right)$ (see inequality (34) and Remark G.7), where $\rho$ is a constant that depends on the training trajectory. In worst cases, $\rho$ is upper bounded by $n^{2.5}$, but we find the practical $\rho$ to be much smaller. In Appendix B, we estimate $\rho$ on MNIST and CIFAR-10. In practical training process, we empirically observe that $\rho \approx \mathcal{O}(n)$, thus the required $\gamma_1(n) \approx 1 - \mathcal{O}\left(n^{-3}\right)$. Note that our threshold of $\beta_2$ is a sufficient condition for convergence, so there may be a gap between the practical choices and the theoretical bound of $\beta_2$. Closing the gap will be an interesting future direction.

We find that $\gamma_1(n)$ increases with $n$. This property suggests that larger $\beta_2$ should be used when $n$ is large. This phenomenon is also verified by our experiments in Appendix B. We also remark that $\gamma_1(n)$ slowly increases with $\beta_1$. This property is visualized in Figure 1 (d) where the lower boundary of blue region slightly lifts up when $\beta_1$ increases.

**Remark 2: the choice of $\beta_1$.** Theorem 3.1 requires $\beta_1 < \sqrt{\beta_2}$. Since $\beta_2$ is suggested to be large, our convergence result can cover flexible choice of $\beta_1 \in [0,1)$. For instance, $\beta_2 = 0.999$ brings the threshold of $\beta_1 < 0.9995$, which covers basically all practical choices of $\beta_1$ reported in the literature (see Section 1), including the default setting $\beta_1 = 0.9$. This result is much stronger than those in the RMSProp literature (e.g. (Shi et al., 2020; Zaheer et al., 2018)). To our knowledge, we are the first to prove convergence of Adam under any $\beta_1 \in [0,1)$ without bounded gradient assumption.

**Remark 3: convergence to a bounded region.** When $D_0 > 0$, Adam converges to a bounded region near critical points. As discussed in Section 2.1, converging to bounded region is common for stochastic methods including constant-stepsize SGD (Yan et al., 2018; Yu et al., 2019; Liu et al., 2020b) and diminishing-stepsize RMSProp (Zaheer et al., 2018; Shi et al., 2020). This phenomenon is also observed in practice: even for convex quadratic function with $D_0 > 0$, Adam with diminishing stepsize *cannot* reach exactly zero gradient (see Figure 4 (a) in Section 6). This is because: even though $\eta_k$ is decreasing, the effective stepsize $\eta_k/\sqrt{v_{k,i}}$ might not decay. The good news is that, the constant $\mathcal{O}(\sqrt{D_0})$ vanishes to 0 as $\beta_2$ goes to 1 (both in theory and experiments). The relation between $\beta_2$ and constant $\mathcal{O}(\sqrt{D_0})$ are introduced in Remark G.14 in Appendix G.9. The size shrinks to 0 because the movement of $\sqrt{v_{k,i}}$ shrinks as $\beta_2$ increases.

As a corollary of Theorem 3.1, we have the following result under SGC (i.e., $D_0 = 0$).

**Corollary 3.2.** *Under the setting in Theorem 3.1. When $D_0 = 0$ for Assumption 2.2, we have*

$$\min_{k\in[k_m,T]} \mathbb{E}\|\nabla f(x_{k,0})\|_2 = \mathcal{O}\left(\frac{\log T}{\sqrt{T}}\right).$$

Under SGC (i.e. $D_0 = 0$), Corollary 3.2 states that Adam can converge to critical points. This is indeed the case in practice. For instance, function (2) satisfies SGC and we observe 0 gradient norm after Adam converges (see Section 6 and Appendix B). The convergence rate in Corollary 3.2 is comparable to that of SGD under the same condition in (Vaswani et al., 2019).

---

[5]When $\beta_1 = 0.9$, $k_m = 15$ for any $n \geq 1$.

## 3.2 Divergence Results

Theorem 3.1 shows that when $\beta_2$ is large, any $\beta_1 < \sqrt{\beta_2}$ ensures convergence. Now we consider the case where $\beta_2$ is small. We will show that in this case, a wide range of $\beta_1$ is facing the risk of diverging to infinity. The divergence of small-$\beta_2$ Adam suggests that "large $\beta_2$" is necessary in the convergence result Theorem 3.1. We construct a counter-example in $\mathcal{F}_{L,D_0,D_1}^{n,f^*}(\mathbb{R}^d)$. Consider $f(x) = \sum_{i=0}^{n-1} f_i(x)$ for $x \in \mathbb{R}$ , we define $f_i(x)$ as:

$$
f_i(x) = \begin{cases} nx, & x \geq -1 \\ \frac{n}{2}(x+2)^2 - \frac{3n}{2}, & x < -1 \end{cases} \text{ for } i = 0,
$$

$$
f_i(x) = \begin{cases} -x, & x \geq -1 \\ -\frac{1}{2}(x+2)^2 + \frac{3}{2}, & x < -1 \end{cases} \text{ for } i > 0. \tag{2}
$$

Summing up all the $f_i(x)$, we can see that

$$
f(x) = \begin{cases} x, & x \geq -1 \\ \frac{1}{2}(x+2)^2 - \frac{3}{2}, & x < -1 \end{cases}
$$

is a lower bounded convex smooth function with optimal solution $x^* = -2$. Function (2) allows both iterates and gradients to diverge to infinity. As shown in Figure 1 (a), when running Adam on (2), there exists a red large-error region. This shows the sign of divergence. We further theoretically verify the conjecture in Proposition 3.3.

**Proposition 3.3.** *For any function class $\mathcal{F}_{L,D_0,D_1}^{n,f^*}(\mathbb{R}^d)$, there exists a $f(x) \in \mathcal{F}_{L,D_0,D_1}^{n,f^*}(\mathbb{R}^d)$, s.t. when $(\beta_1, \beta_2)$ satisfies analytic condition (12), (13), (14) in Appendix E, Adam's iterates and function values diverge to infinity. By solving these conditions in* `NumPy`*, we plot the orange region in Figure 2. The size of the region depends on $n$ and it expands to the whole region when $n$ goes to infinity.*

The proof can be seen in Appendix E. We find the "divergence region" always stays below the "convergence threshold" $\gamma_1(n)$ in Theorem 3.1, so the two results are self-consistent (see the remark in Appendix E). Proposition 3.3 states the divergence of iterates and function values. Consistently, our experiments also show the divergence of gradient (see Section 6 and Appendix B). These results characterize Adam's divergence behavior both numerically and theoretically.

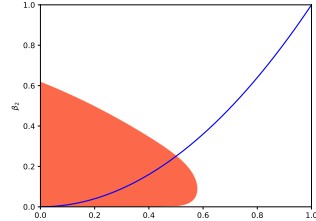

Figure 2: On function (2) with $n = 20$, Adam diverges in the colored region. The region is plotted by solving condition (12), (13), (14) in `NumPy`. The blue curve satisfies $\beta_1 = \sqrt{\beta_2}$.

We emphasize that the orange region is *not* discussed in (Reddi et al., 2018) because we consider $n$ fixed while they allow $n$ changing. When $n$ is allowed to increase, our orange region will expand to the whole region and thus we can derive a similar (actually stronger) result as (Reddi et al., 2018). We provide more explanation in Section 4. Combining Theorem 3.1 and Proposition 3.3, we establish a clearer image on the relation between $(\beta_1, \beta_2)$ and qualitative behavior of Adam.

## 4 Reconciling Our Results with (Reddi et al., 2018)

We discuss more on the relation between (Reddi et al., 2018) and our results. The divergence result shown in Section 1 does not contradict with our convergence results in Theorem 3.1. Further, it is different from our divergence result in Proposition 3.3. The key difference lies in whether $(\beta_1, \beta_2)$ is picked *before or after* picking the function class $\mathcal{F}_{L,D_0,D_1}^{n,f^*}(\mathbb{R}^d)$. We discuss the following two cases.

**Case I: When $(\beta_1, \beta_2)$ is picked before picking $\mathcal{F}_{L,D_0,D_1}^{n,f^*}(\mathbb{R}^d)$.** As discussed in Section 1, the divergence result requires different $n$ for different $(\beta_1, \beta_2)$. In this sense, the considered function class is constantly changing. It does *not* contradict with our Theorem 3.1 which considers a fixed function class with fixed $n$. For **Case I**, we illustrate Adam's behavior in Figure 3. The red region is proved by (Reddi et al., 2018). For completeness, we remove the condition

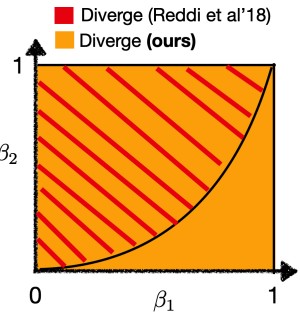

Figure 3: Adam's behavior when $(\beta_1, \beta_2)$ in **Case I**.

"$\beta_1 < \sqrt{\beta_2}$" and further prove that Adam will diverge to infinity for *any* $(\beta_1, \beta_2) \in [0, 1)^2$. The result is shown in the following Corollary 4.1.

**Corollary 4.1.** *For any* $(\beta_1, \beta_2) \in [0, 1)^2$, *there exists a function satisfying Assumption 2.1 and 2.2 that the Adam's iterates and function values diverge to infinity.*

Proof of Corollary 4.1 can be seen in the final paragraph in Appendix E. In the proof, we also require different $n$ to cause divergence for different $(\beta_1, \beta_2)$. So the function class is constantly changing. As a result, in **Case I**, we cannot prove any convergence result.

Table 1: Possible algorithmic behaviors of Adam in **Case II**.

| Setting | Hyperparameters | Adam's behavior |
|---|---|---|
| $\forall f \in \mathcal{F}_{L,D_0,D_1}^{n,f^*}(\mathbb{R}^d)$ with $D_0 = 0$ | $\beta_2$ is large and $\beta_1 < \sqrt{\beta_2}$ | converge to critical points **(Ours)** |
| $\forall f \in \mathcal{F}_{L,D_0,D_1}^{n,f^*}(\mathbb{R}^d)$ with $D_0 \neq 0$ | $\beta_2$ is large and $\beta_1 < \sqrt{\beta_2}$ | converge to a bounded region with size $\mathcal{O}(D_0)$ **(Ours)** |
| $\exists f \in \mathcal{F}_{L,D_0,D_1}^{n,f^*}(\mathbb{R}^d)$ | The orange region in Figure 2 | diverge to infinity **(Ours)** |

**Case II: When $(\beta_1, \beta_2)$ is picked after picking $\mathcal{F}_{L,D_0,D_1}^{n,f^*}(\mathbb{R}^d)$.** When the function class is picked in advance, sample size $n$ will also be fixed. This case is closer to most practical applications. In this case, we find that Adam's behavior changes significantly in the different region of Figure 3. First, $\forall f(x) \in \mathcal{F}_{L,D_0,D_1}^{n,f^*}(\mathbb{R}^d)$ will converge when $\beta_1 < \sqrt{\beta_2}$ and $\beta_2$ is large. Second, $\exists f(x) \in \mathcal{F}_{L,D_0,D_1}^{n,f^*}(\mathbb{R}^d)$ will diverge to infinity when $(\beta_1, \beta_2)$ are in the orange region in Figure 2. Since **Case II** is closer to practical scenarios, these results can provide better guidance for hyperparameter tuning for Adam users. We provide some suggestions for practitioners in Appendix C.

For **Case II**, we summarize the possible behaviors of Adam in Table 1. We also illustrate our convergence and divergence results in Figure 1 (d). Note that there are some blanket areas where Adam's behavior remains unknown, this part will be left as interesting future work.

## 5  Proof Ideas for the Convergence Result

We now (informally) introduce our proof ideas for the convergence result in Theorem 3.1. Simply put, we want to control the update direction $m_{k,i}/\sqrt{v_{k,i}}$ inside the dual cone of gradient direction. Namely:

$$\mathbb{E}\langle \nabla f(x_{k,0}), \sum_{i=0}^{n-1} \frac{m_{k,i}}{\sqrt{v_{k,i}}} \rangle > 0. \tag{3}$$

However, directly proving (3) could be difficult because both $m_{k,i}$ and $v_{k,i}$ distort the trajectory. To start with, we try to control the movement of $v_{k,i}$ by increasing $\beta_2$ (similar idea as (Shi et al., 2020; Zou et al., 2019; Chen et al., 2021)). Recall $v_{k,i} = (1-\beta_2)\sum_{j=1}^{i} \beta_2^{i-j} \nabla f_{\tau_{k,j}}(x_{k,j}) \circ \nabla f_{\tau_{k,j}}(x_{k,j}) + \beta_2^i v_{k,0}$, we have $v_{k,i} \approx v_{k,0}$ when $\beta_2$ is large. In this case, we have:

$$\mathbb{E}\left\langle \nabla f(x_{k,0}), \sum_{i=0}^{n-1} \frac{m_{k,i}}{\sqrt{v_{k,i}}} \right\rangle \approx \mathbb{E}\left\langle \frac{\nabla f(x_{k,0})}{\sqrt{v_{k,0}}}, \sum_{i=0}^{n-1} m_{k,i} \right\rangle \approx \mathbb{E}\left\langle \frac{\nabla f(x_{k,0})}{\sqrt{v_{k,0}}}, \nabla f(x_{k,0}) \right\rangle > 0,$$

where the first "$\approx$" is due to the large $\beta_2$ and the second "$\approx$" is our goal. Now we need to show:

$$\mathbb{E}\left\langle \frac{\nabla f(x_{k,0})}{\sqrt{v_{k,0}}}, \left(\sum_{i=0}^{n-1} m_{k,i}\right) - \nabla f(x_{k,0}) \right\rangle \stackrel{(*)}{=} \mathbb{E}\left( \sum_{l=1}^{d} \sum_{i=0}^{n-1} \frac{\partial_l f(x_{k,0})}{\sqrt{v_{l,k,0}}} \left( m_{l,k,i} - \partial_l f_{\tau_{k,i}}(x_{k,0}) \right) \right) \approx 0, \tag{4}$$

where $\partial_l f(x_{k,0})$ is the $l$-th component of $\nabla f(x_{k,0})$, similarly for $m_{l,k,0}$ and $v_{l,k,0}$. $(*)$ is due to the finite-sum structure. However, it is not easy to prove (4). We point out some technical issues below.

**Issue I: massive momentum.** Directly proving (4) is still not easy. We need to first consider a simplified problem: for every $l \in [d]$, assume we treat $\partial_l f(x_{k,0})/\sqrt{v_{l,k,0}}$ as a constant, how to bound $\mathbb{E}\sum_{i=0}^{n-1} \left( m_{l,k,i} - \partial_l f_{\tau_{k,i}}(x_{k,0}) \right)$?

It turns out that this simplified problem is still non-trivial. When $\beta_1$ is large, $m_{l,k,i}$ contains heavy historical signals which significantly distort the trajectory from gradient direction. Existing literature

(Zaheer et al., 2018; De et al., 2018; Shi et al., 2020) take a naive approach: they set $\beta_1 \approx 0$ so that $m_{l,k,i} \approx \partial_l f_{\tau_{k,i}}(x_{k,i})$. Then we get (4) $\approx 0$. However, this method cannot be applied here since we are interested in practical cases where $\beta_1$ is large in $[0, 1)$.

**Issue II: stochastic non-linear dynamics.** Even if we solve **Issue I**, it is still unclear how to prove (4). This is because: for every $l \in [d]$, $\partial_l f(x_{k,0})/\sqrt{v_{l,k,0}}$ is a r.v. instead of a constant. With this term involved, we are facing with a stochastic non-linear dynamics, which could be difficult to analyze. Further, $\partial_l f(x_{k,0})/\sqrt{v_{l,k,0}}$ is statistically dependent with $(m_{l,k,i} - \partial_l f_{\tau_{k,i}}(x_{k,0}))$, so we are not allowed to handle the expectation $\mathbb{E}(\partial_l f(x_{k,0})/\sqrt{v_{l,k,0}})$ separately .

Unfortunately, even with additional assumptions like bounded gradient, there is no general approach to tackle the above issues. In this work, we propose solutions regardless of gradient magnitude.

**Solution to Issue I.** We prove the following Lemma to resolve **Issue I**.

**Lemma 5.1.** *(Informal) Consider Algorithm 1. For every $l \in [d]$ and any $\beta_1 \in [0, 1)$, we have the following result under Assumption 2.1.*

$$\delta(\beta_1) := \mathbb{E} \sum_{i=0}^{n-1} \left( m_{l,k,i} - \partial_l f_{\tau_{k,i}}(x_{k,0}) \right) = \mathcal{O}\left( \frac{1}{\sqrt{k}} \right),$$

*where $\partial_l f(x_{k,0})$ is the $l$-th component of $\nabla f(x_{k,0})$; $m_{l,k,i} = (1 - \beta_1)\partial_l f_{\tau_{k,i}}(x_{k,i}) + \beta_1 m_{l,k,i-1}$.*

We present the proof idea in Appendix A. Simply put, we construct a simple toy example called "color-ball" model (of the 1st kind). This toy model shows a special property of $\delta(\beta_1)$. We find out: for Algorithm 1, error terms from successive epochs can be canceled, which keeps the momentum roughly in the descent direction. This important property is not revealed in any existing work.

**Remark 4:** When assuming bounded gradient $\|\nabla f(x)\| \leq G$, a naive upper bound would be $\delta(\beta_1) = \mathcal{O}(G)$. However, such constant upper bound does not imply $\delta(\beta_1)$ is close to 0. It will not help prove the convergence. This might be partially the reason why large-$\beta_1$ Adam is hard to analyze even under bounded gradient (see related works in Section 2.2). We emphasize Lemma 5.1 holds true regardless of gradient norm, so it could be deployed in both bounded or unbounded gradient analysis.

**Solution to Issue II.** We try to show (4) by adopting Lemma 5.1. However, the direct application cannot work since $\frac{\partial_l f(x_{k,0})}{\sqrt{v_{l,k,0}}}$ is random. Despite its randomness, we find out that when $\beta_2$ is large, the changes of $\frac{\partial_l f(x_{k,0})}{\sqrt{v_{l,k,0}}}$ shrinks along iteration. As such, although $\frac{\partial_l f(x_{k,0})}{\sqrt{v_{l,k,0}}}$ brings extra perturbation, the quantity in (4) share the similar asymptotic behavior as $\delta(\beta_1)$. We prove the following Lemma 5.2.

**Lemma 5.2.** *(Informal) Under Assumption 2.1 and 2.2, consider Algorithm 1 with large $\beta_2$ and $\beta_1 < \sqrt{\beta_2}$. For those $l$ with gradient component larger than certain threshold, we have:*

$$\left| \frac{\partial_l f(x_{k,0})}{\sqrt{v_{k,0}}} - \frac{\partial_l f(x_{k-1,0})}{\sqrt{v_{k-1,0}}} \right| = \mathcal{O}\left( \frac{1}{\sqrt{k}} \right); \tag{5}$$

$$\mathbb{E}\left( \frac{\partial_l f(x_{k,0})}{\sqrt{v_{l,k,0}}} \sum_{i=0}^{n-1} (m_{l,k,i} - \partial_l f_{\tau_{k,i}}(x_{k,0})) \right) = \mathcal{O}\left( \frac{1}{\sqrt{k}} \right). \tag{6}$$

In Appendix A, we introduce how to derive (6) from (5). To do so, we introduce a new type of "color-ball" model (we call it color-ball of the 2nd kind) which adopts the random perturbation of $\frac{\partial_l f(x_{k,0})}{\sqrt{v_{l,k,0}}}$. Understanding color-ball model of the 2nd kind is crucial for proving Lemma 5.2.

We conclude the proof of (4) by some additional analysis on "those $l$ with small gradient component". This case is a bit easier since it reduces to bounded gradient case. For readers who wants to learn more about the idea of tackling **Issue I** and **II**, please refer to Appendix A where we formally introduce the 1st and 2nd kind of color-ball models. Since the whole proof is quite long, we provide a proof sketch in Appendix G.1. The whole proof is presented in Appendix G.

## 6 Experiments

To support our theory, we provide more simulations and real-data experiments. All the experimental settings and hyperparameters are presented in Appendix B.1. We aim to show:

**(I).** When $\beta_2$ is large, a large range of $\beta_1$ gives good performance, including all $\beta_1 < \sqrt{\beta_2}$.

**(II).** When $\beta_2$ is small, a large range of $\beta_1$ performs relatively badly.

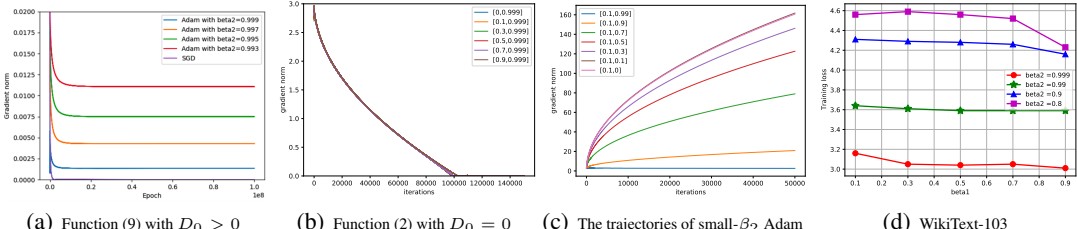

(a) Function (9) with $D_0 > 0$     (b) Function (2) with $D_0 = 0$     (c) The trajectories of small-$\beta_2$ Adam     (d) WikiText-103

Figure 4: The performance of Adam on different tasks. **(a) (b):** large-$\beta_2$ Adam converges to bounded region when $D_0 > 0$ and converges to critical points when $D_0 = 0$. We use diminishing stepsize $\eta_k = 0.1/\sqrt{k}$. **(c):** When $\beta_2$ is small, gradient norm of Adam iterates can be unbounded. We use function 2 with initialization $x = -5$ and $n = 20$. The legends in (b) and (c) stand for $[\beta_1, \beta_2]$. **(d):** The training loss under different $(\beta_1, \beta_2)$ on NLP tasks. We use Adam to train Transformer XL on WikiText-103 dataset.

**Convergence to bounded region when $D_0 > 0$.** In Figure 4, we run large-$\beta_2$ Adam on function (9) (defined later in Appendix B). This function satisfies with $D_0 > 0$. We find that even with diminishing stepsize $\eta_k = 1/\sqrt{k}$, Adam may not converge to an exact critical point. Instead, it converges to a bounded region. This is because: even though $\eta_k$ is decreasing, the effective stepsize $\eta_k/\sqrt{v_{k,i}}$ might not decay. Further, the size of the region shrinks when $\beta_2$ increases. This is because the movement of $\sqrt{v_{k,i}}$ shrinks as $\beta_2$ increases. These phenomena match **Remark 3** and claim **(I)**.

**Convergence to critical points when $D_0 = 0$** Since function (2) satisfies $D_0 = 0$, we run more experiments on (2) with initialization $x = -5$ and $n = 5, 10, 15, 20$. We show the result of $n = 20$ in Figure 4 (a), (b); the rest are shown in Appendix B. We find that: when $\beta_2$ is large, Adam converges to critical points for $\beta_1 < \sqrt{\beta_2}$. These phenomena match claim **(I)**.

**Gradient norm of iterates can be unbounded when $\beta_2$ is small.** On function (2), We further run Adam with small $\beta_2$ at initialization $x = -5$. In this case, gradient norms of iterates increase dramatically. This emphasizes the importance of discarding bounded gradient assumptions. These phenomena match claim **(II)**.

**MNIST and CIFAR-10.** As shown in Figure 1 (b)& (c) in Section 1, the training results match both claim **(I)** and **(II)**. In addition, there is a convex-shaped boundary on the transition from low loss to higher loss, this boundary roughly matches the condition in Theorem 3.1.

**NLP.** We use Adam to train Transformer XL (Dai et al., 2019) on the WikiText-103 dataset (Merity et al., 2016). This architecture and dataset is widely used in NLP tasks (e.g. (Howard & Ruder, 2018; See et al., 2017)). As shown in Figure 4 (d), the training results match both claim **(I)** and **(II).**

## 7 Conclusions

In this work, we explore the (non-)convergence of Adam. When $\beta_2$ is large, we prove that Adam can converge with any $\beta_1 < \sqrt{\beta_2}$. When $\beta_2$ is small, we further show that Adam might diverge to infinity for a wide range of $\beta_1$. One interesting question is to verify the advantage of Adam over SGD. In this work, we focus on the fundamental issue of convergence. Proving faster convergence of Adam would be our future work.

## Acknowledgments and Disclosure of Funding

Yushun Zhang would like to thank Bohan Wang, Reviewer xyuf, Reviewer UR9H, Reviwer tmNN and Reviewer V9yg for the careful proof reading and helpful comments. Yushun Zhang would like to thank Bohan Wang for the valuable discussions around Lemma G.3. Yushun Zhang would also like to thank Reviewer UR9H for the valuable discussions around Lemma G.13. This work is supported by the Internal Project Fund from Shenzhen Research Institute of Big Data under Grant J00220220001. This work is supported by NSFC-A10120170016, NSFC-617310018 and the Guandong Provincial Key Laboratory of Big Data Computing.

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
