## Negative Social Impact

This script may provide better guidance for neural nets training. It would have certain negative social impact if neural nets are deployed for illegal usage.

## Appendix Organization

The Appendix is organized as follows.

- Appendix A introduces two "color-ball" models and their applications in proving Lemma 5.1 and Lemma 5.2. This part is important for proving Theorem 3.1.
- Appendix B introduces more experiments to support our theory.
- Appendix C provide some suggestions for hyperparameter tuning of Adam.
- Appendix D.1 provide more discussions on some recent related works. Appendix D.2 re-state the non-convergence results in (Reddi et al., 2018).
- Appendix E provides detailed proof for Proposition 3.3.
- Appendix F provides some more notations and technical lemmas that serve for the proof of Theorem 3.1.
- Appendix G provides detailed proof for Theorem 3.1. Especially, Appendix G.1 provide a proof roadmap.

## A  Introduction to the "Color-Ball" Models: the Key Ingredients to Prove Theorem 3.1

We now introduce two "color-ball" models and their applications in tackling **Issue I** and **Issue II** mentioned in Section 5. These two color-ball models are important for proving Theorem 3.1.

**Solution to Issue I.**    As discussed in Section 5, we wish to show that $\delta(\beta_1)$ vanishes with $k$. Formally, we wish to get the following equation (7) for every $l \in [d]$.

$$\delta(\beta_1) = \left| \mathbb{E}\left[ \sum_{i=0}^{n-1} (m_{l,k,i} - \partial_l f_i(x_{k,0})) \right] \right| = \mathcal{O}(\beta_1^{nk}), \quad \forall \beta_1 \in [0,1) \tag{7}$$

When $k$ is large, $\mathcal{O}(\beta_1^{nk})$ vanishes faster than $\mathcal{O}(\frac{1}{\sqrt{k}})$ and thus Lemma 5.1 can be proved. In the following context, we will carefully quantify the mismatch between $m_{l,k,i}$ and the $\partial_l f(x_{k,0})$. We find out that the error terms from successive epochs can be cancelled, which keeps the momentum roughly in the descent direction. To help readers understand our idea, we introduce the "color-ball " model (of the 1st kind) as follows.

**The color-ball model of the 1st kind.**    Consider a box containing two balls labeled with constant $c_0, c_1 \in \mathbb{R}$, respectively. In each round (epoch), we randomly sample balls from the box without replacement, then we put them back. We denote the 1st sampled label in the $k$-th epoch as $a_k$ and the 2nd sampled one as $b_k$; $a_k, b_k \in \{c_0, c_1\}$. We define two random variables $m_0$ and $m_1$ as follows (assume $\beta \in [0,1)$):

$$m_1 = \underbrace{b_k + \beta a_k}_{m_{1,k}} + \underbrace{\beta^2 b_{k-1} + \beta^3 a_{k-1}}_{m_{1,k-1}} + \cdots + \underbrace{\beta^{2(k-1)} b_1 + \beta^{2(k-1)+1} a_1}_{m_{1,1}};$$

$$m_0 = \underbrace{a_k}_{m_{0,k}} + \underbrace{\beta^1 b_{k-1} + \beta^2 a_{k-1}}_{m_{0,k-1}} + \cdots + \underbrace{\beta^{2(k-1)-1} b_1 + \beta^{2(k-1)} a_1}_{m_{1,1}};$$

where $m_{0,k}$ denotes the summand of $m_0$ in $k$-th epoch, similarly for $m_{1,k}$. Note that in each epoch, $m_0$ and $m_1$ share the same sample order. Further, we introduce the following deterministic constants.

$$f_1 = c_1(\underbrace{1 + \beta}_{f_{1,k}} + \underbrace{\beta^2 + \beta^3}_{f_{1,k-1}} \cdots + \underbrace{\beta^{2(k-1)} + \beta^{2(k-1)+1}}_{f_{1,1}});$$

$$f_0 = c_0(\underbrace{1 + \beta}_{f_{0,k}} + \underbrace{\beta^2 + \beta^3}_{f_{0,k-1}} \cdots + \underbrace{\beta^{2(k-1)} + \beta^{2(k-1)+1}}_{f_{0,1}});$$

where $f_{0,k}$ denotes the summand of $f_0$ in $k$-th epoch, similarly for $f_{1,k}$. Now we prove the following Lemma A.1.

**Lemma A.1.** *In the color-ball model of the 1st kind, we have*

$$\left| \mathbb{E} \left[ \sum_{i=0}^{1} m_i - \sum_{i=0}^{1} f_i \right] \right| = \beta^{2(k-1)+1} \left( \frac{c_0}{2} + \frac{c_1}{2} \right),$$

*where the expectation is taken on all the possible draws. For the color-ball example with $n \geq 2$ balls, we have*

$$\left| \mathbb{E} \left[ \sum_{i=0}^{n-1} m_i - \sum_{i=0}^{n-1} f_i \right] \right| = \beta^{n(k-1)} \sum_{i=0}^{n-1} c_i(\frac{1}{n}\beta^1 \cdots + \frac{n-1}{n}\beta^{n-1}).$$

**How is Lemma A.1 related to (7)?** In this color-ball toy example, $m_i$ mimics the possible realization of momentum up to the $i$-th inner loop in $k$-th epoch. $f_i$ mimics the stochastic gradient $\nabla f_i(x_{k,0})$. This is because we can expand $\nabla f_i(x_{k,0})$ into an infinite-sum sequence $\nabla f_i(x_{k,0}) = (1 - \beta_1)\nabla f_i(x_{k,0}) \sum_{j=0}^{\infty} \beta_1^j$, which shares a similar structure as $f_i$. In this sense, Lemma A.1 may provide ideas to prove (7). Nevertheless, there are still gap between these two, we will explain the gap later.

*Proof.* We use $\mathbb{E}_k[\cdot]$ to denote the conditional expectation given all the history up to the beginning of $k$-th epoch. Since $\mathbb{E}[\cdot] = \mathbb{E}[\mathbb{E}_k[\cdot]]$, we first calculate $\mathbb{E}_k\left[\sum_{i=0}^{1} m_i\right]$. Since all the history before $k$-th epoch is fixed, we relegate this part to later discussion and first focus on the expectation of $\sum_{i=0}^{1} m_{i,k}$. As shown in Figure 5 (upper part), there are 2 possible realization.

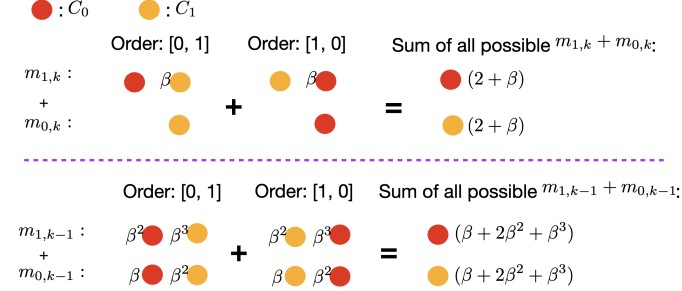

Figure 5: All possible realization of $\sum_{i=0}^{1} m_{i,k}$ and $\sum_{i=0}^{1} m_{i,k-1}$.

With the help of Figure 5 (upper part), we have the following result.

$$\mathbb{E}_k \left[ \sum_{i=0}^{1} m_{i,k} - \sum_{i=0}^{1} f_{i,k} \right] = \mathbb{E}_k \left[ \sum_{i=0}^{1} m_{i,k} \right] - (1+\beta)(c_0 + c_1)$$

$$= -\frac{\beta}{2}(c_0 + c_1) \qquad (8)$$

Now we move one step further to calculate $\mathbb{E}_{k-1}\mathbb{E}_k\left[\sum_{i=0}^1 m_i - \sum_{i=0}^1 f_i\right]$. Using the similar strategy as (8), we have the following result. The calculation is illustrated in Figure 5 (lower part) and Figure 6.

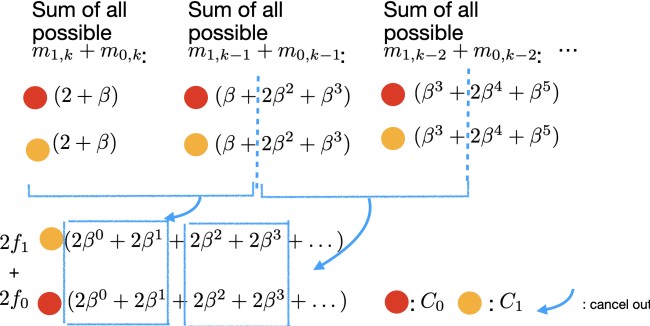

Figure 6: For every $k$, $\mathbb{E}_k\left[\sum_{i=0}^1 m_{i,k} - \sum_{i=0}^1 f_{i,k}\right]$ will create residues, while these residues will be canceled out in the $(k-1)$-th epoch.

$$\mathbb{E}_{k-1}\left\{\mathbb{E}_k\left[\sum_{i=0}^1 m_{i,k} - \sum_{i=0}^1 f_{i,k} + \sum_{i=0}^1 m_{i,k-1} - \sum_{i=0}^1 f_{i,k-1}\right]\right\}$$

$$\overset{(8)}{=} \quad -\frac{\beta}{2}(c_0 + c_1) + \mathbb{E}_{k-1}\left\{\sum_{i=0}^1 m_{i,k-1}\right\} - \sum_{i=0}^1 f_{i,k-1}$$

$$\overset{\text{Figure 5 and 6}}{=} \quad -\frac{\beta}{2}(c_0 + c_1) + \frac{1}{2}(\beta + 2\beta^2 + \beta^3)(c_0 + c_1) - (\beta^2 + \beta^3)(c_0 + c_1)$$

$$= \quad -\frac{\beta^3}{2}(c_0 + c_1)$$

We observe that *only the highest order term remains in the calculation.* Repeat this process until $k = 1$, we will get the results in Lemma A.1. The above analysis also holds for general $n \geq 2$.

$\square$

**The gap between Lemma A.1 and Equation** (7). Lemma A.1 shows the key idea of proving equation (7). However, due to its idealized setting, the color-ball toy example is still far from our real goal (7). We list some of the gaps here.

- In each possible trajectory: $x_{k,i}$ is changing with $k$ and $i$, while the balls are fixed in the color-ball example.

- When taking expectation: $x$ is changing in different trajectory, while the balls is fixed in the color-ball example.

- $x$ is a vector in $\mathbb{R}^d$ while the label of the balls are constant in $\mathbb{R}$.

It requires extra technical lemmas to handle these differences. We provide more discussions in Appendix G.1.

**Solution to Issue II.** We now discuss how to resolve **Issue II** mentioned in Section 5. The solution contains two parts. First, we need to prove (5). Second, we derive (6) from (5). Due to the limited space, we relegate the first part to Appendix G.1 (related to Lemma G.2). Now, we discuss the second part: Assume we have (5), how do we use it to prove (6)? To answer this question, we introduce the color-ball model of the 2nd kind.

**The color-ball model of the 2nd kind.** Consider the same setting as the color-ball model in **Step 1**. We define a sequence of real random variable $\{r_j\}_{j=1}^k$ with the following relation.

$$|r_j - r_{j-1}| = \frac{1}{\sqrt{j}}, \quad j = 1, \cdots k.$$

Further, we assume $r_j$ is fixed when fixing the history up to $j$-th round. The sequence $\{r_j\}_{j=1}^k$ mimics the sequence $\{\frac{\partial_l f(x_{k,0})}{\sqrt{v_{l,k,0}}}\}_{k=1}^\infty$ in (5). Now, we consider the following quantities.

$$r_k m_1 = r_k \left( \underbrace{b_k + \beta a_k}_{m_{1,k}} + \underbrace{\beta^2 b_{k-1} + \beta^3 a_{k-1}}_{m_{1,k-1}} + \cdots + \underbrace{\beta^{2(k-1)} b_1 + \beta^{2(k-1)+1} a_1}_{m_{1,1}} \right);$$

$$r_k m_0 = r_k \left( \underbrace{a_k}_{m_{0,k}} + \underbrace{\beta^1 b_{k-1} + \beta^2 a_{k-1}}_{m_{0,k-1}} + \cdots + \underbrace{\beta^{2(k-1)-1} b_1 + \beta^{2(k-1)} a_1}_{m_{1,1}} \right);$$

$$r_k f_1 = r_k \left( c_1 (\underbrace{1 + \beta}_{f_{1,k}} + \underbrace{\beta^2 + \beta^3}_{f_{1,k-1}} \cdots + \underbrace{\beta^{2(k-1)} + \beta^{2(k-1)+1}}_{f_{1,1}}) \right);$$

$$r_k f_0 = r_k \left( c_0 (\underbrace{1 + \beta}_{f_{0,k}} + \underbrace{\beta^2 + \beta^3}_{f_{0,k-1}} \cdots + \underbrace{\beta^{2(k-1)} + \beta^{2(k-1)+1}}_{f_{0,1}}) \right);.$$

We now prove the following Lemma A.2.

**Lemma A.2.** *Consider the color-ball model of the 2nd kind, we have*

$$\mathbb{E}\left[ \sum_{i=0}^1 r_k m_i - \sum_{i=0}^1 r_k f_i \right] = \beta^{2(k-1)+1} \left( -\frac{c_0}{2} - \frac{c_1}{2} \right) + \mathcal{O}(\frac{1}{\sqrt{k}}).$$

*For general $n = 1, 2, 3, \cdots$, we have*

$$\mathbb{E}\left[ \sum_{i=0}^{n-1} r_k m_i - \sum_{i=0}^{n-1} r_k f_i \right] = \sum_{i=0}^{n-1} c_i \beta^{(k-1)n} (-\frac{1}{n}\beta^1 \cdots - \frac{n-1}{n}\beta^{n-1}) + \mathcal{O}(\frac{1}{\sqrt{k}}).$$

*Proof.* We only describe the proof idea here. The proof contains the following 4 steps.

**Step 2.1.** We firstly take $\mathbb{E}_k[\cdot]$ and thus $r_k$ can be viewed as a constant. We use the color-ball procedure as in Figure 7 to calculate $\mathbb{E}_k \left[ r_k \sum_{i=0}^1 m_{i,k} - r_k \sum_{i=0}^1 f_i \right]$.

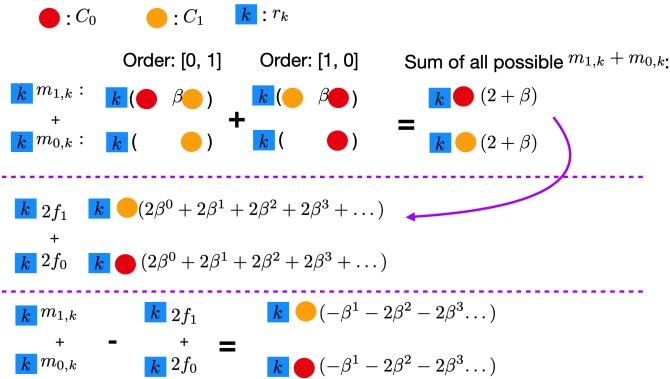

Figure 7: **Step 2.1** in the new color-ball model.

**Step 2.2.** We change $\mathbb{E}_k\left[r_k \sum_{i=0}^{1} m_{i,k-1}\right]$ into $\mathbb{E}_k\left[r_{k-1} \sum_{i=0}^{1} m_{i,k-1}\right]$ + Error; where Error = $\mathcal{O}(1/\sqrt{k})$.

**Step 2.3.** We calculate $\mathbb{E}_{k-1}\mathbb{E}_k\left[r_{k-1}\sum_{i=0}^{1} m_{i,k-1}\right] = \mathbb{E}_{k-1}\left[r_{k-1}\sum_{i=0}^{1} m_{i,k-1}\right]$. This part is illustrated in the top row in Figure 8

**Step 2.4.** For the leftovers in **Step 2.1**, we change all $r_k$ into $r_{k-1}$. Then we do the cancellation to calculate $\mathbb{E}_{k-1}\mathbb{E}_k\left[r_{k-1}\sum_{i=0}^{1} m_{i,k-1} + r_{k-1}\sum_{i=0}^{1} m_{i,k} - r_{k-1}\sum_{i=0}^{1} f_i\right]$. This step is shown in the second and third row in Figure 8.

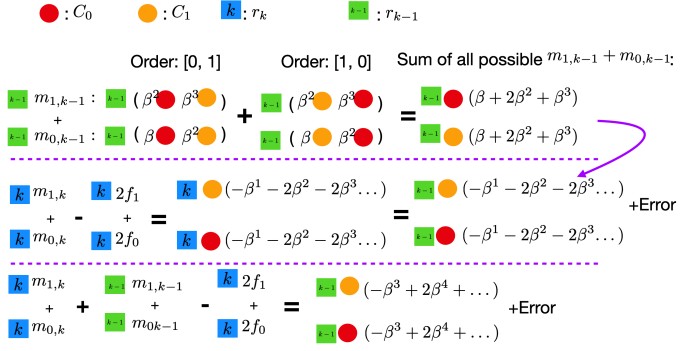

Figure 8: **Step 2.3 and 2.4** in the new color-ball model.

Repeat this process to the 1st epoch. We can prove the Lemma A.2.

$\square$

We emphasize that here are still some gap between Lemma A.2 and our goal in (6) in Lemma 5.2. First, we have the similar gap as discussed at the end of the **Solution to Issue I**. Second, the condition in Lemma 5.2 has requirement on the gradient norm, while this requirement is temporarily ignored in the color-ball method of the 2nd kind. It requires some technical lemmas to handle these gaps.

For more details, please refer to the complete proof in Appendix G.

## B More experiments

**Estimation on $\rho$ in Theorem 3.1.** To ensure convergence, Theorem 3.1 requires $\beta_2 \geq \gamma_1(n) = 1 - \mathcal{O}((1-\beta_1^n)/n^2\rho)$. Now we estimate the constant $\rho$. According to our definition in Appendix F.1 and Remark G.7 in Appendix G.1, $\rho = \rho_1\rho_2\rho_3$, where $\rho_1, \rho_2, \rho_3$ are defined as follows.

$$\rho_1 \geq \frac{\sum_{i=1}^{n}|\partial_l f_i(x_{k,0})|}{\sqrt{\sum_{i=1}^{n}|\partial_l f_i(x_{k,0})|^2}};$$

$$\rho_2 \geq \frac{|\max_i \partial_l f_i(x_{k,0})|^2}{\frac{1}{n}\sum_{i=1}^{n}|\partial_l f_i(x_{k,0})|^2};$$

$$\rho_3 \geq \frac{|\sum_{i=1}^{n}\partial_l f_i(x_{k,0})|}{\sqrt{\frac{1}{n}\sum_{i=1}^{n}|\partial_l f_i(x_{k,0})|^2}}.$$

These constants are firstly introduced by (Shi et al., 2020). In worst case, we have $0 \leq \rho_3 \leq \sqrt{n}\rho_1 \leq n$. However, $\rho$ is highly dependent on the problem instance $f(x)$ and training process. We now estimate how $\rho$ changes with Adam's trajectory on MNIST and CIFAR-10. We use $\beta_1 = 0.9$, $\beta_2 = 0.99$. On both datasets, we set batchsize to be 64, which brings $n = 937$ on MNIST and

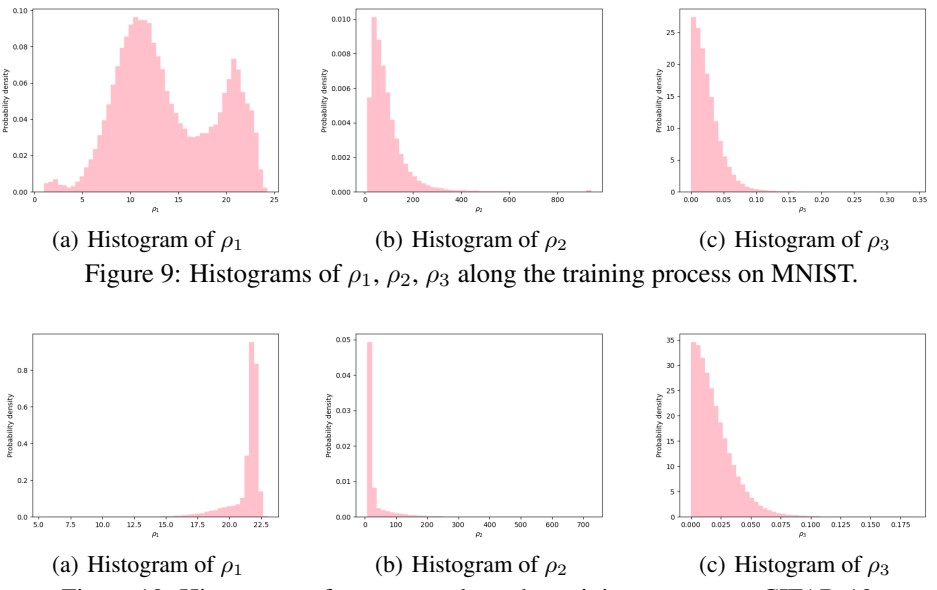

(a) Histogram of $\rho_1$      (b) Histogram of $\rho_2$      (c) Histogram of $\rho_3$

Figure 9: Histograms of $\rho_1$, $\rho_2$, $\rho_3$ along the training process on MNIST.

(a) Histogram of $\rho_1$      (b) Histogram of $\rho_2$      (c) Histogram of $\rho_3$

Figure 10: Histograms of $\rho_1$, $\rho_2$, $\rho_3$ along the training process on CIFAR-10.

$n = 781$ on CIFAR-10. We collect $\rho_1, \rho_2, \rho_3$ along the training process and estimate their distribution density. The results are shown in Figure 9 and 10.

On both CIFAR-10 and MNIST, we observe that the maximal $\rho_1 < 25 \approx \mathcal{O}(\sqrt{n})$, $\rho_2 < 400 \approx \mathcal{O}(n)$, $\rho_3 < 0.1 \approx \mathcal{O}(1/\sqrt{n})$. Therefore, $\rho = \rho_1 \rho_2 \rho_3 \approx \mathcal{O}(n)$.

**Batchsize and $\beta_2$.** As shown in Figure 11: on MNIST, smaller batchsize requires larger $\beta_2$ to reach small loss. Since batchsize equals to (number of total sample)/(number of batches). In the context of finite-sum setting with $n$ summand, $n$ usually stands for the number of batches (e.g., In the extreme case when batchsize $= 1$, $n$ equals to the number of total samples). Therefore, smaller batchsize brings larger $n$. As such, Figure 11 matches the message by Theorem 3.1: the threshold of $\beta_2$ increases with $n$.

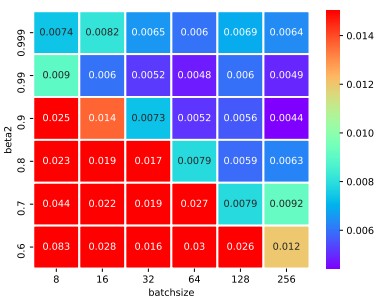

Figure 11: The training loss on MNIST under different batchsize and $\beta_2$. Here, $\beta_1$ is fixed to be 0.9

**More experiments on function** (2)**.** Figure 1 (a) shows the optimality gap after 50k iterations when $n = 10$ and initialization $x = 1$. Here, we provide more relevant experiments. First, when initialized at $x = 1$, we run experiments with $n = 5, 10, 15, 20$. The results are shown in Figure 12. We observe that the blue region shrinks as $n$ increases. This matches our conclusion in Theorem 3.1: when $n$ increases, the convergent threshold of $\beta_2$ increases, which means we need larger $\beta_2$ to ensure convergence. It also matches the conclusion in Theorem 3.3: when $n$ increases, the divergence region will expand (more evidence can be seen in Appendix E).

When initializing at $x = -5$, we further demonstrate that the gradient norm of $f(x)$ can dramatically increase. All the setting is the same as that in Figure 12 except for the change of initialization.

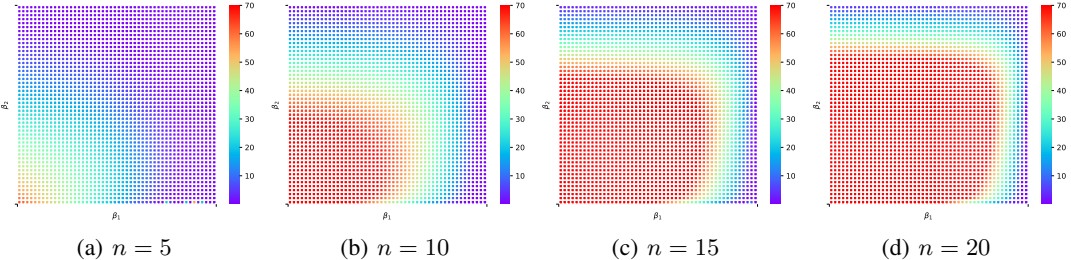

(a) $n = 5$      (b) $n = 10$      (c) $n = 15$      (d) $n = 20$

Figure 12: The optimality gap $x - x^*$ after running 50k iterations of Adam on function (2). We use initialization $x = 1$.

Starting at $x = -5$, the algorithm can touch the "quadratic" side of function (2) where the gradient is unbounded. The results are shown in Figure 13. We observe a similar pattern as that in Figure 12. As a result, the gradient norm of $f(x)$ is large in the left bottom corner.

We further plot the change of gradient norm along the iterations. We pick $\beta_1 = 0.1$ and $\beta_2 = 0, 0.1, 0.3, 0.5, 0.7, 0.9, 0.99$ to see the phase transition when increasing $\beta_2$. The result is shown in Figure 14. When $\beta_2$ is small, the gradient norm of $f(x)$ increases rapidly along the iteration. Most of them are even much larger than the upper bound of color bar in Figure 13. As a result, there is a phase transition from diverge to converge when increasing $\beta_2$ from 0 to 1.

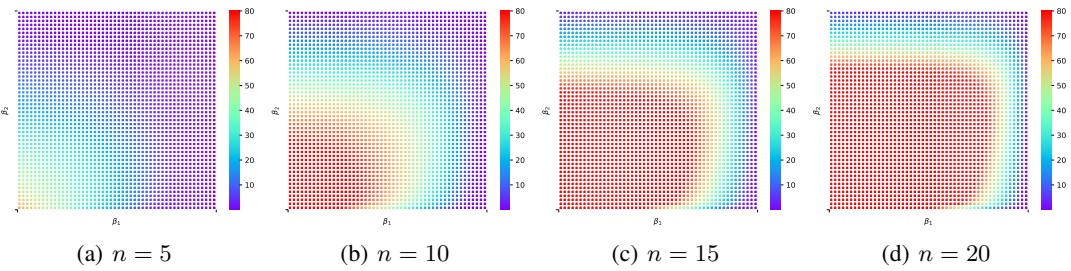

(a) $n = 5$      (b) $n = 10$      (c) $n = 15$      (d) $n = 20$

Figure 13: The gradient norm of $f(x)$ after running 50k iterations of Adam on function (2). We use initialization $x = -5$.

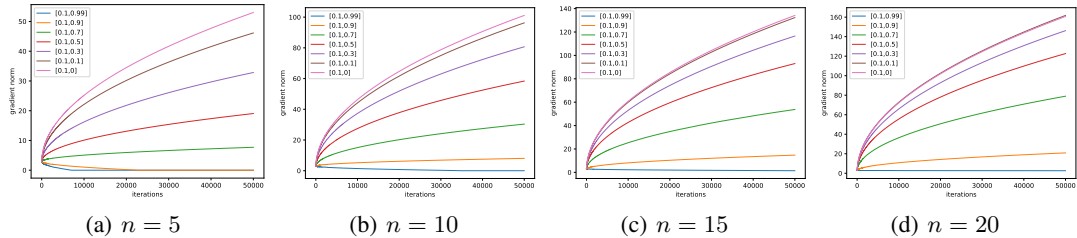

(a) $n = 5$      (b) $n = 10$      (c) $n = 15$      (d) $n = 20$

Figure 14: The change of gradient norm along the iterations of Adam on function (2). We use initialization $x = -5$. We use $[\beta_1, \beta_2]$ to label the curves trained with corresponding hyperparameters.

**Adam converges to a exact critical point when $D_0 = 0$.** Since function (2) satisfies $D_0 = 0$, we further provide empirical evidence that Adam converges to a exact critical point when $D_0 = 0$. We use function (2) $n = 20$ and initialization $x = -5$. We choose some large enough $\beta_2$ to ensure the convergence. As shown in Figure 15, We observe 0 gradient norm after Adam converges. All the hyperparameter setting is the same as before.

**Adam converges to a bounded region when $D_0 > 0$.** Now we show that Adam may not converge to an exact critical point when $D_0 > 0$. Instead, it converges to a bounded region near the critical point. For this part, we re-state the example from (Shi et al., 2020). Consider the following function.

$$f_j(x) = \begin{cases} (x - a)^2 & \text{if } j = 0 \\ -0.1\left(x - \frac{10}{9}a\right)^2 & \text{if } 1 \leq j \leq 9 \end{cases} \tag{9}$$

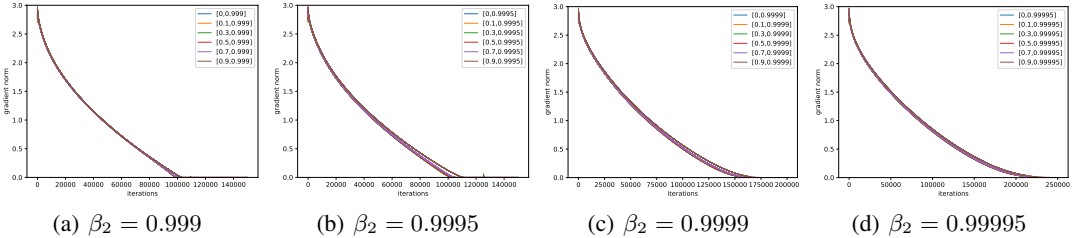

|   (a) $\beta_2 = 0.999$   |   (b) $\beta_2 = 0.9995$   |   (c) $\beta_2 = 0.9999$   |   (d) $\beta_2 = 0.99995$ |

Figure 15: Under SGC (When $D_0 = 0$), we observe 0 gradient norm after Adam converges. We use function (2) with $n = 20$ and initialization $x = -5$. We use $[\beta_1, \beta_2]$ to label the curves trained with corresponding hyperparameters.

Summing up $f_j(x)$ we get

$$f(x) = \sum_{j=0}^{9} f_j(x) = \frac{1}{10}x^2 - \frac{1}{9}a^2.$$

This $f(x)$ is lower bounded by $\frac{-a^2}{9}$ with optimal solution $x^* = 0$. At the optimal point $x^* = 0$, $\nabla f_j(x^*) \neq 0$ so we have $D_0 > 0$. When running Algorithm 1 on this function, we observe that Adam with diminishing stepsize does not converge to 0 gradient norm. Instead, it converge to a bounded region. Further, the size of the region shrinks when $\beta_2$ increases. These phenomena matches our discussion in **Remark 3**. The result is shown in Figure 4 (a). In the experiment, we choose $\beta_1 = 0.9$, $a = 3$, $x_0 = -2$ and diminishing stepsize $\eta_k = \frac{0.1}{\sqrt{k}}$.

### B.1 Experimental Settings

Here, we introduce our experimental settings.

- **Experiments on function** (2)**.** We use Algorithm 1 with cyclic order $f_0$, $f_1$, $f_2$ and so on. We report the optimality gap $x - x^*$ after 50k iteration, or equivalently $50000/n$ epochs. We use $\epsilon = 10^{-8}$ for numerical stability. We use diminishing stepsize $\eta_k = 0.1/\sqrt{k}$, where $k$ is the index of epoch. Unless otherwise stated, this setting applies to all the other experiments on function (2). In Figure 1 (a), we use $n = 10$ and initialization $x = 1$. We will report more results with different $n$ and different initialization in Appendix B.

- **MNIST (Deng, 2012).** We use one-hidden-layer neural network with width =16. We set batchsize =1, weight decay =0, stepsize =0.0001 and train for 20 epochs. We use $\epsilon = 10^{-8}$ for numerical stability.

- **CIFAR-10 (Krizhevsky et al., 2009).** We use ResNet-18 (He et al., 2016) as the architecture. We choose batchsize =16, weight decay =5e-4 , initial stepsize=1e-3. We use a stage-wise constant learning rate scheduling with a multiplicative factor of 0.1 on epoch 30, 60 and 90. We use $\epsilon = 10^{-8}$ for numerical stability.

  For MNIST and CIFAR-10, larger batchsize will bring similar pattern as that in Figure 1, but the phase transition will occur at some smaller $\beta_2$.

- **NLP.** The WikiText-103 dataset is a collection of over 100 million tokens extracted from the set of verified 'Good' and 'Featured' articles on Wikipedia. The base model of Transformer XL contains 16 self-attention layers. In each self-attention layer, there are 10 heads. The encoding dimension of each head is set to be 41. We set batchsize = 60, number of iteration = 200k, and initial stepsize = 0.00025. We use cosine learning rate scheduler, which is a popular choice for training Transformer. We use $\epsilon = 10^{-8}$ for numerical stability.

## C    Some Potential Implications for Practical Users

Theorem 3.1 and Proposition 3.3 establish a clearer image on the relation between $(\beta_1, \beta_2)$ and qualitative behavior of Adam, which may have certain implication for practical users. Many practitioners

are still doing grid or random search over $(\beta_1, \beta_2)$, which could be costly. The following advice may cut down a large portion of search space. Suppose we start with some random point $(\beta_1^*, \beta_2^*)$, we provide the following suggestions for tuning $\beta_1$ and $\beta_2$.

**Case 1: If $(\beta_1^*, \beta_2^*)$ lies above the blue curve in Figure 2.** We point out two sub-cases. First, if Adam with $(\beta_1^*, \beta_2^*)$ diverges, then any points with "$\beta_1 < \sqrt{\beta_2}$" and "$\beta_2 < \beta_2^*$" shall not be tried due to the risk of divergence. Second, if Adam with $(\beta_1^*, \beta_2^*)$ converges, then any points above this point will converge. You may either fix $\beta_1 = \beta_1^*$ and increase $\beta_2$, or fix $\beta_2 = \beta_2^*$ and try smaller $\beta_1$. Both ways have convergence guarantee.

**Case 2: If $(\beta_1^*, \beta_2^*)$ lies below the blue curve in Figure 2.** We still discuss two sub-cases. First, if you observe divergence at $(\beta_1^*, \beta_2^*)$, then do not further try any point on the left. We do not suggest exploring the points in below either, since the majority of them will still face the risk of divergence. Instead, we suggest fix $\beta_1 = \beta_1^*$ and increase $\beta_2$. Since our Theorem 3.1 applies for any $\beta_1 < \sqrt{\beta_2}$, algorithm will converge when $\beta_2$ is large enough.

Second, if you observe convergence at $(\beta_1^*, \beta_2^*)$ (which is also possible according to Figure 1 (a)), then we suggest: (i) either fix $\beta_1 = \beta_1^*$ and increase $\beta_2$, or (ii) fix $\beta_2 = \beta_2^*$ and try smaller $\beta_1$. Both ways push $(\beta_1, \beta_2)$ into the region of Theorem 3.1 with convergence guarantee.

# D    More Discussion on Related Works.

## D.1    More Related Works on the Convergence Analysis of Adam Family

**New variants of Adam.**    Ever since Reddi et al. (2018) pointed out the non-convergence issue of Adam, one active line of work has tried to design new variants of Adam that can be proved to converge. For instance, Zou et al. (2019); Gadat & Gavra (2020); Chen et al. (2018b, 2021) replace the constant hyperparameters by iterate-dependent ones e.g. $\beta_{1t}$ or $\beta_{2t}$. AMSGrad (Reddi et al., 2018) and AdaFom (Chen et al., 2018b) modify $\{v_t\}$ to be an non-decreasing sequence. Similarly, AdaBound (Luo et al., 2019) impose lower and upper bounds on $\{v_t\}$ to prevent the effective stepsize from vanishing or exploding. Zhou et al. (2018b) also adopt a new estimate of $v_t$ to correct the bias. There are also attempts to combine Adam with Nesterov momentum (Dozat, 2016) as well as warm-up techniques (Liu et al., 2020a). Padam (Chen et al., 2018a) also introduce a partial adaptive parameter to improve the generalization performance. There are also some works providing theoretical analysis on the variants of Adam. For instance, Zhou et al. (2018a) study the convergence of AdaGrad and AMSGrad under bounded gradient condition. Gadat & Gavra (2020) study the asymptotic behavior of a subclass of adaptive gradient methods from landscape point of view. Their analysis applies to Adam-variants with $\beta_1 = 0$ and $\beta_2$ increasing along the iterates (it could also be understood as RMSProp with increasing $\beta_2$). When this script is under review, a new work (Iiduka, 2022) appear on line. Iiduka (2022) analyze the convergence of AMSGrad by relaxing the Lipschitz-gradient condition. However, their analysis requires extra conditions on both bounded gradient and bounded domain.

**Some more discussions on (Guo et al., 2021) and (Huang et al., 2021).**    Here, we discuss more on two recent works Guo et al. (2021) and Huang et al. (2021). As mentioned in Section 2.2, they both require some extra conditions. First, both Guo et al. (2021) and Huang et al. (2021) requires bounded gradient assumption. This can be seen in Assumption 2 in (Guo et al., 2021). In (Huang et al., 2021), they require bounded iterates ( their Theorem 1) or change Adam into AdaBound (Luo et al., 2019) by clipping (their Remark 2, Corollary 1). Both settings inherent boundedness on gradient.

Besides bounded gradient, both (Huang et al., 2021) and (Guo et al., 2021) requires $1/(\sqrt{v_t}+\epsilon) \leq C_u$. This condition is stated in Assumption 2 in (Guo et al., 2021) and Assumption 3 in (Huang et al., 2021) (they presented it as $H_t \succeq \rho I_d \succ 0$, where matrix $H_t = \text{diag}(\sqrt{v_t}+\epsilon)$). Combining these two conditions, the effective stepsize of Adam will be bounded in certain interval $\frac{1}{\sqrt{v_t}+\epsilon} \in [C_l, C_u]$. Such boundedness condition changes Adam into AdaBound (Luo et al., 2019) and thus cannot explain the observations of Adam in Section 1.

## D.2 A Brief Introduction to (Reddi et al., 2018)

Here, we re-state two counter examples by (Reddi et al., 2018). For the consistence of notation, we will re-state their results under our notation in the full script. They consider the convex problem ((Reddi et al., 2018)): $\min \sum_{i=0}^{n-1} f_i(x)$ where $x \in [-1, 1]$, $n \geq 3$:

$$f_i(x) = \begin{cases} nx, & \text{for } i = 0 \\ -x, & \text{otherwise}, \end{cases} \tag{10}$$

Note that (10) satisfy both Assumption 2.1 and 2.2 (with $D_1 = n^2 + n - 1$ and $D_0 = 0$), so our assumptions do not rule out this counter-example a priori. This is a constrained problem with feasible set $x \in [-1, 1]$, the optimal solution is $x^* = -1$. Since they consider constrained problems, their claimed "divergence" actually means the iterates will stay in a huge region with the size of whole feasible set. Here, we call it "non-convergence" to distinguish from our result of "diverge to infinity" in Proposition 3.3.

They consider sampling $f_i$ in the cyclic order: $f_0$, $f_1$, $f_2$. In (Reddi et al., 2018), Function (10) is presented as an "online optimization problem with non-zero average regret". We choose to use the form of (10) since it is more consistent with our notation in Algorithm 1. We re-state their results as follows.

**Theorem D.1** (Theorem 2 in (Reddi et al., 2018)). *For any fixed $(\beta_1, \beta_2)$ s.t. $\beta_1 < \sqrt{\beta_2}$, there exists function (10) with large enough $n$, s.t. Adam will converge to highly sub-optimal solution $x = 1$.*

We briefly re-state the non-convergent condition for this Theorem. As stated in Equation (7), Appendix B in (Reddi et al., 2018), for every fixed $(\beta_1, \beta_2)$, they need a "constant $n$ that depends on $\beta_1$ and $\beta_2$". As such, they require different $n$ to cause non-convergence on different $(\beta_1, \beta_2)$. So the considered function class is constantly changing.

For completeness, we further re-state Theorem 1 in (Reddi et al., 2018).

**Theorem D.2** (Theorem 1 in (Reddi et al., 2018)). *For function (10), when $\beta_1 = 0$ and $\beta_2 = 1/(n^2 + 1)$, Adam will converge to highly sub-optimal solution $x = 1$.*

This theorem considers choosing $(\beta_1, \beta_2)$ after $n$. However, this result only show non-convergence on *a single point* $(\beta_1, \beta_2) = (0, 1/(n^2 + 1))$. This point lies somewhere on the left boundary of Figure 3. It seems unclear how Adam's behavior would change as we change the $(\beta_1, \beta_2)$ to anywhere else.

# E  Proof of Proposition 3.3

We restate our counter-example here. Consider $f(x) = \sum_{i=0}^{n-1} f_i(x)$ for $x \in \mathbb{R}$ , we define $f_i(x)$ as follows:

$$\begin{aligned} f_i(x) &= \begin{cases} nx, & x \geq -1 \\ \frac{n}{2}(x+2)^2 - \frac{3n}{2}, & x < -1 \end{cases} \quad \text{for } i = 0, \\ f_i(x) &= \begin{cases} -x, & x \geq -1 \\ -\frac{1}{2}(x+2)^2 + \frac{3}{2}, & x < -1 \end{cases} \quad \text{for } i > 0. \end{aligned} \tag{11}$$

Summing up all the $f_i(x)$, we can see that

$$f(x) = \begin{cases} x, & x \geq -1 \\ \frac{1}{2}(x+2)^2 - \frac{3}{2}, & x < -1 \end{cases}$$

is a convex smooth function with optimal solution $x^* = -2$ and optimal value $f(x^*) = -3/2$.

However, we are going to show that, for any fixed $n > 2$, there exists an orange region shown in Figure 2, s.t., Adam with any $(\beta_1, \beta_2)$ combination in the yellow region diverge to $x = \infty$ rather than the optimal solution $x = -2$, causing the divergence. Now we introduce the formal statement of Proposition 3.3.

**Proposition E.1. (Formal statement of Proposition 3.3.)** *Consider the convex function* (11) *for a fixed* $n$. *Starting at the initialization* $x = 1$ *and initial stepsize* $\eta_1$, *the iterates of Adam diverge to infinity if the following holds:*

$$(\mathbf{C1}) : \left( n - 1 - \min\left\{ n - 1, \log_{\beta_2}\left(\frac{1}{10n^2}\right) \right\} \right) \frac{1 - \beta_1^{\min\left\{ n-1, \log_{\beta_2}\left(\frac{1}{10n^2}\right) \right\}}}{\sqrt{1 + \max\left\{ 0.1, \beta_2^{n-1}n^2 \right\}}} \geq \frac{1 - \beta_1}{\sqrt{1 - \beta_2}} + \frac{\beta_1}{\sqrt{1 - \beta_2}}n;$$

(12)

$$(\mathbf{C2}) : (1 - \beta_1^{n-1}) > (1 - \beta_1)\beta_1^{n-1}n.$$

(13)

$$(\mathbf{C3}) : \eta_1 \leq 2\sqrt{(1 - \beta_2)\beta_2^n}.$$

(14)

The analysis is motivated by that in (Reddi et al., 2018, Theorem 1). However, (Reddi et al., 2018, Theorem 1) considers a simplified case with $\beta_1 = 0$. Here, we consider non-zero $\beta_1$, especially for those $\beta_1 > \sqrt{\beta_2}$. To show the divergence, we aim to prove the following claim: (we denote $x_{k,i}$ as the value of $x$ at the $k$-th outer loop and $i$-th inner loop )

> Claim: for any fixed $n > 2$, there exists an orange region shown in Figure 2 s.t., Adam with any $\beta_1$-$\beta_2$ combination in the orange region gives $x_{k+1,0} > 1$ as long as $x_{k,0} = 1$.

Since the gradient stays constant when $x > 1$, so $x$ will go to infinity if the claim holds. To prove this claim, we only need to analyze the trajectory of Adam within one particular outer loop, e.g., the $k$-th outer loop. We will show that $x_{k+1,0} > 1$ if this outer loop is initialized with $x_{k,0} = 1$. Similarly as (Reddi et al., 2018), we assume $f_i(x)$ are sampled in the order of $f_0(x), f_1(x), \cdots, f_{n-1}(x)$ within the $k$-th outer loop.

Now let us prove the claim. For function (11), the update rule of Adam is shown as follows.

$$x_{k,1} = (x_{k,0} + \delta_{k,0}), \quad \delta_{k,0} = -\frac{\eta_1}{\sqrt{k}} \left( \frac{n(1 - \beta_1) + \beta_1 m_{k-1,n-1}}{\sqrt{(1 - \beta_2)n^2 + \beta_2 v_{k-1,n-1}}} \right)$$

(15)

$$x_{k,i+1} = (x_{k,i} + \delta_{k,i}), \quad i = 1, \cdots, n - 1;$$

(16)

where $\delta_{k,i} = -\frac{\eta_1}{\sqrt{k}} \left( \frac{(1-\beta_1)\sum_{j=0}^{i-1}(-1)\beta_1^j + (1-\beta_1)\beta_1^i n + \beta_1^{i+1}m_{k-1,n-1}}{\sqrt{(1-\beta_2) + \beta_2 v_{k,i-1}}} \right)$.

We decompose the total movement $\sum_{i=0}^{n-1} \delta_{k,i}$ into three terms as follows.

$$
\begin{aligned}
\sum_{i=0}^{n-1} \delta_{k,i} = {} & \frac{\eta_1}{\sqrt{k}} \underbrace{\left( -\frac{\beta_1 m_{k-1,n-1}}{\sqrt{(1-\beta_2)n^2 + \beta_2 v_{k-1,n-1}}} - \frac{\beta_1^2 m_{k-1,n-1}}{\sqrt{(1-\beta_2) + \beta_2 v_{k,0}}} - \cdots - \frac{\beta_1^n m_{k-1,n-1}}{\sqrt{(1-\beta_2) + \beta_2 v_{k,n-2}}} \right)}_{(I)} \\
& + \frac{\eta_1}{\sqrt{k}} \underbrace{\left( \frac{1 - \beta_1}{\sqrt{(1-\beta_2) + \beta_2 v_{k,0}}} + \frac{(1-\beta_1) + \beta_1(1-\beta_1)}{\sqrt{(1-\beta_2) + \beta_2 v_{k,1}}} + \cdots + \frac{(1-\beta_1)\sum_{j=0}^{n-2}\beta_1^j}{\sqrt{(1-\beta_2) + \beta_2 v_{k,n-2}}} \right)}_{(II)} \\
& + \frac{\eta_1}{\sqrt{k}} \underbrace{\left( -\frac{n(1 - \beta_1)}{\sqrt{(1-\beta_2)n^2 + \beta_2 v_{k-1,n-1}}} - \frac{n(1-\beta_1)\beta_1}{\sqrt{(1-\beta_2) + \beta_2 v_{k,0}}} - \cdots - \frac{n(1-\beta_1)\beta_1^{n-1}}{\sqrt{(1-\beta_2) + \beta_2 v_{k,n-2}}} \right)}_{(III)}.
\end{aligned}
$$

We will show that for some $\beta_1$ and $\beta_2$: $(I), (II) > 0$ and $(III) < 0$. However, $(I)$ and $(II)$ outweigh $(III)$, causing the divergence.

First, we show that $m_{k-1,n-1} < 0$ when $\beta_1$ is small.

$$-m_{k-1,n-1} = (1-\beta_1)\sum_{j=0}^{n-2}\beta_1^j - (1-\beta_1)\beta_1^{n-1}n - \beta_1^n m_{k-2,n-1}$$

$$= (1-\beta_1^{n-1}) - (1-\beta_1)\beta_1^{n-1}n - \beta_1^n m_{k-2,n-1}$$

$$= \left[(1-\beta_1^{n-1}) - (1-\beta_1)\beta_1^{n-1}n\right]\sum_{j=0}^{k}(\beta_1^n)^j .$$

when $\beta_1$ is small, we have $(1-\beta_1^{n-1}) > (1-\beta_1)\beta_1^{n-1}n$, which implies $-m_{k-1,n-1} > 0$. For these choices of $\beta_1$, we have $(I) > 0$. Now we derive a lower bound for $(II)$.

$$(II) \geq \frac{1-\beta_1}{\sqrt{1+\beta_2 n^2}} + \frac{(1-\beta_1)+\beta_1(1-\beta_1)}{\sqrt{1+\beta_2^2 n^2}} + \cdots + \frac{(1-\beta_1)\sum_{j=0}^{n-2}\beta_1^j}{\sqrt{1+\beta_2^{n-1}n^2}}$$

$$= \frac{1-\beta_1}{\sqrt{1+\beta_2 n^2}} + \frac{1-\beta_1^2}{\sqrt{1+\beta_2^2 n^2}} + \cdots + \frac{1-\beta_1^{n-1}}{\sqrt{1+\beta_2^{n-1}n^2}}.$$

The inequality is due to the fact that $v_{k,0} \leq n^2$. Since $\beta_2^j n^2$ is small when $\beta_2$ is small and $j$ is close to $n$, there exists some small $\beta_2$ such that $\beta_2^j n^2 \leq 0.1$ for at least one $j < n$. For these small enough $\beta_2$, we keep the summand with $j \geq \log_{\beta_2}(0.1/n^2)$ and drop the rest. We have the following lower bound for $(II)$.

$$(II) \geq \left(n-1-\log_{\beta_2}(\frac{1}{10n^2})\right)\frac{1-\beta_1^{\log_{\beta_2}(\frac{1}{10n^2})}}{\sqrt{1+0.1}}$$

However, this lower bound only holds for the small $\beta_2$. With simple modification, we derive a universal lower bound of $(II)$ for any $\beta_2 \in (0,1)$.

$$(II) \geq \left(n-1-\min\left\{n-1,\log_{\beta_2}(\frac{1}{10n^2})\right\}\right)\frac{1-\beta_1^{\min\{n-1,\log_{\beta_2}(\frac{1}{10n^2})\}}}{\sqrt{1+\max\{0.1,\beta_2^{n-1}n^2\}}}$$

Now we derive a lower bound for $(III)$.

$$(III) = -\frac{n(1-\beta_1)}{\sqrt{(1-\beta_2)n^2+\beta_2 v_{k-1,n-1}}} - \frac{n(1-\beta_1)\beta_1}{\sqrt{(1-\beta_2)+\beta_2 v_{k,0}}} - \cdots - \frac{n(1-\beta_1)\beta_1^{n-1}}{\sqrt{(1-\beta_2)+\beta_2 v_{k,n-2}}}$$

$$\geq -\frac{1-\beta_1}{\sqrt{1-\beta_2}}\left(1+n(\sum_{j=1}^{n-1}\beta_1^j)\right)$$

$$= -\frac{1-\beta_1}{\sqrt{1-\beta_2}} - \frac{\beta_1(1-\beta_1^{n-1})}{\sqrt{1-\beta_2}}n$$

$$\geq -\frac{1-\beta_1}{\sqrt{1-\beta_2}} - \frac{\beta_1}{\sqrt{1-\beta_2}}n.$$

The remaining step is to show for small enough step size $\eta_1$, the iterates will stay in the linear region, thus the above gradient holds for all iterates in the trajectory.

As we initial $x$ as $x_0 = 1$, if for all $m < n$ and $k > 0$, $\sum_{i=0}^{m}\delta_{k,i} \geq -2$, we can conclude all iterates in the trajectory stay in the linear region.

Because it holds that $m_{k-1,n-1} \leq 0$ and $v_{k-1,n-1} \geq 0$, we have the following result after dropping most negative terms in the definition of $\delta_{k,i}$:

$$\delta_{k,i} \geq -\frac{\eta_1}{\sqrt{k}} \frac{n(1-\beta_1)\beta_1^i}{\sqrt{n^2(1-\beta_2)\beta_2^i}} \geq -\eta_1 \frac{(1-\beta_1)\beta_1^i}{\sqrt{(1-\beta_2)\beta_2^i}}.$$

Therefore, to make $\sum_{i=0}^m \delta_{k,i} \geq \sum_{i=0}^m -\eta_1 \frac{(1-\beta_1)\beta_1^i}{\sqrt{(1-\beta_2)\beta_2^n}} \geq -2$, we have

$$\eta_1 \leq 2\sqrt{(1-\beta_2)\beta_2^n}.$$

To show the divergence, we want to show that there exists some $\beta_1$ and $\beta_2$ s.t. the both of the following conditions hold:

$$(\mathbf{C1}) : \left(n - 1 - \min\left\{n-1, \log_{\beta_2}\left(\frac{1}{10n^2}\right)\right\}\right) \frac{1 - \beta_1^{\min\left\{n-1, \log_{\beta_2}\left(\frac{1}{10n^2}\right)\right\}}}{\sqrt{1 + \max\left\{0.1, \beta_2^{n-1}n^2\right\}}} \geq \frac{1-\beta_1}{\sqrt{1-\beta_2}} + \frac{\beta_1}{\sqrt{1-\beta_2}}n;$$

$$(\mathbf{C2}) : \beta_1 \text{ is small s.t. } (1 - \beta_1^{n-1}) > (1-\beta_1)\beta_1^{n-1}n.$$

$$(\mathbf{C3}) : \eta_1 \leq 2\sqrt{(1-\beta_2)\beta_2^n}.$$

The proof of Theorem 3.3 is completed. With the help of Python, we visualize the region where $(\mathbf{C1})$ and $(\mathbf{C2})$ hold. The results are shown in Figure 16. We use orange color to indicate the region where $(\mathbf{C1})$ holds. White color is used for the counter part. As for $(\mathbf{C2})$, we use the gray vertical line to indicate the line where $(1 - \beta_1^{n-1}) = (1-\beta_1)\beta_1^{n-1}n$. Note that there are two solutions to this equation: one solution is $\beta_1 = 1$ and the other solution lies in $0 < \beta_1 < 1$, this is why there are two vertical lines in the figure. $(\mathbf{C2})$ holds on the left hand side of the left gray vertical line.

The intersection of two regions will be the region where Adam diverges, which is actually the orange region in Figure 16. As $n$ increases, $(\mathbf{C2})$ holds for a wide range of $\beta_1$, so the grey vertical lines move towards $\beta_1 = 1$ and finally get overlapped. In addition, the size of divergence region increases with $n$

**Relation with $\gamma_1(n)$ in Theorem 3.1** According to Theorem 3.1, $\gamma_1(n)$ is at least in the order of $1 - \mathcal{O}(n^{-2})$. Combining with Figure 16. It is not hard to see that $\gamma_1(n)$ is always larger than the upper boundary of the orange region, so there is no contradiction.

**Proof of Corollary 4.1.** For any $(\beta_1, \beta_2) \in [0, 1)^2$, condition $(\mathbf{C1}), (\mathbf{C2})$ and $(\mathbf{C3})$ can be satisfied by some sufficiently large $n$. Therefore, Adam will diverge and the proof is concluded.

# F  Some More Notations and Useful Lemmas for Convergence Analysis

## F.1  More notations

- We denote $x_{k,i}, m_{k,i}, v_{k,i} \in \mathbb{R}^d$ as the value of $x, m, v$ at the $k$-th outer loop and $i$-th inner loop. Further, we denote $x_{l,k,i}, m_{l,k,i}, v_{l,k,i} \in \mathbb{R}$ as the $l$-th component of $x_{k,i}, m_{k,i}, v_{k,i}$.

- We denote $\eta_k$ as the stepsize at the $k$-th epoch (outer loop). We will focus mainly on diminishing step size, especially $\eta_k = \frac{\eta_1}{\sqrt{nk}}$, where $n$ is the number of batches (inner loop).

- We denote $\partial_l f(x) = \frac{\partial}{\partial x_l} f(x)$, $\partial_l f_j(x) = \frac{\partial}{\partial x_l} f_j(x)$. Further, we will use $\tau_{k,i}$ to index the $i$-th randomly chosen batch in the $k$-th epoch. In this sense, we denote $\partial_l f_{\tau_{k,i}}(x)$ as $\frac{\partial}{\partial x_l} f_{\tau_{k,i}}(x)$.

- Given an epoch $k$, we denote $\alpha$ as the index of the coordinate with the greatest gradient:

$$\alpha = \arg\max_{l=1,2,\cdots,d} |\partial_l f(x_{k,0})|.$$

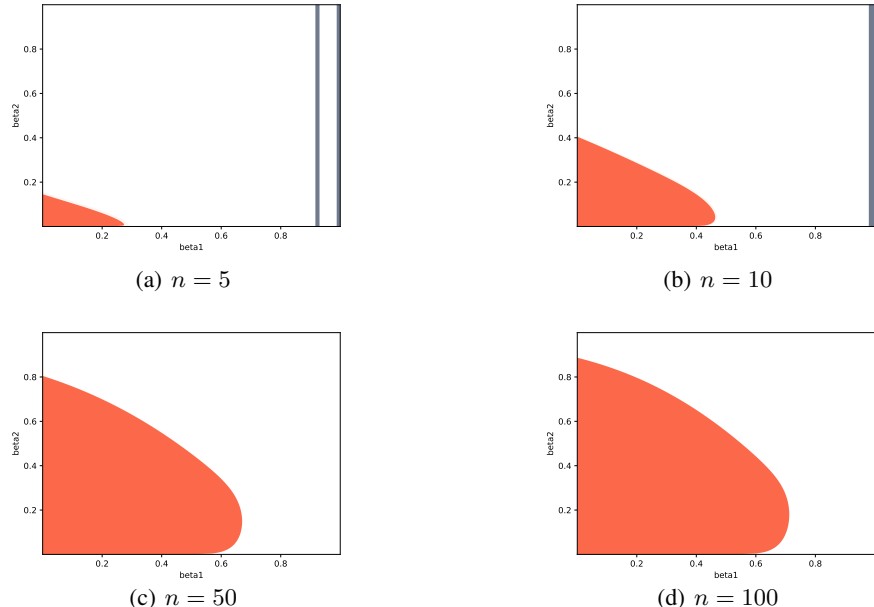

(a) $n = 5$  (b) $n = 10$

(c) $n = 50$  (d) $n = 100$

Figure 16: This figure illustrates the region where both (**C1**) and (**C2**) in Proposition E.1 hold. The orange color indicates the region where (**C1**) holds. White color is used for the counter part. The gray vertical lines are used to indicate the boundary of (**C2**). Note that there are two solutions to the equation in (**C2**): one solution is $\beta_1 = 1$ and the other solution lies in $0 < \beta_1 < 1$, this is why there are two vertical lines in the figure. (**C2**) holds on the left hand side of the left gray vertical line. This figure is visualized in Python.

- We define $\rho_1, \rho_2, \rho_3$ as constants satisfying the following condition for any $l = 1, \cdots, d$:

$$\rho_1 \geq \frac{\sum_{i=1}^{n} |\partial_l f_i(x_{k,0})|}{\sqrt{\sum_{i=1}^{n} |\partial_l f_i(x_{k,0})|^2}};$$

$$\rho_2 \geq \frac{|\max_i \partial_l f_i(x_{k,0})|^2}{\frac{1}{n} \sum_{i=1}^{n} |\partial_l f_i(x_{k,0})|^2};$$

$$\rho_3 \geq \frac{|\sum_{i=1}^{n} \partial_l f_i(x_{k,0})|}{\sqrt{\frac{1}{n} \sum_{i=1}^{n} |\partial_l f_i(x_{k,0})|^2}}.$$

$\rho_1, \rho_2, \rho_3$ are problem-dependent constants. In worst case, we have $0 \leq \rho_3 \leq \sqrt{n}\rho_1 \leq n$. These constants are firstly introduced by (Shi et al., 2020).

- We define the constant $\triangle_x := \frac{\eta_0}{\sqrt{x}} \frac{L\sqrt{d}}{\sqrt{1-\beta_2}} \frac{1-\beta_1}{1-\frac{\beta_1}{\sqrt{\beta_2}}}$. This constant will be used repeatedly, especially in Lemma F.2.

- We define $Q_k := \triangle_1 \frac{n\sqrt{n}}{\sqrt{k}} \frac{32\sqrt{2}}{(1-\beta_2)^n \beta_2^n}$. This constant will be used repeatedly.

- We define

$$\delta_1 = \frac{(1 - \beta_2)4n\rho_2}{\beta_2^n} + \left( \frac{1}{\sqrt{\beta_2^n}} - 1 \right). \tag{17}$$

This constant will be used repeatedly. Note that $\delta_1 \to 0$ when $\beta_2 \to 1$.

- $\mathbb{E}(\cdot)$ means taking expectation over the whole trajectory.

- We will use $\mathbb{E}_k [\cdot]$ as a shorthand for $\mathbb{E}[\cdot \mid x_{k,0}, x_{k-1,n-1}, \cdots, x_{1,0}]$, i.e. the conditional expectation given all the history up to $x_{k,0}$. Similarly, $\mathbb{E}_{k-1}$ stands for the conditional expectation given all the history up to $x_{k-1,0}$.

We further introduce some 'index' notation of the stochastic gradients, these notations will be useful in the proof. For all the stochastic gradient, we have

- **True index:** every stochastic gradient has its own true index: $f_1(\cdot), f_2(\cdot), \cdots, f_{n-1}(\cdot)$. All these $\{f_i(\cdot)\}_{i=0}^{n-1}$ are fixed once the optimization problem is formulated.
  Note that for a fixed $x$, we have the following relationship between $f_i$ and $f$:

$$\sum_{i=0}^{n-1} \partial_l f_i(x) = \partial_l f(x). \tag{18}$$

- **Random-shuffle index:** At the $k$-th epoch, all the stochastic gradients are sampled in the order of $f_{\tau_{k,0}}(\cdot), f_{\tau_{k,1}}(\cdot), \cdots, f_{\tau_{k,n-1}}(\cdot)$.
  Since Algorithm 1 is sampling without replacement, there is an implicit bijective mapping between $\{f_i(\cdot)\}_{i=0}^{n-1}$ and $\{f_{\tau_{k,i}}(\cdot)\}_{i=0}^{n-1}$. Further, for a fixed $x$, we have the following useful property:

$$\sum_{i=0}^{n-1} \partial_l f_{\tau_{k,i}}(x) = \partial_l f(x). \tag{19}$$

- **Index for $m$ and $v$:** As shown in Algorithm 1, we denote :

$$m_{l,k,i} = (1 - \beta_1)\{\partial_l f_{\tau_{k,i}}(x_{k,i}) + \cdots + \beta_1^i \partial_l f_{\tau_{k,0}}(x_{k,0})\} + \beta_1^{i+1} m_{l,k,0},$$
$$v_{l,k,i} = (1 - \beta_2)\{\partial_l f_{\tau_{k,i}}(x_{k,i})^2 + \cdots + \beta_1^i \partial_l f_{\tau_{k,0}}(x_{k,0})^2\} + \beta_2^{i+1} v_{l,k,0}.$$

### F.2   Some useful lemmas

We begin with proving some useful lemmas.

**Lemma F.1.** *For any $\beta \neq 1$, we have*

$$(1 - \beta) \sum_{j=1}^{\infty} \beta^{j-1} = 1,$$

$$(1 - \beta) \sum_{j=1}^{\infty} j\beta^{j-1} = \frac{1}{1 - \beta},$$

$$(1 - \beta) \sum_{j=1}^{\infty} j^2\beta^{j-1} = \frac{1 + \beta}{(1 - \beta)^2}.$$

*Proof.* Proof only involves basic calculation, we omit the proof here. $\square$

**Lemma F.2.** *Under Assumption 2.1, if $\beta_1 < \sqrt{\beta_2}$, we have*

$$\left|\partial_l f_{\tau_{k,i}}(x_{k,i+1}) - \partial_l f_{\tau_{k,i}}(x_{k,i})\right| \leq \frac{\eta_0}{\sqrt{nk}} \frac{L\sqrt{d}}{\sqrt{1 - \beta_2}} \frac{1 - \beta_1}{1 - \frac{\beta_1}{\sqrt{\beta_2}}} = \triangle_{nk}. \tag{20}$$

*Proof.* We start with bounding $|x_{l,k,i+1} - x_{l,k,i}|$. By the update formula of Adam, we have

$$
\begin{aligned}
|x_{l,k,i+1} - x_{l,k,i}| &= \eta_k \frac{|m_{l,k,i}|}{\sqrt{v_{l,k,i}}} \\
&\leq \eta_k(1 - \beta_1) \left\{\beta_1^0 \frac{|\partial_l f_{\tau_{k,i}}(x_{k,i})|}{\sqrt{v_{l,k,i}}} + \beta_1 \frac{|\partial_l f_{\tau_{k,i-1}}(x_{k,i-1})|}{\sqrt{v_{l,k,i}}} + \ldots\right\} \\
&\leq \eta_k(1 - \beta_1) \left\{\beta_1^0 \frac{|\partial_l f_{\tau_{k,i}}(x_{k,i})|}{\sqrt{(1 - \beta_2)\beta_2^0}|\partial_l f_{\tau_{k,i}}(x_{k,i})|} + \beta_1 \frac{|\partial_l f_{\tau_{k,i-1}}(x_{k,i-1})|}{\sqrt{(1 - \beta_2)\beta_2}|\partial_l f_{\tau_{k,i-1}}(x_{k,i-1})|} + \ldots\right\} \\
&\leq \eta_k \frac{(1 - \beta_1)}{\sqrt{1 - \beta_2}} \left(\left(\frac{\beta_1}{\sqrt{\beta_2}}\right)^0 + \left(\frac{\beta_1}{\sqrt{\beta_2}}\right)^1 + \left(\frac{\beta_1}{\sqrt{\beta_2}}\right)^2 + \ldots\right) \\
&= \eta_k \frac{(1 - \beta_1)}{\sqrt{1 - \beta_2}} \frac{1}{1 - \frac{\beta_1}{\sqrt{\beta_2}}}. \tag{21}
\end{aligned}
$$

Now, by Assumption 2.1, we have

$$
\begin{aligned}
\left|\partial_l f_{\tau_{k,i}}(x_{k,i+1}) - \partial_l f_{\tau_{k,i}}(x_{k,i})\right| &\leq \|\nabla f_{\tau_{k,i}}(x_{k,i+1}) - \nabla f_{\tau_{k,i}}(x_{k,i})\|_2 \\
&\leq L\|x_{k,i+1} - x_{k,i}\|_2 \\
&\leq L\sqrt{d}\max_l |x_{l,k,i+1} - x_{l,k,i}| \\
&\leq \eta_k \frac{L\sqrt{d}}{\sqrt{1-\beta_2}}\frac{1-\beta_1}{1-\frac{\beta_1}{\sqrt{\beta_2}}} \\
&= \frac{\eta_0}{\sqrt{nk}}\frac{L\sqrt{d}}{\sqrt{1-\beta_2}}\frac{1-\beta_1}{1-\frac{\beta_1}{\sqrt{\beta_2}}}
\end{aligned}
\tag{22}
$$

Proof is completed.

$\square$

**Lemma F.3.** *Under Assumption 2.1 and 2.2, for any integer $i \in [0, n-1]$, we have the following two results,*

$$
|\partial_l f_{\tau_{k,i}}(x_{k,i})| \leq i\triangle_{nk} + \sqrt{D_1}\rho_1 d\left(|\partial_\alpha f(x_{k,0})| + \sqrt{\frac{D_0}{D_1 d}}\right),
\tag{23}
$$

$$
|\partial_l f_{\tau_{k,i}}(x_{k,0})| \leq \sum_{i=0}^{n-1} |\partial_l f_{\tau_{k,i}}(x_{k,0})| \leq \sum_{l=1}^{d}\sum_{i=0}^{n-1} |\partial_l f_{\tau_{k,i}}(x_{k,0})| \leq \sqrt{D_1}\rho_1 d\left(|\partial_\alpha f(x_{k,0})| + \sqrt{\frac{D_0}{D_1 d}}\right),
\tag{24}
$$

*where $|\partial_\alpha f(x_{k,0})| = \max_{l \in [1,d]} |\partial_l f(x_{k,0})|$.*

*Proof.*

$$
\begin{aligned}
|\partial_l f_{\tau_{k,j}}(x_{k,j})| \quad &\overset{\text{Lemma F.2}}{\leq} \quad j\triangle_{nk} + |\partial_l f_{\tau_{k,j}}(x_{k,0})| \\
&\leq \quad j\triangle_{nk} + \sum_{l=1}^{d}\sum_{j=0}^{n-1} |\partial_l f_{\tau_{k,j}}(x_{k,0})| \\
&\overset{(*)}{\leq} \quad j\triangle_{nk} + \rho_1 \sum_{l=1}^{d}\sqrt{\sum_{i=0}^{n-1} |\partial_l f_i(x_{k,0})|^2} \\
&\overset{\text{Cauchy-Swartz inequality}}{\leq} \quad j\triangle_{nk} + \rho_1\sqrt{d}\sqrt{\sum_{l=1}^{d}\sum_{i=0}^{n-1} |\partial_l f_i(x_{k,0})|^2} \\
&\overset{\text{Assumption 2.2}}{\leq} \quad j\triangle_{nk} + \rho_1\sqrt{d}\sqrt{D_1\|\nabla f(x_{k,0})\|_2^2 + D_0} \\
&\overset{(**)}{\leq} \quad j\triangle_{nk} + \sqrt{D_1}\rho_1 d\sqrt{|\partial_\alpha f(x_{k,0})|^2 + \frac{D_0}{D_1 d}} \\
&\leq \quad j\triangle_{nk} + \sqrt{D_1}\rho_1 d\left(|\partial_\alpha f(x_{k,0})| + \sqrt{\frac{D_0}{D_1 d}}\right)
\end{aligned}
$$

where $(*)$ is because of the definition (see Appendix F.1): $\rho_1$ is a constant satisfying $\rho_1 \geq \frac{\sum_{i=1}^{n} |\partial_l f_i(x_{k,0})|}{\sqrt{\sum_{i=1}^{n} |\partial_l f_i(x_{k,0})|^2}}$; $(**)$ is due to $\|\nabla f(x_{k,0})\|_2^2 \leq d|\partial_\alpha f(x_{k,0})|^2$. The proof of (24) follows the same procedure as above. $\square$

# G  Proof of Theorem 3.1

## G.1  A roadmap of the proof

Our proof is based on the Descent Lemma:

$$
\mathbb{E}f(x_{k+1,0}) - \mathbb{E}f(x_{k,0}) \quad \leq \quad \mathbb{E}\langle \nabla f(x_{k,0}), x_{k+1,0} - x_{k,0}\rangle + \frac{L}{2}\mathbb{E}\|x_{k+1,0} - x_{k,0}\|_2^2
$$

The expectation $\mathbb{E}(\cdot)$ is taken on the whole trajectory. Summing both sides from the initialization $k = t_0$ to $k = T$, we have the following re-arranged inequality: (usually we set $t_0 = 1$.)

$$
\mathbb{E}\sum_{k=t_0}^{T} \langle \nabla f(x_{k,0}), x_{k,0} - x_{k+1,0}\rangle \quad \leq \quad \frac{L}{2}\sum_{k=t_0}^{T} \mathbb{E}\|x_{k+1,0} - x_{k,0}\|_2^2 + \mathbb{E}f(x_{t_0,0}) - \mathbb{E}f(x_{T+1,0}). \quad (25)
$$

To prove the convergence, we need an upper bound for $\frac{L}{2}\mathbb{E}\|x_{k+1,0} - x_{k,0}\|_2^2$, as well as a lower bound for $\mathbb{E}\langle \nabla f(x_{k,0}), x_{k,0} - x_{k+1,0}\rangle$ (and such a lower bound should be in the order of $\frac{1}{\sqrt{k}}\mathbb{E}\|\nabla f(x_{k,0})\|$).

The upper bound for $\frac{L}{2}\mathbb{E}\|x_{k+1,0} - x_{k,0}\|_2^2$ is relatively easy to get: according to the update rule of Adam, this term is in the order of $\mathcal{O}(\eta_k^2 m_{l,k,i}/v_{l,k,i}) = \mathcal{O}(\eta_k^2\|\nabla f(x_{k,0})\|/\|\nabla f(x_{k,0})\|) = \mathcal{O}(\frac{1}{k})$. Further, recall $\sum_{k=t_0}^{T} \frac{1}{k} \leq \log\frac{T+1}{t_0}$, so $\frac{L}{2}\mathbb{E}\|x_{k+1,0} - x_{k,0}\|_2^2$ contributes to the log term in Theorem 3.1. The proof is shown in Lemma F.2.

However, the lower bound for $\mathbb{E}\langle \nabla f(x_{k,0}), x_{k,0} - x_{k+1,0}\rangle = \mathbb{E}\left[\sum_{l=1}^{d}\sum_{i=0}^{n-1} \partial_l f(x_{k,0})\frac{m_{l,k,i}}{\sqrt{v_{l,k,i}}}\right]$ requires sophisticated derivation, we explain as follows. Before taking the expectation, we first work on every possible realization of $\sum_{l=1}^{d}\sum_{i=0}^{n-1} \partial_l f(x_{k,0})\frac{m_{l,k,i}}{\sqrt{v_{l,k,i}}}$. We perform the following decomposition for every $l \in [1, d]$.

$$
\sum_{i=0}^{n-1} \partial_l f(x_{k,0})\frac{m_{l,k,i}}{\sqrt{v_{l,k,i}}} = \sum_{i=0}^{n-1}\frac{\partial_l f(x_{k,0})}{\sqrt{v_{l,k,i}}}\left(\partial_l f_i(x_{k,0}) + m_{l,k,i} - \partial_l f_i(x_{k,0})\right)
$$

$$
= \underbrace{\left[\sum_{i=0}^{n-1}\frac{\partial_l f(x_{k,0})}{\sqrt{v_{l,k,i}}}\partial_l f_i(x_{k,0})\right]}_{(a)} + \underbrace{\left[\partial_l f(x_{k,0})\sum_{i=0}^{n-1}\frac{1}{\sqrt{v_{l,k,i}}}(m_{l,k,i} - \partial_l f_i(x_{k,0}))\right]}_{(b)}.
$$

$$
\quad (26)
$$

First of all, we introduce the following Lemma G.1 to further decompose $(a)$ and $(b)$. The intuition of this decomposition is as follows: by increasing $\beta_2$, we can control the movement of the moving average factor $v_{l,k,i}$ (similar idea as (Shi et al., 2020; Zou et al., 2019; Chen et al., 2021)).

**Lemma G.1.** *Under Assumption 2.1, for those $l$ satisfying $\max_i |\partial_l f_i(x_{k,0})| \geq Q_k := \triangle_1 \frac{n\sqrt{n}}{\sqrt{k}}\frac{32\sqrt{2}}{(1-\beta_2)^n\beta_2^n}$, we have the following lower bound for $(a)$ in (26):*

$$
\sum_{i=0}^{n-1}\frac{\partial_l f(x_{k,0})}{\sqrt{v_{l,k,i}}}\partial_l f_i(x_{k,0}) \quad \geq \quad \frac{\partial_l f(x_{k,0})^2}{\sqrt{v_{l,k,0}}} - \delta_1\left|\frac{\partial_l f(x_{k,0})}{\sqrt{v_{l,k,0}}}\right|\sum_i |\partial_l f_i(x_{k,0})|, \quad (27)
$$

*where $\delta_1 = \frac{(1-\beta_2)4n\rho_2}{\beta_2^n} + \left(\frac{1}{\sqrt{\beta_2^n}} - 1\right)$. $\rho_2, \rho_3$ are constants satisfying $\rho_2 \geq \frac{|\max_i \partial_l f_i(x_{k,0})|^2}{\frac{1}{n}\sum_{i=1}^{n}|\partial_l f_i(x_{k,0})|^2}$, $\rho_3 \geq \frac{|\sum_{i=1}^{n}\partial_l f_i(x_{k,0})|}{\sqrt{\frac{1}{n}\sum_{i=1}^{n}|\partial_l f_i(x_{k,0})|^2}}$. Note that $\delta_1 \to 0$ when $\beta_2 \to 1$.*

*Under the same condition, we also have a lower bound for $(b)$ in (26):*

$$\sum_{i=0}^{n-1} \frac{\partial_l f(x_{k,0})}{\sqrt{v_{l,k,i}}} (m_{l,k,i} - \partial_l f_i(x_{k,0}))$$

$$\geq \frac{\partial_l f(x_{k,0})}{\sqrt{v_{l,k,0}}} \sum_{i=0}^{n-1} (m_{l,k,i} - \partial_l f_i(x_{k,0})) - \delta_1 \left| \frac{\partial_l f(x_{k,0})}{\sqrt{v_{l,k,0}}} \right| \sum_{i=0}^{n-1} |m_{l,k,i} - \partial_l f_i(x_{k,0})|. \quad (28)$$

The proof can be seen in Appendix G.2 and it is motivated from (Shi et al., 2020). Lemma G.1 decomposes both $(a)$ and $(b)$: in the denominator, we approximate $v_{l,k,i}$ (which is changing with $i$) by $v_{l,k,0}$ (which is fixed). Accordingly, the approximation error can be controlled by increasing $\beta_2$ (since $\delta_1 \to 0$ when $\beta_2 \to 1$). However, similarily as (Shi et al., 2020), Lemma G.1 can only be applied to those $l$ with $\max_i |\partial_l f_i(x_{k,0})| \geq Q_k$, so we need to discuss two cases as below.

**Case 1: unbounded gradient.** Given $x_{k,0}$, consider those $l$ with $\max_i |\partial_l f_i(x_{k,0})| \geq Q_k$, we have the following decomposition:

$$\sum_{i=0}^{n-1} \partial_l f(x_{k,0}) \frac{m_{l,k,i}}{\sqrt{v_{l,k,i}}} \overset{(26)}{\geq} \underbrace{\left[ \sum_{i=0}^{n-1} \frac{\partial_l f(x_{k,0})}{\sqrt{v_{l,k,i}}} \partial_l f_i(x_{k,0}) \right]}_{(a)} + \underbrace{\left[ \partial_l f(x_{k,0}) \sum_{i=0}^{n-1} \frac{1}{\sqrt{v_{l,k,i}}} (m_{l,k,i} - \partial_l f_i(x_{k,0})) \right]}_{(b)}.$$

$$\overset{\text{Lemma G.1}}{\geq} \frac{\partial_l f(x_{k,0})^2}{\sqrt{v_{l,k,0}}} - \delta_1 \left| \frac{\partial_l f(x_{k,0})}{\sqrt{v_{l,k,0}}} \right| \sum_{i=0}^{n-1} |\partial_l f_i(x_{k,0})|$$

$$+ \frac{\partial_l f(x_{k,0})}{\sqrt{v_{l,k,0}}} \sum_{i=0}^{n-1} (m_{l,k,i} - \partial_l f_i(x_{k,0})) - \delta_1 \left| \frac{\partial_l f(x_{k,0})}{\sqrt{v_{l,k,0}}} \right| \sum_{i=0}^{n-1} |m_{l,k,i} - \partial_l f_i(x_{k,0})|$$

$$\overset{\text{Lemma G.10}}{\geq} \underbrace{\left[ \frac{\partial_l f(x_{k,0})^2}{\sqrt{v_{l,k,0}}} \right]}_{(a_1)} - \underbrace{\left[ \delta_1 \sqrt{\frac{2\rho_3^2}{\beta_2^n}} \sum_{i=0}^{n-1} |\partial_l f_i(x_{k,0})| \right]}_{(a_2)}$$

$$+ \underbrace{\left[ \frac{\partial_l f(x_{k,0})}{\sqrt{v_{l,k,0}}} \sum_{i=0}^{n-1} (m_{l,k,i} - \partial_l f_i(x_{k,0})) \right]}_{(b_1)} - \underbrace{\left[ \delta_1 \sqrt{\frac{2\rho_3^2}{\beta_2^n}} \sum_{i=0}^{n-1} |m_{l,k,i} - \partial_l f_i(x_{k,0})| \right]}_{(b_2)},$$

$$(29)$$

**Case 2: bounded gradient.** Given $x_{k,0}$, consider those $l$ with $\max_i |\partial_l f_i(x_{k,0})| \leq Q_k$, the analysis degenerates to the "bounded gradient" scenario, we have the following lower bound:

$$\sum_{i=0}^{n-1} \partial_l f(x_{k,0}) \frac{m_{l,k,i}}{\sqrt{v_{l,k,i}}} \geq -\sum_{i=0}^{n-1} |\partial_l f(x_{k,0})| \left| \frac{m_{l,k,i}}{\sqrt{v_{l,k,i}}} \right|$$

$$\overset{(21)}{\geq} -|\partial_l f(x_{k,0})| \frac{1-\beta_1}{\sqrt{1-\beta_2}} \frac{1}{1 - \frac{\beta_1}{\sqrt{\beta_2}}} n$$

$$\overset{(*)}{\geq} -n Q_k \frac{1-\beta_1}{\sqrt{1-\beta_2}} \frac{1}{1 - \frac{\beta_1}{\sqrt{\beta_2}}} n$$

$$= -\triangle_1 \frac{n^2 \sqrt{n}}{\sqrt{k}} \frac{32\sqrt{2}}{(1-\beta_2)^n \beta_2^n} \frac{1-\beta_1}{\sqrt{1-\beta_2}} \frac{1}{1 - \frac{\beta_1}{\sqrt{\beta_2}}} n$$

$$:= -F_1 \frac{1}{\sqrt{k}},$$

where $F_1 := \triangle_1 n^2 \sqrt{n} \frac{32\sqrt{2}}{(1-\beta_2)^n \beta_2^n} \frac{1-\beta_1}{\sqrt{1-\beta_2}} \frac{1}{1-\frac{\beta_1}{\sqrt{\beta_2}}} n$, $(*)$ is due to the fact that $|\partial_l f(x_{k,0})| \leq n \max_i |\partial_l f_i(x_{k,0})| \leq nQ_k$.

Combining **Case 1 & 2** together, we have the following result (note that $(a_1), (a_2), (b_1), (b_2)$ vary with $l$. A more precise notation will be $(a_1)_l$, etc. We drop the subscript for brevity).

$$
\begin{aligned}
\sum_{l=1}^{d} \sum_{i=0}^{n-1} \partial_l f(x_{k,0}) \frac{m_{l,k,i}}{\sqrt{v_{l,k,i}}} &\geq \underbrace{\sum_{l \text{ large}} \{(a_1) - (a_2) + (b_1) - (b_2)\}}_{\text{unbounded gradient}} - \underbrace{\sum_{l \text{ small}} F_1 \frac{1}{\sqrt{k}}}_{\text{bounded gradient}} \\
&\geq \underbrace{\sum_{l \text{ large}} \{(a_1) - (a_2) + (b_1) - (b_2)\}}_{\text{unbounded gradient}} - \underbrace{\sum_{l=1}^{d} \left\{ F_1 \frac{1}{\sqrt{k}} \right\}}_{\text{bounded gradient}} \\
&\geq \left\{ \sum_{l \text{ large}} (a_1) \right\} + \left\{ \sum_{l \text{ large}} (b_1) \right\} - \left\{ \sum_{l \text{ large}} \{(a_2) + (b_2)\} \right\} - dF_1 \frac{1}{\sqrt{k}},
\end{aligned}
$$
(30)

where "$l$ large" stands for the gradient component in **Case 1** and "$l$ small" indicates those in **Case 2**. We assume "$l$ large" is not an empty set, otherwise the analysis degenerates to the case with bounded gradient assumption.

Now we take expectation on (30). The expectation is taken on all the possible trajectories up to the $i$-th iteration in $k$-th epoch.

$$
\mathbb{E} \left[ \sum_{l=1}^{d} \sum_{i=0}^{n-1} \partial_l f(x_{k,0}) \frac{m_{l,k,i}}{\sqrt{v_{l,k,i}}} \right] \overset{(30)}{\geq} \mathbb{E} \left\{ \sum_{l \text{ large}} (a_1) + \sum_{l \text{ large}} (b_1) - \sum_{l \text{ large}} \{(a_2) + (b_2)\} \right\} - dF_1 \frac{1}{\sqrt{k}},
$$
(31)

In the following context, we will discuss how to bound all terms in (31), respectively. First and foremost, we derive a lower bound for $\mathbb{E}\left[\sum_{l \text{ large}}(b_1)\right]$. Since $\mathbb{E}\left[\sum_{l \text{ large}}(b_1)\right]$ contains all the historical gradient information, it involves great effort to handle it. We will show that the lower bound of $\mathbb{E}\left[\sum_{l \text{ large}}(b_1)\right]$ will vanish when $\beta_2$ is large and $k$ goes to infinity.

Recall $\mathbb{E}\left[\sum_{l \text{ large}}(b_1)\right] = \mathbb{E}\left[\sum_{l \text{ large}} \frac{\partial_l f(x_{k,0})}{\sqrt{v_{l,k,0}}} \sum_{i=0}^{n-1} (m_{l,k,i} - \partial_l f_i(x_{k,0}))\right]$. When $\beta_1$ is large, $m_{l,k,i}$ contains heavy historical signals. It seems unclear how large $\mathbb{E}\left[\sum_{l \text{ large}}(b_1)\right]$ would be when $\beta_1$ goes to 1. Existing literatures (Zaheer et al., 2018; De et al., 2018; Shi et al., 2020) take a naive approach: they set $\beta_1 \approx 0$ so that $m_{k,i} \approx \nabla f_{\tau_{k,i}}(x_{k,i})$. Then we get $\delta(\beta_1) \approx 0$. However, this naive method cannot be applied here since we are interested in practical cases where $\beta_1$ is large in $[0, 1)$. We emphasize the following technical difficulties in bounding $\mathbb{E}\left[\sum_{l \text{ large}}(b_1)\right]$ for any $\beta_1 \in [0, 1)$:

- **Issue (i)** In order to bound $\mathbb{E}\left[\sum_{l \text{ large}}(b_1)\right]$, we need to first know how to bound its simpler version: $\mathbb{E}\left[\sum_{i=0}^{n-1}(m_{l,k,i} - \partial_l f_i(x_{k,0}))\right]$. This term measures the difference between the current gradient and weighted historical gradients. It seems unclear that how large this term could be when $\beta_1$ goes to 1.

- **Issue (ii)** Even if we can bound $\mathbb{E}\left[\sum_{i=0}^{n-1}\left(m_{l,k,i}-\partial_l f_i(x_{k,0})\right)\right]$, it is still unclear how to bound $\mathbb{E}\left[\frac{\partial_l f(x_{k,0})}{\sqrt{v_{l,k,0}}}\sum_{i=0}^{n-1}\left(m_{l,k,i}-\partial_l f_i(x_{k,0})\right)\right]$, which is further multiplied by a random variable $\frac{\partial_l f(x_{k,0})}{\sqrt{v_{l,k,0}}}$.

- **Issue (iii)** Even if we can bound $\mathbb{E}\left[\frac{\partial_l f(x_{k,0})}{\sqrt{v_{l,k,0}}}\sum_{i=0}^{n-1}\left(m_{l,k,i}-\partial_l f_i(x_{k,0})\right)\right]$, it is still different from $\mathbb{E}\left[\sum_{l\,\text{large}}(b_1)\right]$ which contains additional operation "$\sum_{l\,\text{large}}$" inside the expectation. Note that the set "$l$ large" is a random variable which changes along different trajectories, so there is still non-negligible gap between bounding $\mathbb{E}\left[\sum_{l\,\text{large}}(b_1)\right]$ and $\mathbb{E}\left[\frac{\partial_l f(x_{k,0})}{\sqrt{v_{l,k,0}}}\sum_{i=0}^{n-1}\left(m_{l,k,i}-\partial_l f_i(x_{k,0})\right)\right]$.

To our best knowledge, there is no general approach to tackle the above issues. In the following content, we will overcome difficulties (i), (ii) and (iii) in Step 1, 2 and 3 respectively. In Step 1, we will discuss how to bound $\mathbb{E}\left[\sum_{i=0}^{n-1}\left(m_{l,k,i}-\partial_l f_i(x_{k,0})\right)\right]$, which is a simplified version of $\mathbb{E}\left[(b_1)\right]$. Bounding this term will shed light on bounding the whole term $\mathbb{E}\left[\sum_{l\,\text{large}}(b_1)\right]$. Then in Step 2 and 3, we will prove several technical lemmas to handle the effect of $\frac{\partial_l f(x_{k,0})}{\sqrt{v_{l,k,0}}}$ and "$\sum_{l\,\text{large}}$", by which we can tackle **(ii)** and **(iii)**. Combining all together we can bound $\mathbb{E}\left[\sum_{l\,\text{large}}(b_1)\right]$.

**Bounding** $\mathbb{E}\left[\sum_{l\,\textbf{large}}(b_1)\right]$**:** **Step 1.** We now introduce the key idea of bounding $\mathbb{E}\left[\sum_{i=0}^{n-1}\left(m_{l,k,i}-\partial_l f_i(x_{k,0})\right)\right]$. In Appendix A, we distill our idea into a toy example called "the color-ball model of the 1st kind" and thus we prove Lemma A.1. This lemma is crucial for bounding $\mathbb{E}\left[\sum_{i=0}^{n-1}\left(m_{l,k,i}-\partial_l f_i(x_{k,0})\right)\right]$. We refer the readers to Appendix A for more explanation.

Lemma A.1 can provide insights in bounding $\mathbb{E}\left[\sum_{i=0}^{n-1}\left(m_{l,k,i}-\partial_l f_i(x_{k,0})\right)\right]$. However, there is still certain gap between $\mathbb{E}\left[\sum_{i=0}^{n-1}\left(m_{l,k,i}-\partial_l f_i(x_{k,0})\right)\right]$ and the quantities in Lemma A.1. We elaborate as follows:

- To mimic the color-ball example, we need to expand $\sum_{i=0}^{n-1}\partial_l f_i(x_{k,0})$ into an *infinite* sum sequence: $\sum_{i=0}^{n-1}\partial_l f_i(x_{k,0})(1-\beta_1)(1+\beta_1+\cdots\beta_1^{\infty})$. However, $\sum_{i=0}^{n-1}m_{l,k,i}$ is a *finite* sum sequence up to the order of $\beta^{kn}$. In contrast, both sequences in the color-ball example are "finite sum". As such, there is an error term caused by "finite sum v.s. infinite sum".

- When taking the expectation, the variable $x$ in each possible trajectory is different. In contrast, in the color-ball example, $\{a_i\}_{i=0}^{2}$ are fixed in all shuffling order (so it is much easier to calculate the expectation by summing them up).

  To mimic the color-ball example, we repeatedly take conditional expectation at the beginning of each $k$-th epoch. In this way, $x_{k,0}$ will be fix. Despite $x_{k,i}$ is still changing across the trajectory, we can transform $x_{k,i}$ into $x_{k,0}$ by using Lipschitz property.

- In each trajectory of $\sum_{i=0}^{n-1}m_{l,k,i}$, the variable $x$ in the summand of $m_{l,k,i}$ varies with $k$ and $i$; while the variable in $\partial_l f_i(x_{k,0})$ is fixed to be $x_{k,0}$. In contrast, in the color-ball example, $\{a_i\}_{i=0}^{2}$ are the same across the epoch.

  To mimic the final step in the color-ball example (Figure 6), at each step of conditional expectation, we need to simultaneously move the variables in $m_{l,k,i}$ and $\partial_l f_i(x_{k,0})$ (using Lipschitz property) so that they can match and cancel out with each other. This operation will introduce new error terms and it is our duty to put them under control.

We omit the proof for bounding $\mathbb{E}\left[\sum_{i=0}^{n-1}\left(m_{l,k,i}-\partial_l f_i(x_{k,0})\right)\right]$ since this is not our actual goal. Instead, we will directly use the above ideas to bound $\mathbb{E}\left[(b_1)\right]$. To do so, we need to further tackle the issue **(ii)** and **(iii)** mentioned before. We explain as follows.

**Bounding** $\mathbb{E}\left[\sum_{l \text{ large}}(b_1)\right]$**: Step 2.** We now resolve issue **(ii)**, i.e., handle the effect of $\frac{\partial_l f(x_{k,0})}{\sqrt{v_{l,k,0}}}$. The key idea is as follows: when we calculate $\mathbb{E}[(b_1)]$, we sequentially take conditional expectation $\mathbb{E}_k(\cdot)$, $\mathbb{E}_{k-1}(\cdot)$, etc.. When taking $\mathbb{E}_k(\cdot)$, we will fix all the historical information up to $k$-th epoch, so $\frac{\partial_l f(x_{k,0})}{\sqrt{v_{l,k,0}}}$ can be regarded as a constant. In this sense, $\mathbb{E}_k\left[\frac{\partial_l f(x_{k,0})}{\sqrt{v_{l,k,0}}}\sum_{i=0}^{n-1}(m_{l,k,i} - \partial_l f_i(x_{k,0}))\right]$ can be calculated following the same idea as color-ball toy example.

However, when taking $\mathbb{E}_{k-1}(\cdot)$, $\frac{\partial_l f(x_{k,0})}{\sqrt{v_{l,k,0}}}$ will become a random variable which changes with different trajectories. In this case, the color-ball method *cannot* be applied. To fix this issue, we introduce the following lemma to change $\frac{\partial_l f(x_{k,0})}{\sqrt{v_{l,k,0}}}$ into $\frac{\partial_l f(x_{k-1,0})}{\sqrt{v_{k-1,0}}}$, which can again be regarded as a fixed constant when taking $\mathbb{E}_{k-1}(\cdot)$.

**Lemma G.2.** *Suppose Assumption 2.1 holds and $\beta_1 < \sqrt{\beta_2}$. For any integer $j \in [0, k]$, if* $\max_i |\partial_l f_i(x_{k,0})| \geq Q_k$, $\max_i |\partial_l f_i(x_{k-1,0})| \geq Q_{k-1}$, $\cdots$, $\max_i |\partial_l f_i(x_{k-j,0})| \geq Q_{k-j}$ *(where* $Q_k := \triangle_1 \frac{n\sqrt{n}}{\sqrt{k}} \frac{32\sqrt{2}}{(1-\beta_2)^n \beta_2^n}$ *), then we have the following result:*

$$
\left| \frac{\partial_l f(x_{k,0})}{\sqrt{v_{l,k,0}}} - \frac{\partial_l f(x_{k-j,0})}{\sqrt{v_{l,k-j,0}}} \right| \leq \frac{1}{1 - \frac{1}{\sqrt{\beta_2^n}}} \frac{n^2 \triangle_{n(k-j)}}{\sqrt{v_{l,k-j,0}}} + j\sqrt{\frac{2\rho_3^2}{\beta_2^n}} \frac{1}{\left(1 - \frac{(1-\beta_2)4n\rho_2}{\beta_2^n}\right)} \delta_1,
$$

*where $\delta_1$ is defined in* (17).

The proof of Lemma G.2 can be seen in Appendix G.3. To proceed, we combine Lemma G.2 and the "color-ball method of the 2nd kind", and thus we can prove Lemma A.2. This lemma is crucial for our current goal: bounding $\mathbb{E}(b_1)$. We refer the readers to Appendix A for more information.

We emphasize that here are still the following gap between Lemma A.2 and our goal $\mathbb{E}(b_1)$.

- We have the similar gap as discussed at the end of the **Step 1**.

- The condition in Lemma G.2 has requirement on the gradient norm, while this requirement is temporarily ignored in the color-ball method of the 2nd kind.

- The result in Lemma G.2 has additional error terms other than $\mathcal{O}(1/\sqrt{k})$. This is slightly different from the setting in Lemma A.2.

It requires some technical lemmas to handle these gaps. We fill in these gaps in Lemma G.4. The technical details can be seen Appendix G.4.

**Bounding** $\mathbb{E}\left[\sum_{l \text{ large}}(b_1)\right]$**: Step 3.** Now we shift gear to tackle (iii): handling the random variable "$l$ large". We rewrite "$l$ large" into the indicator function as follows:

$$
\mathbb{E}\left[\sum_{l \text{ large}}(b_1)\right] = \mathbb{E}\left[\sum_{l \text{ large}} \frac{\partial_l f(x_{k,0})}{\sqrt{v_{l,k,0}}} \sum_{i=0}^{n-1}(m_{l,k,i} - \partial_l f_i(x_{k,0}))\right] = \mathbb{E}\left[\sum_{l=1}^{d} \mathbb{I}_k \frac{\partial_l f(x_{k,0})}{\sqrt{v_{l,k,0}}} \sum_{i=0}^{n-1}(m_{l,k,i} - \partial_l f_i(x_{k,0}))\right],
$$

where $\mathbb{I}_k := \mathbb{I}\left(\max_i |\partial_l f_i(x_{k,0})| \geq Q_k := \triangle_1 \frac{n\sqrt{n}}{\sqrt{k}} \frac{32\sqrt{2}}{(1-\beta_2)^n \beta_2^n}\right)$ is the indicator function ($\mathbb{I}(A) = 1$ when event $A$ holds and $\mathbb{I}(A) = 0$ otherwise.) Similarly as before, when taking $\mathbb{E}_k(\cdot)$, "$\mathbb{I}_k$" can be regarded as a constant index. Therefore, $\mathbb{E}_k\left[\sum_{l=1}^{d} \mathbb{I}_k \frac{\partial_l f(x_{k,0})}{\sqrt{v_{l,k,0}}} \sum_{i=0}^{n-1}(m_{l,k,i} - \partial_l f_i(x_{k,0}))\right]$ can be calculated following the same idea as color-ball model of the 1st kind. However, when taking $\mathbb{E}_{k-1}(\cdot)$, $\mathbb{I}_k$ will become a random variable which changes with different trajectories. In this case, the color-ball method *cannot* be applied. Similarly as in **Step 2**, we introduce the following lemma to change $\mathbb{I}_k$ into $\mathbb{I}_{k-1}$ (defined later), which can again be regarded as a fixed when taking $\mathbb{E}_{k-1}(\cdot)$.

**Lemma G.3.** *Suppose Assumption 2.1 holds and $\beta_1 < \sqrt{\beta_2}$. For $0 \leq j \leq k$, we define* $\mathbb{I}_{k-j} := \mathbb{I}\left(\max_i |\partial_l f_i(x_{k-j,0})| \geq \sum_{p=k-j}^{k} Q_p\right)$, *where* $Q_k := \triangle_1 \frac{n\sqrt{n}}{\sqrt{k}} \frac{32\sqrt{2}}{(1-\beta_2)^n \beta_2^n}$, *then we have the following results.*

$$\mathbb{I}_k = \mathbb{I}\left(\max_i |\partial_l f_i(x_{k,0})| \geq Q_k \text{ and } \max_i |\partial_l f_i(x_{k-j,0})| \geq \sum_{p=k-j}^k Q_p\right)$$

$$+ \mathbb{I}\left(\max_i |\partial_l f_i(x_{k,0})| \geq Q_k \text{ and } \max_i |\partial_l f_i(x_{k-j,0})| \leq \sum_{p=k-j}^k Q_p\right), \quad (32)$$

$$\mathbb{I}\left(\max_i |\partial_l f_i(x_{k,0})| \geq Q_k \text{ and } \max_i |\partial_l f_i(x_{k-j,0})| \geq \sum_{p=k-j}^k Q_p\right) = \mathbb{I}\left(\max_i |\partial_l f_i(x_{k-j,0})| \geq \sum_{p=k-j}^k Q_p\right) = \mathbb{I}_{k-j}.$$
$$(33)$$

*Proof.* Equation (32) is straightforward, we only prove (33) here. Under Assumption 2.1, we have $|\partial_l f_i(x_{k,0}) - \partial_l f_i(x_{k-j,0})| \leq n\triangle_{n(k-1)} + \cdots + n\triangle_{n(k-j)} \leq Q_{k-1} + \cdots + Q_{k-j}$. To show the second inequality, it requires comparing the value between $Q_k$ and $n\triangle_{n(k)}$, which are both problem-dependent constants. Here, the inequality holds when $n\triangle_{n(k)} \leq Q_k$. If otherwise, we can always define $\tilde{Q}_k := \max\{Q_k, n\triangle_{n(k)}\}$ and the inequality still holds by changing all the $Q_k$ into $\tilde{Q}_k$. We temporarily omit this step for now.

Since $|\partial_l f_i(x_{k,0}) - \partial_l f_i(x_{k-j,0})| \leq Q_{k-1} + \cdots + Q_{k-j}$, the event $\left\{\max_i |\partial_l f_i(x_{k-j,0})| \geq \sum_{p=k-j}^k Q_p\right\}$ implies the event $\{\max_i |\partial_l f_i(x_{k,0})| \geq Q_k\}$, so the proof is completed. $\square$

Now we are ready to bound $\mathbb{E}\left[\sum_{l \text{ large}}(b_1)\right]$. Combining **Step 1, 2 and 3** together, we prove the following Lemma G.4.

**Lemma G.4.** *Under Assumption 2.1, consider $\beta_1 < \sqrt{\beta_2}$, when $k$ is large such that: $k \geq 4$; $\beta_1^{(k-1)n} \leq \frac{\beta_1^n}{\sqrt{k-1}}$, we have the following result:*

$$\mathbb{E}\left[\sum_{l \text{ large}}(b_1)\right] \geq -\mathcal{O}\left(\frac{1}{\sqrt{k}}\right) - \mathcal{O}\left(\delta_1 \mathbb{E}\left[\sum_{l=1}^d \sum_{i=0}^{n-1} |\partial_l f_i(x_{k,0})|\right]\right),$$

*where $\delta_1 = \frac{(1-\beta_2)4n\rho_2}{\beta_2^n} + \left(\frac{1}{\sqrt{\beta_2^n}} - 1\right)$, which goes to 0 when $\beta_2$ goes to 1.*

Proof can be seen in Appendix G.4. Now we bound the error terms: $\mathbb{E}\left[\sum_{l \text{ large}}\{(a_2) + (b_2)\}\right]$. Since they are multiplied by $\delta_1$, these two terms vanish when $\beta_2 \to 1$.

**Bounding** $\mathbb{E}\left[\sum_{l \text{ large}}\{(a_2) + (b_2)\}\right]$. We bound these two terms in the following Lemma G.5.

**Lemma G.5.** *Given all the history up to $x_{k,0}$, we denote $\alpha$ as the index of the coordinate with the greatest gradient: $\alpha = \arg\max_{l=1,2,\cdots,d} |\partial_l f(x_{k,0})|$. Under Assumption 2.1 and 2.2, consider $\beta_1 < \sqrt{\beta_2}$, when $k$ is large such that: $k \geq 4$; $\beta_1^{(k-1)n} \leq \frac{\beta_1^n}{\sqrt{k-1}}$, we have the following results:*

$$\mathbb{E}\left[\sum_{l \text{ large}}(a_2)\right] := \mathbb{E}\left[\sum_{l \text{ large}} \delta_1 \sqrt{\frac{2\rho_3^2}{\beta_2^n}} \sum_{i=0}^{n-1} |\partial_l f_i(x_{k,0})|\right] \overset{(24)}{\leq} \mathcal{O}\left(\delta_1 \mathbb{E} |\partial_\alpha f(x_{k,0})|\right) + \mathcal{O}\left(\delta_1 \sqrt{D_0}\right),$$

$$\mathbb{E}\left[\sum_{l \text{ large}}(b_2)\right] := \mathbb{E}\left[\sum_{l \text{ large}} \delta_1 \sqrt{\frac{2\rho_3^2}{\beta_2^n}} \sum_{i=0}^{n-1} |m_{l,k,i} - \partial_l f_i(x_{k,0})|\right] \leq \mathcal{O}\left(\delta_1 \mathbb{E} |\partial_\alpha f(x_{k,0})|\right) + \mathcal{O}\left(\delta_1 \sqrt{D_0}\right) + \mathcal{O}\left(\frac{1}{\sqrt{k}}\right),$$

*where $\delta_1$ is defined in (17), $D_0$ is defined in Assumption 2.2.*

Detailed proof of Lemma G.5 is shown in Appendix G.8.

**Bounding** $\mathbb{E}\left\{\sum_{l\,\mathbf{large}}(a_1) + \sum_{l\,\mathbf{large}}(b_1) - \sum_{l\,\mathbf{large}}\{(a_2) + (b_2)\}\right\}$. With the help of Lemma G.4 and G.5, we can bound $\mathbb{E}\left\{\sum_{l\,\text{large}}(a_1) + \sum_{l\,\text{large}}(b_1) - \sum_{l\,\text{large}}\{(a_2) + (b_2)\}\right\}$. The intuition is as follows:

- $(a_1)$ is in the order of $|\partial_l f(x_{k,0})|$, multiplied by some positive constant;
- $\sum_{l\,\text{large}}\{(a_2) + (b_2)\}$ vanishes when $\beta_2 \to 1$.
- $\mathbb{E}\left\{\sum_{l\,\text{large}}(b_1)\right\}$ vanishes when $\beta_2 \to 1$ and $k$ goes to infinity.

Therefore, $\mathbb{E}\left\{\sum_{l\,\text{large}}(a_1) + \sum_{l\,\text{large}}(b_1) - \sum_{l\,\text{large}}\{(a_2) + (b_2)\}\right\}$ is still in the order of $|\partial_l f(x_{k,0})|$ (multiplied by a positive constant when $\beta_2$ is large). More formal results are shown in Lemma G.6.

**Lemma G.6.** *Under Assumption 2.1 and 2.2, when the hyperparameters satisfy: i) $\beta_1 < \sqrt{\beta_2} < 1$, ii) $\beta_2$ is large enough such that $A(\beta_2)$ is small enough to satisfy (34), where $A(\beta_2)$ is a non-negative constant that approaches 0 when $\beta_2$ approaches 1. More Specifically, $A(\beta_2)$ needs to satisfy ($\rho_1$, $\rho_2$ and $\rho_3$ are defined in Appendix F.1).*

$$
\begin{aligned}
A(\beta_2) \quad &:= \quad \left\{\frac{(1-\beta_2)4n\rho_2}{\beta_2^n} + \left(\frac{1}{\sqrt{\beta_2^n}} - 1\right)\right\}\left(\sqrt{\frac{2\rho_3^2}{\beta_2^n}}4n + \left(\sqrt{\frac{2\rho_3^2}{\beta_2^n}}\frac{3n}{\left(1 - \frac{(1-\beta_2)4n\rho_2}{\beta_2^n}\right)}\right)\frac{1}{(1-\beta_1^n)}\right)\sqrt{D_1}\rho_1 d \\
&\leq \quad \frac{1}{\sqrt{10D_1 d}}, \quad\quad\quad\quad\quad\quad\quad\quad\quad\quad\quad\quad\quad\quad\quad\quad\quad\quad\quad\quad\quad\quad (34)
\end{aligned}
$$

*Then, we have the following result when $k$ is large enough such that $\beta_1^{(k-1)n} \leq \frac{\beta_1^n}{\sqrt{k-1}}$ and $k \geq 4$:*

$$
\begin{aligned}
&\mathbb{E}\left\{\sum_{l\,large}(a_1) + \sum_{l\,large}(b_1) - \sum_{l\,large}\{(a_2) + (b_2)\}\right\} \\
:=\quad &\mathbb{E}\left[\sum_{l\,large}\frac{\partial_l f(x_{k,0})^2}{\sqrt{v_{l,k,0}}}\right] + \mathbb{E}\left[\sum_{l\,large}\frac{\partial_l f(x_{k,0})}{\sqrt{v_{l,k,0}}}\sum_{i=0}^{n-1}(m_{l,k,i} - \partial_l f_i(x_{k,0}))\right] \\
&-d\delta_1\sqrt{\frac{2\rho_3^2}{\beta_2^n}}\mathbb{E}\left[\sum_{i=0}^{n-1}|\partial_l f_i(x_{k,0})|\right] - d\delta_1\sqrt{\frac{2\rho_3^2}{\beta_2^n}}\mathbb{E}\left[\sum_{i=0}^{n-1}|m_{l,k,i} - \partial_l f_i(x_{k,0})|\right] \\
\geq\quad &\frac{1}{d\sqrt{10D_1 d}}\mathbb{E}\min\left\{\sqrt{\frac{2D_1 d}{D_0}}\|\nabla f(x_{k,0})\|_2^2, \|\nabla f(x_{k,0})\|_1\right\} \\
&-\mathcal{O}(\frac{1}{\sqrt{k}}) - \mathcal{O}(\sqrt{D_0}).
\end{aligned}
$$

Proof of Lemma G.6 can be seen in Appendix G.9.

*Remark* G.7. Condition (34) specifies the smallest threshold of $\beta_2$ to ensure the convergence. This condition can be translated into the threshold funtion $\gamma_1(n)$ mentioned in Theorem 3.1. As a rough estimate, Lemma G.6 requires $\beta_2 \geq \gamma_1(n) = 1 - \mathcal{O}\left((1-\beta_1^n)/(n^2\rho)\right)$, where $\rho = \rho_1\rho_2\rho_3$. As discussed in Appendix F.1, we have $0 \leq \rho_3 \leq \sqrt{n}\rho_1 \leq n$. When $\rho_1, \rho_2, \rho_3$ achieve their upper bound at the same time, we get the worst case bound $\beta_2 \geq \gamma_1(n) = 1 - \mathcal{O}\left((1-\beta_1^n)/n^{4.5}\right)$. However, $\rho$ is highly dependent on the problem instance $f(x)$ and training process. Our experiments in Appendix B shows that $\rho$ is often much smaller than its worst case bound, making the threshold of $\beta_2$ lower than it appears to be. Note that for adaptive gradient methods, we are not the first to provide the threshold on $\beta_2$ for convergence guarantee: a similar threshold on $\beta_2$ was firstly provided for RMSProp by (Shi et al., 2020). We prove that the threshold also exists for Adam, but with extra dependence on $\beta_1$.

*Remark* G.8. We emphasize that the constant term $\mathcal{O}(\sqrt{D_0})$ will vanish to 0 as $\beta_2$ goes to 1. This can be seen in the proof in Appendix G.9 (the definition of $F_4$) and Remark G.14.

Based on Lemma G.6, we can further rewrite (31) as follows.

$$
\mathbb{E}\left[\sum_{l=1}^{d}\sum_{i=0}^{n-1}\partial_l f(x_{k,0})\frac{m_{l,k,i}}{\sqrt{v_{l,k,i}}}\right] \overset{(31)}{\geq} \mathbb{E}\left\{\sum_{l\text{ large}}(a_1)+\sum_{l\text{ large}}(b_1)-\sum_{l\text{ large}}\{(a_2)+(b_2)\}\right\}-dF_1\frac{1}{\sqrt{k}},
$$

$$
\overset{\text{Lemma G.6}}{\geq} \frac{1}{d\sqrt{10D_1 d}}\mathbb{E}\min\left\{\sqrt{\frac{2D_1 d}{D_0}}\|\nabla f(x_{k,0})\|_2^2, \|\nabla f(x_{k,0})\|_1\right\}
$$

$$
-\mathcal{O}(\frac{1}{\sqrt{k}})-\mathcal{O}(\sqrt{D_0})-d\frac{F_1}{\sqrt{k}} \tag{35}
$$

The proof of Theorem 3.1 is concluded by plugging (35) into Descent Lemma (25) and then taking telescope some from $k = t_0$ to $k = T$. These steps are quite standard in non-convex optimization. We finish these calculation in Lemma G.9

**Lemma G.9.** *When inequality* (35) *holds, we have the following results based on Descent Lemma* (25)*:*

$$
\min_{k\in[1,T]}\mathbb{E}\left[\min\left\{\sqrt{\frac{2D_1 d}{D_0}}\|\nabla f(x_{k,0})\|_2^2, \|\nabla f(x_{k,0})\|_1\right\}\right]
$$

$$
=\mathcal{O}\left(\frac{\log T}{\sqrt{T}}\right)+\mathcal{O}(\sqrt{D_0}).
$$

Proof of Lemma G.9 can be seen in Appendix G.10. Now, the whole proof of Theorem 3.1 is completed.

## G.2 Proof of Lemma G.1

To prove Lemma G.1, we only prove (28). The proof for (27) can be analogized from the proof of (28). We discuss the following two cases:

**Case 1:** When $\partial_l f(x_{k,0})(m_{l,k,i}-\partial_l f_i(x_{k,0}))\leq 0$, we have

$$
\frac{\partial_l f(x_{k,0})(m_{l,k,i}-\partial_l f_i(x_{k,0}))}{\sqrt{v_{l,k,i}}} \overset{(a)}{\geq} \frac{\partial_l f(x_{k,0})(m_{l,k,i}-\partial_l f_i(x_{k,0}))}{\sqrt{v_{l,k,0}}}\frac{1}{\sqrt{\beta_2^i}},
$$

where $(a)$ is because of $\sqrt{v_{l,k,i}}\geq\sqrt{\beta_2^i}\sqrt{v_{l,k,0}}$.

**Case 2:** When $\partial_l f(x_{k,0})(m_{l,k,i}-\partial_l f_i(x_{k,0}))\geq 0$, we have

$$
\frac{\partial_l f(x_{k,0})(m_{l,k,i}-\partial_l f_i(x_{k,0}))}{\sqrt{v_{l,k,i}}} = \frac{\partial_l f(x_{k,0})(m_{l,k,i}-\partial_l f_i(x_{k,0}))}{\sqrt{v_{l,k,0}}}\frac{1}{\sqrt{(1+(v_{l,k,i}-v_{l,k,0})/v_{l,k,0})}}
$$

$$
\overset{(*)}{\geq} \frac{\partial_l f(x_{k,0})(m_{l,k,i}-\partial_l f_i(x_{k,0}))}{\sqrt{v_{l,k,0}}}\left(1-\frac{|v_{l,k,i}-v_{l,k,0}|}{2v_{l,k,0}}\right) \tag{36}
$$

where $(*)$ is due to $\frac{1}{\sqrt{1+x}}\geq 1-\frac{x}{2}$. We further have

$$
\begin{aligned}
|v_{l,k,i} - v_{l,k,0}| \quad &= \quad (1-\beta_2)(\partial_l f_{\tau_{k,i}}(x_{\tau_{k,i}})^2 + \beta_2 \partial_l f_{\tau_{k,i-1}}(x_{\tau_{k,i-1}})^2 + \cdots + \beta_2^{i-1} \partial_l f_{\tau_{k,1}}(x_{\tau_{k,1}})^2 \\
&\qquad + (\beta_2^i - 1)\partial_l f_{\tau_{k,0}}(x_{\tau_{k,0}})^2 + (\beta_2^{i+1} - 1)\partial_l f_{\tau_{k-1,n-1}}(x_{\tau_{k-1,n-1}})^2 + \cdots) \\
&\overset{\text{since } \beta_2^i - 1 < 0}{\leq} \quad (1-\beta_2)\sum_{j=0}^{i-1}\beta_2^j \partial_l f_{\tau_{k,i-j}}(x_{\tau_{k,i-j}})^2 \\
&\leq \quad (1-\beta_2)\sum_{j=0}^{i-1}\beta_2^j \left(\partial_l f_{\tau_{k,i-j}}(x_{\tau_{k,0}})^2 + 2(i-j)\triangle_{nk}|\partial_l f_{\tau_{k,i-j}}(x_{\tau_{k,0}})| + (i-j)^2 \triangle_{nk}^2 \right) \\
&\leq \quad (1-\beta_2)\sum_{j=0}^{i-1}\beta_2^j \left(\max_i |\partial_l f_i(x_{\tau_{k,0}})|^2 + 2(i-j)\triangle_{nk}\max_i |\partial_l f_i(x_{\tau_{k,0}})| + (i-j)^2 \triangle_{nk}^2 \right) \\
&\overset{(**)}{\leq} \quad (1-\beta_2)\sum_{j=0}^{i-1}\beta_2^j \left(\max_i |\partial_l f_i(x_{\tau_{k,0}})|^2 + 2\max_i |\partial_l f_i(x_{\tau_{k,0}})|^2 + \max_i |\partial_l f_i(x_{\tau_{k,0}})|^2 \right) \\
&\leq \quad (1-\beta_2)4n \max_i |\partial_l f_i(x_{\tau_{k,0}})|^2
\end{aligned}
$$

where $(**)$ is due to the condition in the Lemma $\max_i |\partial_l f_i(x_{\tau_{k,0}})| \geq Q_k \geq n\triangle_{nk}$. Plug the above result into (36) we have

$$
\begin{aligned}
\frac{\partial_l f(x_{k,0})(m_{l,k,i} - \partial_l f_i(x_{k,0}))}{\sqrt{v_{l,k,i}}} \quad &\geq \quad \frac{\partial_l f(x_{k,0})(m_{l,k,i} - \partial_l f_i(x_{k,0}))}{\sqrt{v_{l,k,0}}}\left(1 - \frac{(1-\beta_2)4n\max_i |\partial_l f_i(x_{\tau_{k,0}})|^2}{2v_{l,k,0}}\right) \\
&\geq \quad \frac{\partial_l f(x_{k,0})(m_{l,k,i} - \partial_l f_i(x_{k,0}))}{\sqrt{v_{l,k,0}}}\left(1 - \frac{(1-\beta_2)4n\rho_2 \sum_{i=0}^{n-1}|\partial_l f_i(x_{\tau_{k,0}})|^2/n}{2v_{l,k,0}}\right) \\
&\overset{(***)}{\geq} \quad \frac{\partial_l f(x_{k,0})(m_{l,k,i} - \partial_l f_i(x_{k,0}))}{\sqrt{v_{l,k,0}}}\left(1 - \frac{(1-\beta_2)4n\rho_2}{\beta_2^n}\right) \qquad (37)
\end{aligned}
$$

In the last step $(***)$, we use the following Lemma G.10, which is based on (Shi et al., 2020).

**Lemma G.10.** *Under Assumption 2.1, if the $l$-th component of $\nabla f_i(x_{k,0})$ satisfies* $\max_i |\partial_l f_i(x_{k,0})| \geq Q_k := \triangle_1 \frac{n\sqrt{n}}{\sqrt{k}}\frac{32\sqrt{2}}{(1-\beta_2)^n \beta_2^n}$, *we have*

$$
\frac{v_{l,k,0}}{\frac{1}{n}\sum_i \partial_l f_{\tau_{k,i}}(x_{k,0})^2} \geq \frac{\beta_2^n}{2},
$$

$$
\frac{v_{l,k,0}}{(\partial_l f(x_{k,0}))^2} \geq \frac{\beta_2^n}{2\rho_3^2}.
$$

*Proof.* The proof idea of Lemma G.10 is similar as Lemma F.1 in (Shi et al., 2020), but with the following differences:

1. The condition of Lemma F.1 in (Shi et al., 2020) both require the full batch gradient $|\partial_l f(x_{k,0})| \geq nQ_k$, which is different from our condition. Here, we choose the condition of Lemma G.10 because it meets the need of our decomposition strategy in (31) .

2. The constant $Q_k$ is different. The difference is due to the different update formula of RMSProp and general Adam.

Anyhow, Lemma G.10 can be easily proved following most of the steps in Lemma F.1 in (Shi et al., 2020). We omit the proof for brevity. $\square$

Combining the above **Case 1** and **Case 2**, we have

$$\sum \frac{\partial_l f\left(x_{k,0}\right)\left(m_{l,k,i} - \partial_l f_i(x_{k,0})\right)}{\sqrt{v_{l,k,i}}} \geq \sum_{i-} \frac{\partial_l f\left(x_{k,0}\right)\left(m_{l,k,i} - \partial_l f_i(x_{k,0})\right)}{\sqrt{v_{l,k,0}}} \frac{1}{\sqrt{\beta_2^i}}$$

$$+ \sum_{i+} \frac{\partial_l f\left(x_{k,0}\right)\left(m_{l,k,i} - \partial_l f_i(x_{k,0})\right)}{\sqrt{v_{l,k,0}}} \left(1 - \frac{(1 - \beta_2)4n\rho_2}{\beta_2^n}\right)$$

$$= \sum_{i=0}^{n-1} \frac{\partial_l f\left(x_{k,0}\right)\left(m_{l,k,i} - \partial_l f_i(x_{k,0})\right)}{\sqrt{v_{l,k,0}}}$$

$$+ \sum_{i-} \frac{\partial_l f\left(x_{k,0}\right)\left(m_{l,k,i} - \partial_l f_i(x_{k,0})\right)}{\sqrt{v_{l,k,0}}} \left(\frac{1}{\sqrt{\beta_2^i}} - 1\right)$$

$$+ \sum_{i+} \frac{\partial_l f\left(x_{k,0}\right)\left(m_{l,k,i} - \partial_l f_i(x_{k,0})\right)}{\sqrt{v_{l,k,0}}} \left(-\frac{(1 - \beta_2)4n\rho_2}{\beta_2^n}\right)$$

$$\geq \sum_{i=0}^{n-1} \frac{\partial_l f\left(x_{k,0}\right)\left(m_{l,k,i} - \partial_l f_i(x_{k,0})\right)}{\sqrt{v_{l,k,0}}}$$

$$- \delta_1 \left|\frac{\partial_l f\left(x_{k,0}\right)}{\sqrt{v_{l,k,0}}}\right| \sum_{i=0}^{n-1} \left|m_{l,k,i} - \partial_l f_i\left(x_{k,0}\right)\right|,$$

where $\delta_1 = \frac{(1-\beta_2)4n\rho_2}{\beta_2^n} + \left(\frac{1}{\sqrt{\beta_2^n}} - 1\right)$. Proof is completed.

### G.3 Proof of Lemma G.2

We first prove the result for $j = 1$. We discuss the following two cases.

**Case 1: when** $\frac{\partial_l f(x_{k,0})}{\sqrt{v_{l,k,0}}} \geq \frac{\partial_l f(x_{k-1,0})}{\sqrt{v_{l,k-1,0}}}$**:**    when $\partial_l f(x_{k,0}) \leq 0$, we have

$$\frac{\partial_l f(x_{k,0})}{\sqrt{v_{l,k,0}}} \overset{(37)}{\leq} \frac{\partial_l f(x_{k,0})}{\sqrt{v_{l,k-1,0}}} \left(1 - \frac{(1 - \beta_2)4n\rho_2}{\beta_2^n}\right).$$

When $\partial_l f(x_{k,0}) > 0$, we have

$$\frac{\partial_l f(x_{k,0})}{\sqrt{v_{l,k,0}}} \leq \frac{\partial_l f(x_{k,0})}{\sqrt{v_{l,k-1,0}}} \frac{1}{\sqrt{\beta_2^n}}$$

.

In conclusion, we have:

$$\frac{\partial_l f(x_{k,0})}{\sqrt{v_{l,k,0}}} \leq \max\left\{\frac{\partial_l f(x_{k,0})}{\sqrt{v_{l,k-1,0}}} \left(1 - \frac{(1 - \beta_2)4n\rho_2}{\beta_2^n}\right), \frac{\partial_l f(x_{k,0})}{\sqrt{v_{l,k-1,0}}} \frac{1}{\sqrt{\beta_2^n}}\right\}$$

$$\leq \frac{\partial_l f(x_{k,0})}{\sqrt{v_{l,k-1,0}}} + \max\left\{\frac{\partial_l f(x_{k,0})}{\sqrt{v_{l,k-1,0}}} \left(-\frac{(1 - \beta_2)4n\rho_2}{\beta_2^n}\right), \frac{\partial_l f(x_{k,0})}{\sqrt{v_{l,k-1,0}}} \left(\frac{1}{\sqrt{\beta_2^n}} - 1\right)\right\}$$

$$\leq \frac{\partial_l f(x_{k,0})}{\sqrt{v_{l,k-1,0}}} + \frac{|\partial_l f(x_{k,0})|}{\sqrt{v_{l,k-1,0}}} \left(\frac{1}{\sqrt{\beta_2^n}} - 1 + \frac{(1 - \beta_2)4n\rho_2}{\beta_2^n}\right)$$

$$\overset{(37)}{\leq} \frac{\partial_l f(x_{k,0})}{\sqrt{v_{l,k-1,0}}} + \frac{|\partial_l f(x_{k,0})|}{\sqrt{v_{l,k,0}}} \left(\frac{1}{\sqrt{\beta_2^n}} - 1 + \frac{(1 - \beta_2)4n\rho_2}{\beta_2^n}\right) \frac{1}{\left(1 - \frac{(1-\beta_2)4n\rho_2}{\beta_2^n}\right)}$$

$$\overset{\text{Lemma G.10}}{\leq} \quad \frac{\partial_l f(x_{k,0})}{\sqrt{v_{l,k-1,0}}} + \sqrt{\frac{2\rho_3^2}{\beta_2^n}} \left( \frac{1}{\sqrt{\beta_2^n}} - 1 + \frac{(1-\beta_2)4n\rho_2}{\beta_2^n} \right) \frac{1}{\left( 1 - \frac{(1-\beta_2)4n\rho_2}{\beta_2^n} \right)}$$

$$= \quad \frac{\partial_l f(x_{k,0})}{\sqrt{v_{l,k-1,0}}} + \sqrt{\frac{2\rho_3^2}{\beta_2^n}} \frac{\delta_1}{\left( 1 - \frac{(1-\beta_2)4n\rho_2}{\beta_2^n} \right)},$$

where $\delta_1 = \left( \frac{1}{\sqrt{\beta_2^n}} - 1 + \frac{(1-\beta_2)4n\rho_2}{\beta_2^n} \right)$ is a constant that goes to 0 when $\beta_2$ goes to 1. Therefore, we have

$$\frac{\partial_l f(x_{k,0})}{\sqrt{v_{l,k,0}}} - \frac{\partial_l f(x_{k-1,0})}{\sqrt{v_{l,k-1,0}}} \quad \leq \quad \frac{\partial_l f(x_{k,0}) - \partial_l f(x_{k-1,0})}{\sqrt{v_{l,k-1,0}}} + \sqrt{\frac{2\rho_3^2}{\beta_2^n}} \frac{1}{\left( 1 - \frac{(1-\beta_2)4n\rho_2}{\beta_2^n} \right)} \delta_1$$

$$\overset{\text{Lemma F.2}}{\leq} \quad \frac{n^2 \triangle_{n(k-1)}}{\sqrt{v_{l,k-1,0}}} + \sqrt{\frac{2\rho_3^2}{\beta_2^n}} \frac{1}{\left( 1 - \frac{(1-\beta_2)4n\rho_2}{\beta_2^n} \right)} \delta_1.$$

Note that

**Case 2: when $\frac{\partial_l f(x_{k,0})}{\sqrt{v_{l,k,0}}} \leq \frac{\partial_l f(x_{k-1,0})}{\sqrt{v_{l,k-1,0}}}$:** when $\partial_l f(x_{k,0}) \geq 0$, we have

$$\frac{\partial_l f(x_{k,0})}{\sqrt{v_{l,k,0}}} \overset{(37)}{\geq} \frac{\partial_l f(x_{k,0})}{\sqrt{v_{l,k-1,0}}} \left( 1 - \frac{(1-\beta_2)4n\rho_2}{\beta_2^n} \right).$$

When $\partial_l f(x_{k,0}) < 0$, we have

$$\frac{\partial_l f(x_{k,0})}{\sqrt{v_{l,k,0}}} \geq \frac{\partial_l f(x_{k,0})}{\sqrt{v_{l,k-1,0}}} \frac{1}{\sqrt{\beta_2^n}}$$

.

Following the same strategy as in Case 1, we can show that

$$\frac{\partial_l f(x_{k,0})}{\sqrt{v_{l,k,0}}} \quad \geq \quad \frac{\partial_l f(x_{k,0})}{\sqrt{v_{l,k-1,0}}} - \sqrt{\frac{2\rho_3^2}{\beta_2^n}} \frac{1}{\left( 1 - \frac{(1-\beta_2)4n\rho_2}{\beta_2^n} \right)} \delta_1,$$

which further implies

$$\frac{\partial_l f(x_{k,0})}{\sqrt{v_{l,k,0}}} - \frac{\partial_l f(x_{k-1,0})}{\sqrt{v_{l,k-1,0}}} \quad \geq \quad \frac{\partial_l f(x_{k,0}) - \partial_l f(x_{k-1,0})}{\sqrt{v_{l,k-1,0}}} - \sqrt{\frac{2\rho_3^2}{\beta_2^n}} \frac{1}{\left( 1 - \frac{(1-\beta_2)4n\rho_2}{\beta_2^n} \right)} \delta_1$$

$$\overset{\text{Lemma F.2}}{\geq} \quad -\frac{n^2 \triangle_{n(k-1)}}{\sqrt{v_{l,k-1,0}}} - \sqrt{\frac{2\rho_3^2}{\beta_2^n}} \frac{1}{\left( 1 - \frac{(1-\beta_2)4n\rho_2}{\beta_2^n} \right)} \delta_1.$$

Case 1 and Case 2 together, we have

$$\left| \frac{\partial_l f(x_{k,0})}{\sqrt{v_{l,k,0}}} - \frac{\partial_l f(x_{k-1,0})}{\sqrt{v_{l,k-1,0}}} \right| \leq \frac{n^2 \triangle_{n(k-1)}}{\sqrt{v_{l,k-1,0}}} + \sqrt{\frac{2\rho_3^2}{\beta_2^n}} \frac{1}{\left( 1 - \frac{(1-\beta_2)4n\rho_2}{\beta_2^n} \right)} \delta_1.$$

Now we consider the case when $j > 1$. Based on the above inequality, we have

$$\left| \frac{\partial_l f(x_{k,0})}{\sqrt{v_{l,k,0}}} - \frac{\partial_l f(x_{k-j,0})}{\sqrt{v_{l,k-j,0}}} \right| \leq \left( \frac{n^2 \triangle_{n(k-1)}}{\sqrt{v_{l,k-1,0}}} + \frac{n^2 \triangle_{n(k-2)}}{\sqrt{v_{l,k-2,0}}} + \cdots + \frac{n^2 \triangle_{n(k-j)}}{\sqrt{v_{l,k-j,0}}} \right) + j \sqrt{\frac{2\rho_3^2}{\beta_2^n}} \frac{1}{\left( 1 - \frac{(1-\beta_2)4n\rho_2}{\beta_2^n} \right)} \delta_1$$

$$\leq \left( \frac{1}{\sqrt{\beta_2^{jn}}} + \frac{1}{\sqrt{\beta_2^{(j-1)n}}} + \cdots + 1 \right) \frac{n^2 \triangle_{n(k-j)}}{\sqrt{v_{k-j,0}}} + j \sqrt{\frac{2\rho_3^2}{\beta_2^n}} \frac{1}{\left( 1 - \frac{(1-\beta_2)4n\rho_2}{\beta_2^n} \right)} \delta_1$$

$$\leq \frac{1}{1 - \frac{1}{\sqrt{\beta_2^n}}} \frac{n^2 \triangle_{n(k-j)}}{\sqrt{v_{l,k-j,0}}} + j \sqrt{\frac{2\rho_3^2}{\beta_2^n}} \frac{1}{\left( 1 - \frac{(1-\beta_2)4n\rho_2}{\beta_2^n} \right)} \delta_1.$$

The proof is completed.

## G.4  Proof of Lemma G.4

To prove Lemma G.4, we need to further decompose $\mathbb{E}[\sum_{l \text{ large}}(b_1)]$. First and foremost, we write $\sum_{i=0}^{n-1} m_{l,k,i}$ in an explicit form.

$$
\begin{aligned}
m_{l,k,i} =\ & (1-\beta_1)\{\partial_l f_{\tau_{k,i}}(x_{k,i}) + \cdots + \beta_1^i \partial_l f_{\tau_{k,0}}(x_{k,0}) \\
& + \beta_1^{i+1} f_{\tau_{k-1,n-1}}(x_{k-1,n-1}) + \cdots + \beta_1^{i+n} f_{\tau_{k-1,0}}(x_{k-1,0}) \\
& + \\
& \vdots \\
& + \\
& + \beta_1^{(k-2)n+i+1} \partial_l f_{\tau_{1,n-1}}(x_{1,n-1}) + \cdots + \beta_1^{(k-1)n+i} \partial_l f_{\tau_{1,0}}(x_{1,0})\} \\
& + \beta_1^{(k-1)n+i+1} \partial_l f(x_{1,0})
\end{aligned}
\tag{38}
$$

Since $\partial_l f(x_{1,0}) = \partial_l f(x_{1,0})(1-\beta_1)(1 + \beta_1 + \cdots + \beta_1^\infty)$ and $\partial_l f(x_{1,0}) = \sum_{i=0}^{n-1} \partial_l f_i(x_{1,0})$, we have

$$
\begin{aligned}
\beta_1^{(k-1)n+i+1} \partial_l f(x_{1,0}) =\ & (1-\beta_1)\{\beta_1^{(k-1)n+i+1} \partial_l f_0(x_{1,0}) + \cdots + \beta_1^{(k-1)n+i+1} \partial_l f_{n-1}(x_{1,0}) \\
& + \beta_1^{(k-1)n+i+2} \partial_l f_0(x_{1,0}) + \cdots + \beta_1^{(k-1)n+i+2} \partial_l f_{n-1}(x_{1,0}) \\
& + \\
& \vdots \\
& + \\
& + \beta_1^\infty \partial_l f_0(x_{1,0}) + \cdots + \beta_1^\infty \partial_l f_{n-1}(x_{1,0})\}
\end{aligned}
\tag{39}
$$

Plugging (39) into (38), we have

$$
\begin{aligned}
m_{l,k,i} =\ & (1-\beta_1)\{\partial_l f_{\tau_{k,i}}(x_{k,i}) + \cdots + \beta_1^i \partial_l f_{\tau_{k,0}}(x_{k,0}) \\
& + \beta_1^{i+1} f_{\tau_{k-1,n-1}}(x_{k-1,n-1}) + \cdots + \beta_1^{i+n} f_{\tau_{k-1,0}}(x_{k-1,0}) \\
& + \cdots \\
& + \beta_1^{(k-2)n+i+1} \partial_l f_{\tau_{1,n-1}}(x_{1,n-1}) + \cdots + \beta_1^{(k-1)n+i} \partial_l f_{\tau_{1,0}}(x_{1,0}) \\
& + \beta_1^{(k-1)n+i+1} \partial_l f_0(x_{1,0}) + \cdots + \beta_1^{(k-1)n+i+1} \partial_l f_{n-1}(x_{1,0}) \\
& + \beta_1^{(k-1)n+i+2} \partial_l f_0(x_{1,0}) + \cdots + \beta_1^{(k-1)n+i+2} \partial_l f_{n-1}(x_{1,0}) \\
& + \cdots \\
& + \beta_1^\infty \partial_l f_0(x_{1,0}) + \cdots + \beta_1^\infty \partial_l f_{n-1}(x_{1,0})\}
\end{aligned}
\tag{40}
$$

Using (40), we have (For each 'epoch', we suggest readers to read from bottom to the top.)

$$
\begin{aligned}
M_{l,k} := \sum_{i=0}^{n-1} m_{l,k,i} \;=\;& m_{l,k,n-1} + \cdots + m_{l,k,0} \\
=\;& (1-\beta_1)\{ \\
& \underbrace{\begin{aligned}
& \partial_l f_{\tau_{k,n-1}}(x_{k,n-1}) + \cdots + \beta_1^{n-1}\partial_l f_{\tau_{k,0}}(x_{k,0}) \\
& +\partial_l f_{\tau_{k,n-2}}(x_{k,n-2}) + \cdots + \beta_1^{n-2}\partial_l f_{\tau_{k,0}}(x_{k,0}) \\
& +\cdots \\
& +\partial_l f_{\tau_{k,0}}(x_{k,0})
\end{aligned}}_{k\text{-th epoch}} \\[2mm]
& \underbrace{\begin{aligned}
& +\beta_1^{n}\partial_l f_{\tau_{k-1,n-1}}(x_{k-1,n-1}) + \cdots + \beta_1^{n+n-1}\partial_l f_{\tau_{k-1,0}}(x_{k-1,0}) \\
& +\beta_1^{n-1}\partial_l f_{\tau_{k-1,n-1}}(x_{k-1,n-1}) + \cdots + \beta_1^{n+n-2}\partial_l f_{\tau_{k-1,0}}(x_{k-1,0}) \\
& +\cdots \\
& +\beta_1\partial_l f_{\tau_{k-1,n-1}}(x_{k-1,n-1}) + \cdots + \beta_1^{n}\partial_l f_{\tau_{k-1,0}}(x_{k-1,0})
\end{aligned}}_{k-1\text{-th epoch}} \\[2mm]
& + \\
& \vdots \\
& + \\[2mm]
& \underbrace{\begin{aligned}
& +\beta_1^{(k-1)n}\partial_l f_{\tau_{1,n-1}}(x_{1,n-1}) + \cdots + \beta_1^{(k-1)n+n-1}\partial_l f_{\tau_{1,0}}(x_{1,0}) \\
& +\beta_1^{(k-2)n+n-1}\partial_l f_{\tau_{1,n-1}}(x_{1,n-1}) + \cdots + \beta_1^{(k-1)n+n-2}\partial_l f_{\tau_{1,0}}(x_{1,0}) \\
& +\cdots \\
& +\beta_1^{(k-2)n+1}\partial_l f_{\tau_{1,n-1}}(x_{1,n-1}) + \cdots + \beta_1^{(k-1)n}\partial_l f_{\tau_{1,0}}(x_{1,0})
\end{aligned}}_{1\text{-th epoch}} \\[2mm]
& \underbrace{\begin{aligned}
& +\beta_1^{(k-1)n+n}\partial_l f_0(x_{1,0}) + \cdots + \beta_1^{(k-1)n+n}\partial_l f_{n-1}(x_{1,0}) \\
& +\beta_1^{(k-1)n+n-1}\partial_l f_0(x_{1,0}) + \cdots + \beta_1^{(k-1)n+n-1}\partial_l f_{n-1}(x_{1,0}) \\
& +\cdots \\
& +\beta_1^{(k-1)n+1}\partial_l f_0(x_{1,0}) + \cdots + \beta_1^{(k-1)n+1}\partial_l f_{n-1}(x_{1,0})+
\end{aligned}}_{0\text{-th epoch}} \\[2mm]
& \vdots \}
\end{aligned}
\tag{41}
$$

Note that in (41), the power of $\beta_1$ grows slower in the blue part (when $k=0$), such a transition will cause trouble in bounding the $\sum_{i=0}^{n-1}(m_{l,k,i} - \partial_l f_i(x_{k,0}))$. Therefore, we need to define an auxillary sequence $M'_{l,k}$ which does not involve such a phase transition. We define $M'_{l,k}$ as follows. (For each 'epoch', we suggest readers to read from bottom to the top.)

$$
\begin{aligned}
M'_{l,k} \;:=\;& (1-\beta_1)\{ \\
& \underbrace{\begin{aligned}
& \partial_l f_{\tau_{k,n-1}}(x_{k,n-1}) + \cdots + \beta_1^{n-1}\partial_l f_{\tau_{k,0}}(x_{k,0}) \\
& +\partial_l f_{\tau_{k,n-2}}(x_{k,n-2}) + \cdots + \beta_1^{n-2}\partial_l f_{\tau_{k,0}}(x_{k,0}) \\
& +\cdots \\
& +\partial_l f_{\tau_{k,0}}(x_{k,0})
\end{aligned}}_{k\text{-th epoch}}
\end{aligned}
$$

$$+\beta_1^n \partial_l f_{\tau_{k-1,n-1}}(x_{k-1,n-1}) + \cdots + \beta_1^{n+n-1} \partial_l f_{\tau_{k-1,0}}(x_{k-1,0})$$
$$+\beta_1^{n-1} \partial_l f_{\tau_{k-1,n-1}}(x_{k-1,n-1}) + \cdots + \beta_1^{n+n-2} \partial_l f_{\tau_{k-1,0}}(x_{k-1,0})$$
$$+\cdots$$
$$\underbrace{+\beta_1 \partial_l f_{\tau_{k-1,n-1}}(x_{k-1,n-1}) + \cdots + \beta_1^n \partial_l f_{\tau_{k-1,0}}(x_{k-1,0})}_{k-1\text{-th epoch}}$$
$$+$$
$$\vdots$$
$$+$$
$$+\beta_1^{(k-1)n} \partial_l f_{\tau_{1,n-1}}(x_{1,n-1}) + \cdots + \beta_1^{(k-1)n+n-1} \partial_l f_{\tau_{1,0}}(x_{1,0})$$
$$+\beta_1^{(k-2)n+n-1} \partial_l f_{\tau_{1,n-1}}(x_{1,n-1}) + \cdots + \beta_1^{(k-1)n+n-2} \partial_l f_{\tau_{1,0}}(x_{1,0})$$
$$+\cdots$$
$$\underbrace{+\beta_1^{(k-2)n+1} \partial_l f_{\tau_{1,n-1}}(x_{1,n-1}) + \cdots + \beta_1^{(k-1)n} \partial_l f_{\tau_{1,0}}(x_{1,0})}_{1\text{-th epoch}}$$
$$\underbrace{\begin{aligned}&+\beta_1^{(k-1)n+n} \partial_l f_0(x_{1,0}) + \cdots + \beta_1^{(k-1)n+n+n-1} \partial_l f_{n-1}(x_{1,0})\\ &+\beta_1^{(k-1)n+n-1} \partial_l f_0(x_{1,0}) + \cdots + \beta_1^{(k-1)n+2(n-1)} \partial_l f_{n-1}(x_{1,0})\\ &+\cdots\\ &+\beta_1^{(k-1)n+1} \partial_l f_0(x_{1,0}) + \cdots + \beta_1^{(k-1)n+n} \partial_l f_{n-1}(x_{1,0})+\end{aligned}}_{0\text{-th epoch}}$$
$$\begin{aligned}\vdots\\ \}\end{aligned} \tag{42}$$

Now we bound $\sum_{i=0}^{n-1}(m_{l,k,i} - \partial_l f_i(x_{k,0}))$ with the help of $M'_{l,k}$. Denoting $F_{l,k} := \sum_{i=0}^{n-1} \partial_l f_i(x_{k,0})$, we have:

$$\begin{aligned}
\sum_{i=0}^{n-1}(m_{l,k,i} - \partial_l f_i(x_{k,0})) &= \sum_{i=0}^{n-1} m_{l,k,i} - \sum_{i=0}^{n-1} \partial_l f_i(x_{k,0})\\
&:= M_{l,k} - F_{l,k}\\
&\geq -|M_{l,k} - M'_{l,k}| + M'_{l,k} - F_{l,k}
\end{aligned} \tag{43}$$

To proceed, we use the relation $1 = (1 - \beta_1)(1 + \beta_1 + \beta_1^2 + \cdots)$ to rewrite $F_{l,k}$.

$$F_{l,k} := \sum_{i=0}^{n-1} \partial_l f_i(x_{k,0}) = (1 - \beta_1)\{$$
$$\partial_l f_{n-1}(x_{k,0}) + \cdots + \beta_1^{n-1} \partial_l f_{n-1}(x_{k,0})$$
$$+\partial_l f_{n-2}(x_{k,0}) + \cdots + \beta_1^{n-1} \partial_l f_{n-2}(x_{k,0})$$
$$+\cdots$$
$$\underbrace{+\partial_l f_0(x_{k,0}) + \cdots + \beta_1^{n-1} \partial_l f_0(x_{k,0})}_{k\text{-th epoch}}$$
$$+$$
$$\vdots$$
$$+$$

$$+\beta_1^{(k-1)n}\partial_l f_{n-1}(x_{k,0}) + \cdots + \beta_1^{(k-1)n+n-1}\partial_l f_{n-1}(x_{k,0})$$
$$+\beta_1^{(k-1)n}\partial_l f_{n-2}(x_{k,0}) + \cdots + \beta_1^{(k-1)n+n-1}\partial_l f_{n-2}(x_{k,0})$$
$$+\cdots$$
$$\underbrace{+\beta_1^{(k-1)n}\partial_l f_0(x_{k,0}) + \cdots + \beta_1^{(k-1)n+n-1}\partial_l f_0(x_{k,0})}_{\text{1-th epoch}}$$

$$+\beta_1^{kn}\partial_l f_{n-1}(x_{k,0}) + \cdots + \beta_1^{kn+n-1}\partial_l f_{n-1}(x_{k,0})$$
$$+\beta_1^{kn}\partial_l f_{n-2}(x_{k,0}) + \cdots + \beta_1^{kn+n-1}\partial_l f_{n-2}(x_{k,0})$$
$$+\cdots$$
$$\underbrace{+\beta_1^{kn}\partial_l f_0(x_{k,0}) + \cdots + \beta_1^{kn+n-1}\partial_l f_0(x_{k,0})}_{\text{0-th epoch}}$$

$$\vdots \Big\} \tag{44}$$

For the sake of better presentation, we rewrite $F_{l,k}$ and $M'_{l,k}$ as follows:

$$F_{l,k} := (F_{l,k})_k + \cdots + (F_{l,k})_1 + (F_{l,k})_0 + (F_{l,k})_{-1} + \cdots,$$

$$M'_{l,k} := \left(M'_{l,k}\right)_k + \cdots + \left(M'_{l,k}\right)_1 + \left(M'_{l,k}\right)_0 + \left(M'_{l,k}\right)_{-1} + \cdots,$$

where $(F_{l,k})_j$ contains the summand of $F_{l,k}$ in the $j$-th epoch, $j = k, k-1, \cdots - \infty$. Similarly for $M'_{l,k}$. Now, we separate $M'_{l,k} - F_{l,k}$ into two parts.

$$M'_{l,k} - F_{l,k} = \left[\left(M'_{l,k}\right)_k - (F_{l,k})_k + \cdots \left(M'_{l,k}\right)_1 - (F_{l,k})_1\right]$$
$$+ \left[\left(M'_{l,k}\right)_0 - (F_{l,k})_0 + \cdots \left(M'_{l,k}\right)_{-\infty} - (F_{l,k})_{-\infty}\right]$$

Now, we rewrite $\mathbb{E}[\sum_{l \text{ large}}(b_1)]$:

$$\mathbb{E}\left[\sum_{l \text{ large}} \frac{\partial_l f(x_{k,0})}{\sqrt{v_{k,0}}} \sum_{i=0}^{n-1}(m_{l,k,i} - \partial_l f_i(x_{k,0}))\right]$$

$$= \mathbb{E}\left[\sum_{l \text{ large}} \frac{\partial_l f(x_{k,0})}{\sqrt{v_{k,0}}} (M_{l,k} - F_{l,k})\right]$$

$$\geq -\mathbb{E}\left[\sum_{l \text{ large}} \frac{|\partial_l f(x_{k,0})|}{\sqrt{v_{k,0}}}|M_{l,k} - M'_{l,k}|\right] + \mathbb{E}\left[\sum_{l \text{ large}} \frac{\partial_l f(x_{k,0})}{\sqrt{v_{k,0}}}(M'_{l,k} - F_{l,k})\right]$$

$$\geq \underbrace{-\mathbb{E}\left[\sum_{l \text{ large}} \frac{|\partial_l f(x_{k,0})|}{\sqrt{v_{k,0}}}|M_{l,k} - M'_{l,k}|\right]}_{(1)}$$

$$+ \mathbb{E}\left[\sum_{l \text{ large}} \frac{\partial_l f(x_{k,0})}{\sqrt{v_{k,0}}} \left( \left(M'_{l,k}\right)_k - (F_{l,k})_k + \cdots + \left(M'_{l,k}\right)_1 - (F_{l,k})_1 \right)\right]$$

$$\underbrace{\phantom{+ \mathbb{E}\left[\sum_{l \text{ large}} \frac{\partial_l f(x_{k,0})}{\sqrt{v_{k,0}}} \left( \left(M'_{l,k}\right)_k - (F_{l,k})_k + \cdots + \left(M'_{l,k}\right)_1 - (F_{l,k})_1 \right)\right]}}_{(2)}$$

$$- \mathbb{E}\left[\sum_{l \text{ large}} \frac{|\partial_l f(x_{k,0})|}{\sqrt{v_{k,0}}} \left| \left(M'_{l,k}\right)_0 - (F_{l,k})_0 + \cdots \left(M'_{l,k}\right)_{-\infty} - (F_{l,k})_{-\infty} \right|\right]$$

$$\underbrace{\phantom{- \mathbb{E}\left[\sum_{l \text{ large}} \frac{|\partial_l f(x_{k,0})|}{\sqrt{v_{k,0}}} \left| \left(M'_{l,k}\right)_0 - (F_{l,k})_0 + \cdots \left(M'_{l,k}\right)_{-\infty} - (F_{l,k})_{-\infty} \right|\right]}}_{(3)}$$

$$(45)$$

We bound (1), (2) and (3) respectively. We bound (1) and (3) in the following Lemma G.11 and Lemma G.12. Since the difference of $M_{l,k} - M'_{l,k}$ only occurs in the high order terms of the infinite sum sequence, (1) is expected to vanish as $k$ grows. Similarly for (3).

**Lemma G.11.** *When $k$ is large enough such that: $\beta_1^{(k-1)n} \leq \frac{\beta_1^n}{\sqrt{k-1}}$ and $k \geq 4$, then we have*

$$\mathbb{E}\left[\sum_{l \text{ large}} \frac{|\partial_l f(x_{k,0})|}{\sqrt{v_{k,0}}} \left|M_{l,k} - M'_{l,k}\right|\right] \leq G_4 \frac{1}{\sqrt{k}}, \tag{46}$$

*where $G_4 = d\sqrt{\frac{2\rho_3^2}{\beta_2^n}}\beta_1^n \sqrt{2}n(n-1)\sum_{i=0}^{n-1}\mathbb{E}\left(|\partial_\alpha f_i(x_{1,0})|\right)$.*

**Lemma G.12.** *When $k$ is large enough such that: $\beta_1^{(k-1)n} \leq \frac{\beta_1^n}{\sqrt{k-1}}$ and $k \geq 4$, then we have*

$$\mathbb{E}\left[\sum_{l \text{ large}} \frac{|\partial_l f(x_{k,0})|}{\sqrt{v_{k,0}}} \left| \left(M'_{l,k}\right)_0 - (F_{l,k})_0 + \cdots \left(M'_{l,k}\right)_{-\infty} - (F_{l,k})_{-\infty} \right|\right] \leq G_5 \frac{1}{\sqrt{k}}, \tag{47}$$

*where $G_5 = d\sqrt{\frac{2\rho_3^2}{\beta_2^n}}\left(\frac{\beta_1^{2n}2n^3\triangle_1\sqrt{2}}{\sqrt{n}}\frac{1-\beta_1}{(1-\beta_1^n)^2} + n\left(1-\beta_1^{n-1}\right)\sum_{i=0}^{n-1}\mathbb{E}|\partial_\alpha f_i(x_{1,0})|\frac{(1-\beta_1)\beta_1^n\sqrt{2}}{1-\beta_1^n}\right)$.*

We relegate the proof of Lemma G.11 and G.12 to Appendix G.5 and G.6.

Now we bound (2). This part involves the difficulties (i), (ii) and (iii) mentioned in Appendix G.1. We bound (2) in Lemma G.13.

**Lemma G.13.** *When $k$ is large enough such that: $\beta_1^{(k-1)n} \leq \frac{\beta_1^n}{\sqrt{k-1}}$ and $k \geq 2$, then we have*

$$\mathbb{E}\left[\sum_{l \text{ large}} \frac{\partial_l f(x_{k,0})}{\sqrt{v_{k,0}}} \left( \left(M'_{l,k}\right)_k - (F_{l,k})_k + \cdots + \left(M'_{l,k}\right)_1 - (F_{l,k})_1 \right)\right]$$

$$\geq -\frac{G_6}{\sqrt{k}} - G_7\delta_1\mathbb{E}\left(\sum_{l=1}^{d}\sum_{i=0}^{n-1}|\partial_l f_i(x_{k,0})|\right).$$

*The constant terms $G_6$ and $G_7$ are specified in Appendix G.7.*

Proof is shown in Appendix G.7.

Combining Lemma G.11, G.12 and G.13 together, we conclude the proof.

$$\mathbb{E}[\sum_{l \text{ large}} (b_1)] \geq -\frac{1}{\sqrt{k}}\left(G_4 + G_5 + G_6\right) - G_7\delta_1\mathbb{E}\left(\sum_{l=1}^{d}\sum_{i=0}^{n-1}|\partial_l f_i(x_{k,0})|\right).$$

## G.5 Proof of Lemma G.11

By definition of $M_{l,k}$ and $M'_{l,k}$ (see (41) and (42)), they only differ when $k \leq 0$. More specifically, We have (For each 'epoch', we suggest readers to read from bottom to the top.)

$$
\begin{aligned}
\frac{M_{l,k} - M'_{l,k}}{1 - \beta_1} =\ & \left(\beta_1^{(k-1)n+n} - \beta_1^{(k-1)n+n+1}\right)\partial_l f_1(x_{1,0}) + \cdots + \left(\beta_1^{(k-1)n+n} - \beta_1^{(k-1)n+n+n-1}\right)\partial_l f_{n-1}(x_{1,0}) \\
& + \cdots \\
& + \left(\beta_1^{(k-1)n+2} - \beta_1^{(k-1)n+3}\right)\partial_l f_1(x_{1,0}) + \cdots + \left(\beta_1^{(k-1)n+2} - \beta_1^{(k-1)n+n+1}\right)\partial_l f_{n-1}(x_{1,0}) \\
& + \underbrace{\left(\beta_1^{(k-1)n+1} - \beta_1^{(k-1)n+2}\right)\partial_l f_1(x_{1,0}) + \cdots + \left(\beta_1^{(k-1)n+1} - \beta_1^{(k-1)n+n}\right)\partial_l f_{n-1}(x_{1,0})}_{\text{0-th epoch}} \\
& + \left(\beta_1^{(k-1)n+n+1} - \beta_1^{kn+n}\right)\partial_l f_0(x_{1,0}) + \cdots + \left(\beta_1^{(k-1)n+n+1} - \beta_1^{kn+n+n-1}\right)\partial_l f_{n-1}(x_{1,0}) \\
& + \cdots \\
& + \left(\beta_1^{(k-1)n+3} - \beta_1^{kn+2}\right)\partial_l f_0(x_{1,0}) + \cdots + \left(\beta_1^{(k-1)n+3} - \beta_1^{kn+n+1}\right)\partial_l f_{n-1}(x_{1,0}) \\
& + \underbrace{\left(\beta_1^{(k-1)n+2} - \beta_1^{kn+1}\right)\partial_l f_0(x_{1,0}) + \cdots + \left(\beta_1^{(k-1)n+2} - \beta_1^{kn+n}\right)\partial_l f_{n-1}(x_{1,0})}_{\text{-1-th epoch}} \\
& + \left(\beta_1^{(k-1)n+n+2} - \beta_1^{(k+1)n+n}\right)\partial_l f_0(x_{1,0}) + \cdots + \left(\beta_1^{(k-1)n+n+2} - \beta_1^{(k+1)n+n+n-1}\right)\partial_l f_{n-1}(x_{1,0}) \\
& + \cdots \\
& + \left(\beta_1^{(k-1)n+4} - \beta_1^{(k+1)n+2}\right)\partial_l f_0(x_{1,0}) + \cdots + \left(\beta_1^{(k-1)n+4} - \beta_1^{(k+1)n+n+1}\right)\partial_l f_{n-1}(x_{1,0}) \\
& + \underbrace{\left(\beta_1^{(k-1)n+3} - \beta_1^{(k+1)n+1}\right)\partial_l f_0(x_{1,0}) + \cdots + \left(\beta_1^{(k-1)n+3} - \beta_1^{(k+1)n+n}\right)\partial_l f_{n-1}(x_{1,0})}_{\text{-2-th epoch}} \\
& + \cdots
\end{aligned}
\tag{48}
$$

We start with the 1st column in the '0-th epoch' (from the bottom to the top).

$$
\begin{aligned}
\text{The 1st column in the '0-th epoch'} =\ & \beta_1^{(k-1)n+1}(1-\beta_1)|\partial_l f_1(x_{1,0})| \\
& + \beta_1^{(k-1)n+2}(1-\beta_1)|\partial_l f_1(x_{1,0})| \\
& + \cdots \\
& + \beta_1^{(k-1)n+n}(1-\beta_1)|\partial_l f_1(x_{1,0})| \\
\leq\ & \beta_1^{(k-1)n+1} n(1-\beta_1)|\partial_l f_1(x_{1,0})|
\end{aligned}
\tag{49}
$$

Similarly, we can bound every column in the in the '0-th epoch', e.g. last ($n-1$-th) column can be bounded as follows.

$$
\begin{aligned}
\text{The last column in the '0-th epoch'} =\ & \beta_1^{(k-1)n+1}(1-\beta_1^{n-1})|\partial_l f_{n-1}(x_{1,0})| \\
& + \beta_1^{(k-1)n+2}(1-\beta_1^{n-1})|\partial_l f_{n-1}(x_{1,0})| \\
& + \cdots \\
& + \beta_1^{(k-1)n+n}(1-\beta_1^{n-1})|\partial_l f_{n-1}(x_{1,0})| \\
\leq\ & \beta_1^{(k-1)n+1} n(n-1)(1-\beta_1)|\partial_l f_{n-1}(x_{1,0})|
\end{aligned}
\tag{50}
$$

Summing up all the columns, we have

$$\text{The `0-th epoch'} \quad \leq \quad \beta_1^{(k-1)n+1} n(n-1)(1-\beta_1) \sum_{i=1}^{n-1} |\partial_l f_i(x_{1,0})|$$

$$\leq \quad \beta_1^{(k-1)n+1} n(n-1)(1-\beta_1) \sum_{i=0}^{n-1} |\partial_l f_i(x_{1,0})| \tag{51}$$

Using the same technique, it can be shown that

$$\text{The `-1-th epoch'} \quad \leq \quad \beta_1^{(k-1)n+2} 2n(n-1)(1-\beta_1) \sum_{i=0}^{n-1} |\partial_l f_i(x_{1,0})|, \tag{52}$$

$$\text{The `-2-th epoch'} \quad \leq \quad \beta_1^{(k-1)n+3} 3n(n-1)(1-\beta_1) \sum_{i=0}^{n-1} |\partial_l f_i(x_{1,0})| \tag{53}$$

$$\cdots$$

Plugging all these results in (48), we have

$$\frac{|M_{l,k} - M'_{l,k}|}{1-\beta_1} \quad \leq \quad \beta_1^{(k-1)n+1} n(n-1)(1-\beta_1) \sum_{i=0}^{n-1} |\partial_l f_i(x_{1,0})| \left(1 + 2\beta_1 + 3\beta_1^2 + \cdots\right)$$

$$= \quad \beta_1^{(k-1)n+1} n(n-1) \sum_{i=0}^{n-1} |\partial_l f_i(x_{1,0})| \frac{1}{1-\beta_1}.$$

$$\leq \quad \beta_1^{(k-1)n} n(n-1) \sum_{i=0}^{n-1} |\partial_l f_i(x_{1,0})| \frac{1}{1-\beta_1}.$$

That is to say, when $k$ is large enough such that $\beta_1^{(k-1)n} \leq \frac{\beta_1^n}{\sqrt{k-1}}$ we have

$$|M_{l,k} - M'_{l,k}| \quad \leq \quad \frac{\beta_1^n}{\sqrt{k-1}} n(n-1) \sum_{i=0}^{n-1} |\partial_l f_i(x_{1,0})| \tag{54}$$

$$\overset{\text{When } k \geq 2}{\leq} \quad \frac{\beta_1^n \sqrt{2}}{\sqrt{k}} n(n-1) \sum_{i=0}^{n-1} |\partial_l f_i(x_{1,0})|. \tag{55}$$

Combining with Lemma G.10, the proof is completed.

### G.6 Proof of Lemma G.12

We start with $\left(M'_{l,k}\right)_0 - (F_{l,k})_0$.

$$\frac{\left(M'_{l,k}\right)_0 - (F_{l,k})_0}{1-\beta_1} \quad = \quad \beta_1^{(k-1)n+n} \partial_l f_0(x_{1,0}) + \cdots + \beta_1^{(k-1)n+n+n-1} \partial_l f_{n-1}(x_{1,0})$$

$$+ \beta_1^{(k-1)n+n-1} \partial_l f_0(x_{1,0}) + \cdots + \beta_1^{(k-1)n+2(n-1)} \partial_l f_{n-1}(x_{1,0})$$

$$+ \cdots$$

$$\underbrace{+ \beta_1^{(k-1)n+1} \partial_l f_0(x_{1,0}) + \cdots + \beta_1^{(k-1)n+n} \partial_l f_{n-1}(x_{1,0})}_{\left(M'_{l,k}\right)_0}$$

$$- \Big\{ \beta_1^{kn} \partial_l f_{n-1}(x_{k,0}) + \cdots + \beta_1^{kn+n-1} \partial_l f_{n-1}(x_{k,0})$$
$$+ \beta_1^{kn} \partial_l f_{n-2}(x_{k,0}) + \cdots + \beta_1^{kn+n-1} \partial_l f_{n-2}(x_{k,0})$$
$$+ \cdots$$
$$+ \underbrace{\beta_1^{kn} \partial_l f_0(x_{k,0}) + \cdots + \beta_1^{kn+n-1} \partial_l f_0(x_{k,0})}_{(F_{l,k})_0} \Big\}$$
$$:= \sum_{i=0}^{n-1} \delta_i,$$

where

$$\delta_i = \sum_{j=1}^{n} \left( \beta_1^{(k-1)n+j} \partial_l f_i(x_{1,0}) - \beta_1^{kn+j-1} \partial_l f_i(x_{k,0}) \right)$$
$$= \underbrace{\sum_{j=1}^{n} \left( \beta_1^{(k-1)n+j} - \beta_1^{kn+j-1} \right) \partial_l f_i(x_{1,0})}_{\delta_i^a} + \underbrace{\sum_{j=1}^{n} \beta_1^{kn+j-1} \left( \partial_l f_i(x_{1,0}) - \partial_l f_i(x_{k,0}) \right)}_{\delta_i^b}.$$

We further have

$$\delta_i^a = \sum_{j=1}^{n} \beta_1^{(k-1)n+j} \left( 1 - \beta_1^{n-1} \right) \partial_l f_i(x_{1,0})$$
$$\leq n\beta_1^{(k-1)n+1} \left( 1 - \beta_1^{n-1} \right) |\partial_l f_i(x_{1,0})|$$

$$(56)$$

$$\delta_i^b \leq \sum_{j=1}^{n} \beta_1^{kn+j-1} \frac{\triangle_1(k-1)2n}{\sqrt{n(k-1)-1}}$$
$$\leq \beta_1^{kn} \frac{\triangle_1(k-1)2n^2}{\sqrt{n(k-1)-1}}.$$

$$(57)$$

Therefore,

$$\frac{\left| \left( M'_{l,k} \right)_0 - (F_{l,k})_0 \right|}{1 - \beta_1} = \left| \sum_{i=0}^{n-1} \delta_i \right|$$
$$\leq \sum_{i=0}^{n-1} \left( n\beta_1^{(k-1)n+1} \left( 1 - \beta_1^{n-1} \right) |\partial_l f_i(x_{1,0})| + \beta_1^{kn} \frac{\triangle_1(k-1)2n^2}{\sqrt{n(k-1)-1}} \right)$$
$$= \beta_1^{kn} \frac{\triangle_1(k-1)2n^3}{\sqrt{n(k-1)-1}} + n\beta_1^{(k-1)n+1} \left( 1 - \beta_1^{n-1} \right) \sum_{i=0}^{n-1} |\partial_l f_i(x_{1,0})| \quad (58)$$

Using the same calculation, we can get the following result:

$$\frac{\left| \left( M'_{l,k} \right)_{-1} - (F_{l,k})_{-1} \right|}{1 - \beta_1} \leq \beta_1^{(k+1)n} \frac{\triangle_1 k 2n^3}{\sqrt{n(k-1)-1}} + n\beta_1^{kn+1} \left( 1 - \beta_1^{n-1} \right) \sum_{i=0}^{n-1} |\partial_l f_i(x_{1,0})|.$$

Repeat this calculation, we have

$$\frac{1}{1-\beta_1}\left[\left(M'_{l,k}\right)_0 - (F_{l,k})_0 + \cdots \left(M'_{l,k}\right)_{-\infty} - (F_{l,k})_{-\infty}\right]$$

$$\leq \frac{\beta_1^{2n}2n^3\triangle_1}{\sqrt{n(k-1)-1}}\left(\beta_1^{(k-2)n}(k-1) + \beta_1^{(k-1)n}k + \cdots\right)$$

$$+ n\left(1-\beta_1^{n-1}\right)\sum_{i=0}^{n-1}|\partial_l f_i(x_{1,0})|\left(\beta_1^{(k-1)n+1} + \beta_1^{kn+1} + \cdots\right)$$

$$\leq \frac{\beta_1^{2n}2n^3\triangle_1}{\sqrt{n(k-1)-1}}\left(1 + \beta_1^n 2 + \cdots\right)$$

$$+ n\left(1-\beta_1^{n-1}\right)\sum_{i=0}^{n-1}|\partial_l f_i(x_{1,0})|\left(\beta_1^{(k-1)n+1} + \beta_1^{kn+1} + \cdots\right)$$

$$\overset{\text{Lemma F.1}}{\leq} \frac{\beta_1^{2n}2n^3\triangle_1}{\sqrt{n(k-1)-1}}\frac{1}{(1-\beta_1^n)^2}$$

$$+ n\left(1-\beta_1^{n-1}\right)\sum_{i=0}^{n-1}|\partial_l f_i(x_{1,0})|\frac{\beta_1^{(k-1)n}}{1-\beta_1^n}$$

When $\beta_1^{(k-1)n} \leq \frac{\beta_1^n}{\sqrt{k-1}}$, we further have:

$$\frac{1}{1-\beta_1}\left[\left(M'_{l,k}\right)_0 - (F_{l,k})_0 + \cdots \left(M'_{l,k}\right)_{-\infty} - (F_{l,k})_{-\infty}\right]$$

$$\leq \frac{\beta_1^{2n}2n^3\triangle_1}{\sqrt{n(k-1)-1}}\frac{1}{(1-\beta_1^n)^2}$$

$$+ n\left(1-\beta_1^{n-1}\right)\sum_{i=0}^{n-1}|\partial_l f_i(x_{1,0})|\frac{\beta_1^n}{1-\beta_1^n}\frac{1}{\sqrt{k-1}}$$

$$\overset{\frac{1}{\sqrt{k-1}}\leq\sqrt{\frac{2}{k}}}{\leq} \frac{\beta_1^{2n}2n^3\triangle_1}{\sqrt{n(k-1)-1}}\frac{1}{(1-\beta_1^n)^2}$$

$$+ n\left(1-\beta_1^{n-1}\right)\sum_{i=0}^{n-1}|\partial_l f_i(x_{1,0})|\frac{\beta_1^n}{1-\beta_1^n}\sqrt{\frac{2}{k}}$$

$$\overset{(*)}{\leq} \frac{\beta_1^{2n}2n^3\triangle_1\sqrt{2}}{\sqrt{nk}}\frac{1}{(1-\beta_1^n)^2}$$

$$+ n\left(1-\beta_1^{n-1}\right)\sum_{i=0}^{n-1}|\partial_l f_i(x_{1,0})|\frac{\beta_1^n}{1-\beta_1^n}\sqrt{\frac{2}{k}},$$

where $(*): \frac{1}{\sqrt{n(k-1)-1}} \leq \sqrt{\frac{2}{nk}}$ when $k \geq 4$. Therefore, we have

$$\left|\left(M'_{l,k}\right)_0 - (F_{l,k})_0 + \cdots \left(M'_{l,k}\right)_{-\infty} - (F_{l,k})_{-\infty}\right| \leq \frac{1}{\sqrt{k}}\tilde{G}_5$$

,

$\tilde{G}_5 = \frac{\beta_1^{2n}2n^3\triangle_1\sqrt{2}}{\sqrt{n}}\frac{1-\beta_1}{(1-\beta_1^n)^2} + n\left(1-\beta_1^{n-1}\right)\sum_{i=0}^{n-1}|\partial_l f_i(x_{1,0})|\frac{(1-\beta_1)\beta_1^n\sqrt{2}}{1-\beta_1^n}$.

Combining with Lemma G.10, the proof is completed with constant $G_5$ defined in Lemma G.12.

### G.7 Proof of Lemma G.13

In this section, we derive a lower bound for $\mathbb{E}\left[\sum_{l\text{ large}}\frac{\partial_l f(x_{k,0})}{\sqrt{v_{k,0}}}\left(\left(M'_{l,k}\right)_k - (F_{l,k})_k + \cdots + \left(M'_{l,k}\right)_1 - (F_{l,k})_1\right)\right]$.
We will use the ideas mentioned in Appendix G.1 (i.e., Step 1,2 and 3). We first rewrite "$l$ large" into indicator function as follows:

$$\mathbb{E}\left[\sum_{l \text{ large}} \frac{\partial_l f(x_{k,0})}{\sqrt{v_{k,0}}}\left(\left(M'_{l,k}\right)_k - (F_{l,k})_k + \cdots + \left(M'_{l,k}\right)_1 - (F_{l,k})_1\right)\right] = \mathbb{E}\left[\sum_{l=1}^{d} \mathbb{I}_k \frac{\partial_l f(x_{k,0})}{\sqrt{v_{k,0}}}\left(\left(M'_{l,k}\right)_k - (F_{l,k})_k + \cdots + \left(M'_{l,k}\right)_1 - (F_{l,k})_1\right)\right],$$

where $\mathbb{I}_k := \mathbb{I}\left(\max_i |\partial_l f_i(x_{k,0})| \geq Q_k := \triangle_1 \frac{n\sqrt{n}}{\sqrt{k}} \frac{32\sqrt{2}}{(1-\beta_2)^n \beta_2^n}\right)$. We also define $\mathbb{I}_{k-j} := \mathbb{I}\left(\max_i |\partial_l f_i(x_{k-j,0})| \geq \sum_{p=k-j}^{k} Q_p\right)$, it will be used later.

We take the conditional expectation over the history before $x_{k,0}$. We first focus on $\mathbb{E}_k\left[\sum_{l=1}^{d} \mathbb{I}_k \frac{\partial_l f(x_{k,0})}{\sqrt{v_{k,0}}}\left(\left(M'_{l,k}\right)_k - (F_{l,k})_k\right)\right]$ and we delegate the history part for later analysis.

For each possible $\left(M'_{l,k}\right)_k - (F_{l,k})_k$, we first convert all $x_{k,i}$ into $x_{k,0}$ using Lemma F.2. Since $\left(M'_{l,k}\right)_k$ contains $(n-1) + \cdots + 1 = \frac{n(n-1)}{2}$ terms of $x_{k,i}$ (with $i \neq 0$), we have

$$
\begin{aligned}
\left(M'_{l,k}\right)_k \overset{\text{Lemma F.2}}{\geq} \quad & -\frac{(1-\beta_1)n^2(n-1)\triangle_{nk}}{2} + (1-\beta_1)\{ \\
& \partial_l f_{\tau_{k,n-1}}(x_{k,0}) + \cdots + \beta_1^{n-1}\partial_l f_{\tau_{k,0}}(x_{k,0}) \\
& + \partial_l f_{\tau_{k,n-2}}(x_{k,0}) + \cdots + \beta_1^{n-2}\partial_l f_{\tau_{k,0}}(x_{k,0}) \\
& + \cdots \\
& \underbrace{+ \partial_l f_{\tau_{k,0}}(x_{k,0})}_{k\text{-th epoch}} \quad\quad\quad\quad\quad \} \\
:= \quad & -\frac{(1-\beta_1)n^2(n-1)\triangle_{nk}}{2} + \mathscr{M}_{l,k}.
\end{aligned}
$$

Using the same strategy for analyzing the color-ball toy example, we have

$$
\begin{aligned}
\frac{\mathbb{E}_k\left[\mathscr{M}_{l,k} - (F_{l,k})_k\right]}{1 - \beta_1} &= \frac{1}{n!}\left(-(n-1)!\beta_1 - 2(n-1)!\beta_1^2 - \cdots - (n-1)(n-1)!\beta_1^{n-1}\right)\sum_{i=0}^{n-1}\partial_l f_i(x_{k,0}) \\
&= \left(-\frac{1}{n}\beta_1 - \frac{2}{n}\beta_1^2 - \cdots - \frac{n-1}{n}\beta_1^{n-1}\right)\sum_{i=0}^{n-1}\partial_l f_i(x_{k,0})
\end{aligned}
$$

$$
\begin{aligned}
\mathbb{E}_k\left[\sum_{l=1}^{d} \mathbb{I}_k \frac{\partial_l f(x_{k,0})}{\sqrt{v_{k,0}}}\left(\left(M'_{l,k}\right)_k - (F_{l,k})_k\right)\right] &\geq -d\sqrt{\frac{2\rho_3^2}{\beta_2^n}}\frac{(1-\beta_1)n^2(n-1)\triangle_{nk}}{2} + \mathbb{E}_k\left[\sum_{l=1}^{d} \mathbb{I}_k \frac{\partial_l f(x_{k,0})}{\sqrt{v_{k,0}}}\left(\mathscr{M}_{l,k} - (F_{l,k})_k\right)\right] \\
&= -d\sqrt{\frac{2\rho_3^2}{\beta_2^n}}\frac{(1-\beta_1)n^2(n-1)\triangle_{nk}}{2} \\
&\quad + \sum_{l=1}^{d} \mathbb{I}_k(1-\beta_1)\left(-\frac{1}{n}\beta_1 - \frac{2}{n}\beta_1^2 - \cdots - \frac{n-1}{n}\beta_1^{n-1}\right)\sum_{i=0}^{n-1}\partial_l f_i(x_{k,0}) \\
&:= -d\sqrt{\frac{2\rho_3^2}{\beta_2^n}}\frac{(1-\beta_1)n^2(n-1)\triangle_{nk}}{2} + \sum_{l=1}^{d} \mathbb{I}_k \frac{\partial_l f(x_{k,0})}{\sqrt{v_{k,0}}}(1-\beta_1)J_1\sum_{i=0}^{n-1}\partial_l f_i(x_{k,0})
\end{aligned}
$$

We denote $J_1 := \left(-\frac{1}{n}\beta_1 - \frac{2}{n}\beta_1^2 - \cdots - \frac{n-1}{n}\beta_1^{n-1}\right)$, it will be used repeatedly in the following derivation.

Now we move one step further exert $\mathbb{E}_{k-1}(\cdot)$. In particular, we need to calculate

$$
\mathbb{E}_{k-1}\left\{\mathbb{E}_k\left[\sum_{l=1}^{d}\mathbb{I}_k\frac{\partial_l f(x_{k,0})}{\sqrt{v_{k,0}}}\left((M'_{l,k})_k-(F_{l,k})_k\right)+\sum_{l=1}^{d}\mathbb{I}_k\frac{\partial_l f(x_{k,0})}{\sqrt{v_{k,0}}}\left((M'_{l,k})_{k-1}-(F_{l,k})_{k-1}\right)\right]\right\}
$$

$$
\geq \quad \mathbb{E}_{k-1}\left\{\underbrace{\sum_{l=1}^{d}\mathbb{I}_k\frac{\partial_l f(x_{k,0})}{\sqrt{v_{k,0}}}(1-\beta_1)J_1\sum_{i=0}^{n-1}\partial_l f_i(x_{k,0})}_{(a)}+\underbrace{\sum_{l=1}^{d}\mathbb{I}_k\frac{\partial_l f(x_{k,0})}{\sqrt{v_{k,0}}}\left((M'_{l,k})_{k-1}-(F_{l,k})_{k-1}\right)}_{(b)}\right\}
$$

$$
-d\sqrt{\frac{2\rho_3^2}{\beta_2^n}}\frac{(1-\beta_1)n^2(n-1)\triangle_{nk}}{2}. \tag{59}
$$

The blue term is the residue from the $k$-th epoch. The red term is the main component in the $(k-1)$-th epoch. Before calculating $\mathbb{E}_{k-1}(\cdot)$, we need to convert all the variable $x$ into $x_{k-1,0}$ using Lipschitz property. First of all, we work on $(a)$.

$$
(a) \quad = \quad \sum_{l=1}^{d}\mathbb{I}_k\frac{\partial_l f(x_{k,0})}{\sqrt{v_{k,0}}}(1-\beta_1)J_1\sum_{i=0}^{n-1}\partial_l f_i(x_{k,0})
$$

$$
\overset{\text{Lemma G.3}}{=} \quad \sum_{l=1}^{d}\mathbb{I}_{k,k-1}\frac{\partial_l f(x_{k,0})}{\sqrt{v_{k,0}}}(1-\beta_1)J_1\sum_{i=0}^{n-1}\partial_l f_i(x_{k,0})
$$

$$
+\sum_{l=1}^{d}\tilde{\mathbb{I}}_{k,k-1}\frac{\partial_l f(x_{k,0})}{\sqrt{v_{k,0}}}(1-\beta_1)J_1\sum_{i=0}^{n-1}\partial_l f_i(x_{k,0})
$$

$$
\overset{\text{Lemma G.10}}{\geq} \quad \sum_{l=1}^{d}\mathbb{I}_{k,k-1}\frac{\partial_l f(x_{k,0})}{\sqrt{v_{k,0}}}(1-\beta_1)J_1\sum_{i=0}^{n-1}\partial_l f_i(x_{k,0})-\sum_{l=1}^{d}\tilde{\mathbb{I}}_{k,k-1}\sqrt{\frac{2\rho_3^2}{\beta_2^n}}(1-\beta_1)|J_1||\sum_{i=0}^{n-1}\partial_l f_i(x_{k,0})|
$$

$$
\geq \quad \sum_{l=1}^{d}\mathbb{I}_{k,k-1}\frac{\partial_l f(x_{k,0})}{\sqrt{v_{k,0}}}(1-\beta_1)J_1\sum_{i=0}^{n-1}\partial_l f_i(x_{k,0})-\sum_{l=1}^{d}\tilde{\mathbb{I}}_{k,k-1}\sqrt{\frac{2\rho_3^2}{\beta_2^n}}(1-\beta_1)|J_1|\left(\sum_{i=0}^{n-1}|\partial_l f_i(x_{k-1,0})|+n^2\triangle_{n(k-1)}\right)
$$

$$
\geq \quad \sum_{l=1}^{d}\mathbb{I}_{k,k-1}\frac{\partial_l f(x_{k,0})}{\sqrt{v_{k,0}}}(1-\beta_1)J_1\sum_{i=0}^{n-1}\partial_l f_i(x_{k,0})
$$

$$
-\sum_{l=1}^{d}\tilde{\mathbb{I}}_{k,k-1}\sqrt{\frac{2\rho_3^2}{\beta_2^n}}(1-\beta_1)|J_1|\sum_{i=0}^{n-1}|\partial_l f_i(x_{k-1,0})|-d\sqrt{\frac{2\rho_3^2}{\beta_2^n}}(1-\beta_1)|J_1|n^2\triangle_{n(k-1)}
$$

$$
\overset{\text{Lemma G.10 and F.2}}{\geq} \quad \sum_{l=1}^{d}\mathbb{I}_{k,k-1}\frac{\partial_l f(x_{k,0})}{\sqrt{v_{k,0}}}(1-\beta_1)J_1\sum_{i=0}^{n-1}\partial_l f_i(x_{k-1,0})
$$

$$
-(1-\beta_1)d\sqrt{\frac{2\rho_3^2}{\beta_2^n}}|J_1|n^2\triangle_{n(k-1)}-(1-\beta_1)d\sqrt{\frac{2\rho_3^2}{\beta_2^n}}|J_1|n(Q_k+Q_{k-1})-(1-\beta_1)d\sqrt{\frac{2\rho_3^2}{\beta_2^n}}|J_1|n^2\triangle_{n(k-1)}
$$

$$
\geq \quad \sum_{l=1}^{d}\mathbb{I}_{k,k-1}\frac{\partial_l f(x_{k-1,0})}{\sqrt{v_{k-1,0}}}(1-\beta_1)J_1\sum_{i=0}^{n-1}\partial_l f_i(x_{k-1,0})
$$

$$
-\sum_{l=1}^{d}\mathbb{I}_{k,k-1}\left|\frac{\partial_l f(x_{k-1,0})}{\sqrt{v_{k-1,0}}}-\frac{\partial_l f(x_{k,0})}{\sqrt{v_{k,0}}}\right|(1-\beta_1)|J_1|\left|\sum_{i=0}^{n-1}\partial_l f_i(x_{k-1,0})\right|
$$

$$
-2(1-\beta_1)d\sqrt{\frac{2\rho_3^2}{\beta_2^n}}|J_1|n^2\triangle_{n(k-1)}-(1-\beta_1)d\sqrt{\frac{2\rho_3^2}{\beta_2^n}}|J_1|n(Q_k+Q_{k-1})
$$

$$\overset{\text{Lemma G.2}}{\geq} \quad \sum_{l=1}^{d} \mathbb{I}_{k,k-1} \frac{\partial_l f(x_{k-1,0})}{\sqrt{v_{k-1,0}}} (1-\beta_1) J_1 \sum_{i=0}^{n-1} \partial_l f_i(x_{k-1,0})$$

$$- \sum_{l=1}^{d} \mathbb{I}_{k,k-1}(1-\beta_1)|J_1| \left| \sum_{i=0}^{n-1} \partial_l f_i(x_{k-1,0}) \right| \left( \frac{1}{1-\frac{1}{\sqrt{\beta_2^n}}} \frac{n^2 \triangle_{n(k-1)}}{\sqrt{v_{k-1,0}}} + \sqrt{\frac{2\rho_3^2}{\beta_2^n}} \frac{1}{\left(1-\frac{(1-\beta_2)4n\rho_2}{\beta_2^n}\right)} \delta_1 \right)$$

$$-2(1-\beta_1)d\sqrt{\frac{2\rho_3^2}{\beta_2^n}}|J_1|n^2\triangle_{n(k-1)} - (1-\beta_1)d\sqrt{\frac{2\rho_3^2}{\beta_2^n}}|J_1|n(Q_k+Q_{k-1})$$

where $\mathbb{I}_{k,k-j} := \mathbb{I}\left(\max_i |\partial_l f_i(x_{k,0})| \geq Q_k \text{ and } \max_i |\partial_l f_i(x_{k-j,0})| \geq \sum_{p=k-j}^{k} Q_p\right)$ and $\tilde{\mathbb{I}}_{k,k-j} := \mathbb{I}\left(\max_i |\partial_l f_i(x_{k,0})| \geq Q_k \text{ and } \max_i |\partial_l f_i(x_{k-j,0})| \leq \sum_{p=k-j}^{k} Q_p\right)$. By Lemma G.3, we know $\mathbb{I}_{k,k-1} = \mathbb{I}_{k-1}$, so we have:

$$(a) \quad \overset{\text{Lemma G.10}}{\geq} \quad \sum_{l=1}^{d} \mathbb{I}_{k-1} \frac{\partial_l f(x_{k-1,0})}{\sqrt{v_{k-1,0}}} (1-\beta_1) J_1 \sum_{i=0}^{n-1} \partial_l f_i(x_{k-1,0})$$

$$-d(1-\beta_1)|J_1| \frac{1}{1-\frac{1}{\sqrt{\beta_2^n}}} n^2 \triangle_{n(k-1)} \sqrt{\frac{2}{n\beta_2^n}}$$

$$- \sum_{l=1}^{d}(1-\beta_1)|J_1| \left| \sum_{i=0}^{n-1} \partial_l f_i(x_{k-1,0}) \right| \left( \sqrt{\frac{2\rho_3^2}{\beta_2^n}} \frac{1}{\left(1-\frac{(1-\beta_2)4n\rho_2}{\beta_2^n}\right)} \delta_1 \right)$$

$$-2(1-\beta_1)d\sqrt{\frac{2\rho_3^2}{\beta_2^n}}|J_1|n^2\triangle_{n(k-1)} - (1-\beta_1)d\sqrt{\frac{2\rho_3^2}{\beta_2^n}}|J_1|n(Q_k+Q_{k-1})$$

$$\overset{\text{Lemma F.2}}{\geq} \quad \sum_{l=1}^{d} \mathbb{I}_{k-1} \frac{\partial_l f(x_{k-1,0})}{\sqrt{v_{k-1,0}}} (1-\beta_1) J_1 \sum_{i=0}^{n-1} \partial_l f_i(x_{k-1,0})$$

$$-d(1-\beta_1)|J_1| \frac{1}{1-\frac{1}{\sqrt{\beta_2^n}}} n^2 \triangle_{n(k-1)} \sqrt{\frac{2}{n\beta_2^n}}$$

$$- \sum_{l=1}^{d}(1-\beta_1)|J_1| \left( \left| \sum_{i=0}^{n-1} \partial_l f_i(x_{k,0}) \right| + n^2 \triangle_{n(k-1)} \right) \left( \sqrt{\frac{2\rho_3^2}{\beta_2^n}} \frac{1}{\left(1-\frac{(1-\beta_2)4n\rho_2}{\beta_2^n}\right)} \delta_1 \right)$$

$$-2(1-\beta_1)d\sqrt{\frac{2\rho_3^2}{\beta_2^n}}|J_1|n^2\triangle_{n(k-1)} - (1-\beta_1)d\sqrt{\frac{2\rho_3^2}{\beta_2^n}}|J_1|n(Q_k+Q_{k-1}).$$

The blue term will be handled using color-ball method when taking conditional expectation $\mathbb{E}_{k-1}(\cdot)$. Now we derive a lower bound for (b). Similarly as before, we rewrite $\left(M'_{l,k}\right)_{k-1}$ and $(F_{l,k})_{k-1}$ as follows.

$$\left(M'_{l,k}\right)_{k-1} \overset{\text{Lemma F.2}}{\geq} \mathscr{M}_{l,k-1} - (1-\beta_1)\beta_1 n^3 \triangle_{n(k-1)};$$

$$\left(F_{l,k}\right)_{k-1} \overset{\text{Lemma F.2}}{\geq} \left(F_{l,k-1}\right)_{k-1} - (1-\beta_1)\beta_1^n n^3 \triangle_{n(k-1)};$$

where

$$\mathcal{M}_{l,k-1} := (1-\beta_1)\{$$
$$+\beta_1^n \partial_l f_{\tau_{k-1,n-1}}(x_{k-1,0}) + \cdots + \beta_1^{n+n-1} \partial_l f_{\tau_{k-1,0}}(x_{k-1,0})$$
$$+\beta_1^{n-1} \partial_l f_{\tau_{k-1,n-1}}(x_{k-1,0}) + \cdots + \beta_1^{n+n-2} \partial_l f_{\tau_{k-1,0}}(x_{k-1,0})$$
$$+\cdots$$
$$\underbrace{+\beta_1 \partial_l f_{\tau_{k-1,n-1}}(x_{k-1,0}) + \cdots + \beta_1^n \partial_l f_{\tau_{k-1,0}}(x_{k-1,0})\}}_{k-1\text{-th epoch}}$$

$$(F_{l,k-1})_{k-1} := \sum_{i=0}^{n-1} \partial_l f_i(x_{k-1,0}) = (1-\beta_1)\beta_1^n\{$$
$$\partial_l f_{n-1}(x_{k-1,0}) + \cdots + \beta_1^{n-1} \partial_l f_{n-1}(x_{k-1,0})$$
$$+\partial_l f_{n-2}(x_{k-1,0}) + \cdots + \beta_1^{n-1} \partial_l f_{n-2}(x_{k-1,0})$$
$$+\cdots$$
$$\underbrace{+\partial_l f_0(x_{k-1,0}) + \cdots + \beta_1^{n-1} \partial_l f_0(x_{k-1,0})\}}_{(k-1)\text{-th epoch}}$$

We now calculate $(b)$. Using the same idea as in $(a)$, we have

$$(b) = \sum_{l=1}^{d} \mathbb{I}_k \frac{\partial_l f(x_{k,0})}{\sqrt{v_{k,0}}} \left( (M'_{l,k})_{k-1} - (F_{l,k})_{k-1} \right)$$

$$\overset{\text{Lemma G.10 and F.2}}{\geq} \sum_{l=1}^{d} \mathbb{I}_k \frac{\partial_l f(x_{k,0})}{\sqrt{v_{k,0}}} \left( \mathcal{M}_{l,k-1} - (F_{l,k-1})_{k-1} \right) - d\sqrt{\frac{2\rho_3^2}{\beta_2^n}}(1-\beta_1)(\beta_1+\beta_1^n)n^3 \triangle_{n(k-1)}$$

$$\overset{\text{Lemma G.3}}{=} \sum_{l=1}^{d} \mathbb{I}_{k,k-1} \frac{\partial_l f(x_{k,0})}{\sqrt{v_{k,0}}} \left( \mathcal{M}_{l,k-1} - (F_{l,k-1})_{k-1} \right) + \sum_{l=1}^{d} \tilde{\mathbb{I}}_{k,k-1} \frac{\partial_l f(x_{k,0})}{\sqrt{v_{k,0}}} \left( \mathcal{M}_{l,k-1} - (F_{l,k-1})_{k-1} \right)$$
$$-d\sqrt{\frac{2\rho_3^2}{\beta_2^n}}(1-\beta_1)(\beta_1+\beta_1^n)n^3 \triangle_{n(k-1)}$$

$$\overset{\text{Lemma G.10}}{\geq} \sum_{l=1}^{d} \mathbb{I}_{k,k-1} \frac{\partial_l f(x_{k,0})}{\sqrt{v_{k,0}}} \left( \mathcal{M}_{l,k-1} - (F_{l,k-1})_{k-1} \right) - \sum_{l=1}^{d} \tilde{\mathbb{I}}_{k,k-1} \sqrt{\frac{2\rho_3^2}{\beta_2^n}} \left| \mathcal{M}_{l,k-1} - (F_{l,k-1})_{k-1} \right|$$
$$-d\sqrt{\frac{2\rho_3^2}{\beta_2^n}}(1-\beta_1)(\beta_1+\beta_1^n)n^3 \triangle_{n(k-1)}$$

$$\overset{\text{Def of } \tilde{\mathbb{I}}_{k,k-1}}{\geq} \sum_{l=1}^{d} \mathbb{I}_{k,k-1} \frac{\partial_l f(x_{k,0})}{\sqrt{v_{k,0}}} \left( \mathcal{M}_{l,k-1} - (F_{l,k-1})_{k-1} \right) - d\sqrt{\frac{2\rho_3^2}{\beta_2^n}}(1-\beta_1)(\beta_1+\beta_1^n)n^2 \left( Q_k + Q_{k-1} \right)$$
$$-d\sqrt{\frac{2\rho_3^2}{\beta_2^n}}(1-\beta_1)(\beta_1+\beta_1^n)n^3 \triangle_{n(k-1)}$$

$$\overset{\text{Lemma G.2}}{\geq} \sum_{l=1}^{d} \mathbb{I}_{k,k-1} \frac{\partial_l f(x_{k-1,0})}{\sqrt{v_{k-1,0}}} \left( \mathcal{M}_{l,k-1} - (F_{l,k-1})_{k-1} \right)$$
$$-\sum_{l=1}^{d} \left( \frac{1}{1-\frac{1}{\sqrt{\beta_2^n}}} \frac{n^2 \triangle_{n(k-1)}}{\sqrt{v_{k-1,0}}} + \sqrt{\frac{2\rho_3^2}{\beta_2^n}} \frac{1}{\left(1-\frac{(1-\beta_2)4n\rho_2}{\beta_2^n}\right)}\delta_1 \right) \left| \mathcal{M}_{l,k-1} - (F_{l,k-1})_{k-1} \right|$$
$$-d\sqrt{\frac{2\rho_3^2}{\beta_2^n}}(1-\beta_1)(\beta_1+\beta_1^n)n^2 \left( Q_k + Q_{k-1} \right) - d\sqrt{\frac{2\rho_3^2}{\beta_2^n}}(1-\beta_1)(\beta_1+\beta_1^n)n^3 \triangle_{n(k-1)}$$

To proceed, we derive an upper bound for $\left| \mathscr{M}_{l,k-1} - (F_{l,k-1})_{k-1} \right|$.

$$
\begin{aligned}
\left| \mathscr{M}_{l,k-1} - (F_{l,k-1})_{k-1} \right| &\leq |\mathscr{M}_{l,k-1}| + \left| (F_{l,k-1})_{k-1} \right| \\
&\leq (1-\beta_1)\beta_1 n \sum_{i=0}^{n-1} |\partial_l f_i(x_{k-1,0})| + (1-\beta_1)\beta_1^n n \sum_{i=0}^{n-1} |\partial_l f_i(x_{k-1,0})|
\end{aligned}
$$

Therefore, we have:

$$
\begin{aligned}
(b) \quad \geq \quad & \sum_{l=1}^{d} \mathbb{I}_{k,k-1} \frac{\partial_l f(x_{k-1,0})}{\sqrt{v_{k-1,0}}} \left( \mathscr{M}_{l,k-1} - (F_{l,k-1})_{k-1} \right) \\
& - \sum_{l=1}^{d} \left( \frac{1}{1 - \frac{1}{\sqrt{\beta_2^n}}} \frac{n^2 \triangle_{n(k-1)}}{\sqrt{v_{k-1,0}}} + \sqrt{\frac{2\rho_3^2}{\beta_2^n}} \frac{1}{\left(1 - \frac{(1-\beta_2)4n\rho_2}{\beta_2^n}\right)} \delta_1 \right)(1-\beta_1)(\beta_1 + \beta_1^n)n \sum_{i=0}^{n-1} |\partial_l f_i(x_{k-1,0})| \\
& - d\sqrt{\frac{2\rho_3^2}{\beta_2^n}}(1-\beta_1)(\beta_1 + \beta_1^n)n^2 \left(Q_k + Q_{k-1}\right) - d\sqrt{\frac{2\rho_3^2}{\beta_2^n}}(1-\beta_1)\left(\beta_1 + \beta_1^n\right)n^3 \triangle_{n(k-1)}
\end{aligned}
$$

$$
\begin{aligned}
\overset{\text{Lemma G.10 and G.3}}{\geq} \quad & \textcolor{red}{\sum_{l=1}^{d} \mathbb{I}_{k-1} \frac{\partial_l f(x_{k-1,0})}{\sqrt{v_{k-1,0}}} \left( \mathscr{M}_{l,k-1} - (F_{l,k-1})_{k-1} \right)} \\
& - d \left( \frac{1}{1 - \frac{1}{\sqrt{\beta_2^n}}} \sqrt{\frac{2}{n\beta_2^n}} \right)(1-\beta_1)(\beta_1 + \beta_1^n)n^3 \triangle_{n(k-1)} \\
& - \sum_{l=1}^{d} \sqrt{\frac{2\rho_3^2}{\beta_2^n}} \frac{1}{\left(1 - \frac{(1-\beta_2)4n\rho_2}{\beta_2^n}\right)} \delta_1 (1-\beta_1)(\beta_1 + \beta_1^n)n \sum_{i=0}^{n-1} \left( |\partial_l f_i(x_{k-1,0})| \right) \\
& - d\sqrt{\frac{2\rho_3^2}{\beta_2^n}}(1-\beta_1)(\beta_1 + \beta_1^n)n^2 \left(Q_k + Q_{k-1}\right) - d\sqrt{\frac{2\rho_3^2}{\beta_2^n}}(1-\beta_1)\left(\beta_1 + \beta_1^n\right)n^3 \triangle_{n(k-1)}
\end{aligned}
$$

The red term will be handled using color-ball method when taking conditional expectation $Ex_{k-1}(\cdot)$.
Now we have derived lower bounds for both $(a)$ and $(b)$. Combining together, we have:

$$\mathbb{E}_{k-1}\left\{\mathbb{E}_k\left[\sum_{l=1}^{d}\mathbb{I}_k\frac{\partial_l f(x_{k,0})}{\sqrt{v_{k,0}}}\left((M'_{l,k})_k-(F_{l,k})_k\right)+\sum_{l=1}^{d}\mathbb{I}_k\frac{\partial_l f(x_{k,0})}{\sqrt{v_{k,0}}}\left((M'_{l,k})_{k-1}-(F_{l,k})_{k-1}\right)\right]\right\}$$

$$\overset{(59)}{\geq}\mathbb{E}_{k-1}\left\{\underbrace{\sum_{l=1}^{d}\mathbb{I}_k\frac{\partial_l f(x_{k,0})}{\sqrt{v_{k,0}}}(1-\beta_1)J_1\sum_{i=0}^{n-1}\partial_l f_i(x_{k,0})}_{(a)}+\underbrace{\sum_{l=1}^{d}\mathbb{I}_k\frac{\partial_l f(x_{k,0})}{\sqrt{v_{k,0}}}\left((M'_{l,k})_{k-1}-(F_{l,k})_{k-1}\right)}_{(b)}\right\}$$

$$-d\sqrt{\frac{2\rho_3^2}{\beta_2^n}}\frac{(1-\beta_1)n^2(n-1)\triangle_{nk}}{2}$$

$$\geq\mathbb{E}_{k-1}\left\{\sum_{l=1}^{d}\mathbb{I}_{k-1}\frac{\partial_l f(x_{k-1,0})}{\sqrt{v_{k-1,0}}}(1-\beta_1)J_1\sum_{i=0}^{n-1}\partial_l f_i(x_{k-1,0})+\sum_{l=1}^{d}\mathbb{I}_{k-1}\frac{\partial_l f(x_{k-1,0})}{\sqrt{v_{k-1,0}}}\left(\mathscr{M}_{l,k-1}-(F_{l,k-1})_{k-1}\right)\right\}$$

$$-d(1-\beta_1)|J_1|\frac{1}{1-\frac{1}{\sqrt{\beta_2^n}}}n^2\triangle_{n(k-1)}\sqrt{\frac{2}{n\beta_2^n}}$$

$$-\sum_{l=1}^{d}(1-\beta_1)|J_1|\left(\left|\sum_{i=0}^{n-1}\partial_l f_i(x_{k,0})\right|+n^2\triangle_{n(k-1)}\right)\left(\sqrt{\frac{2\rho_3^2}{\beta_2^n}}\frac{1}{\left(1-\frac{(1-\beta_2)4n\rho_2}{\beta_2^n}\right)}\delta_1\right)$$

$$-2(1-\beta_1)d\sqrt{\frac{2\rho_3^2}{\beta_2^n}}|J_1|n^2\triangle_{n(k-1)}-(1-\beta_1)d\sqrt{\frac{2\rho_3^2}{\beta_2^n}}|J_1|n(Q_k+Q_{k-1})$$

$$-d\left(\frac{1}{1-\frac{1}{\sqrt{\beta_2^n}}}\sqrt{\frac{2}{n\beta_2^n}}\right)(1-\beta_1)(\beta_1+\beta_1^n)n^2\triangle_{n(k-1)}$$

$$-\sum_{l=1}^{d}\sqrt{\frac{2\rho_3^2}{\beta_2^n}}\frac{1}{\left(1-\frac{(1-\beta_2)4n\rho_2}{\beta_2^n}\right)}\delta_1(1-\beta_1)(\beta_1+\beta_1^n)n\left(\sum_{i=0}^{n-1}|\partial_l f_i(x_{k,0})|+n^3\triangle_{n(k-1)}\right)$$

$$-d\sqrt{\frac{2\rho_3^2}{\beta_2^n}}(1-\beta_1)(\beta_1+\beta_1^n)n^2(Q_k+Q_{k-1})-d\sqrt{\frac{2\rho_3^2}{\beta_2^n}}(1-\beta_1)(\beta_1+\beta_1^n)n^3\triangle_{n(k-1)}$$

$$-d\sqrt{\frac{2\rho_3^2}{\beta_2^n}}\frac{(1-\beta_1)n^2(n-1)\triangle_{nk}}{2}$$

For the first term in the above inequality, we can calculate it using the idea in the color-ball toy example:

$$\mathbb{E}_{k-1}\left\{\sum_{l=1}^{d}\mathbb{I}_{k-1}\frac{\partial_l f(x_{k-1,0})}{\sqrt{v_{k-1,0}}}(1-\beta_1)J_1\sum_{i=0}^{n-1}\partial_l f_i(x_{k-1,0})+\sum_{l=1}^{d}\mathbb{I}_{k-1}\frac{\partial_l f(x_{k-1,0})}{\sqrt{v_{k-1,0}}}\left(\mathscr{M}_{l,k-1}-(F_{l,k-1})_{k-1}\right)\right\}$$

$$=\sum_{l=1}^{d}\mathbb{I}_{k-1}\frac{\partial_l f(x_{k-1,0})}{\sqrt{v_{k-1,0}}}(1-\beta_1)\beta_1^n J_1\sum_{i=0}^{n-1}\partial_l f_i(x_{k-1,0}).$$

To proceed, we further take $\mathbb{E}_{k-2}(\cdot)$ and bound

$$\mathbb{E}_{k-2}\left\{\underbrace{\sum_{l=1}^{d}\mathbb{I}_{k-1}\frac{\partial_l f(x_{k-1,0})}{\sqrt{v_{k-1,0}}}(1-\beta_1)\beta_1^n J_1\sum_{i=0}^{n-1}\partial_l f_i(x_{k-1,0})}_{(a)}+\underbrace{\sum_{l=1}^{d}\mathbb{I}_k\frac{\partial_l f(x_{k,0})}{\sqrt{v_{k,0}}}\left((M'_{l,k})_{k-2}-(F_{l,k})_{k-2}\right)}_{(b)}\right\}.$$

Repeat this process until $k=1$, we have:

$$\mathbb{E}\left[\sum_{l \text{ large}} \frac{\partial_l f(x_{k,0})}{\sqrt{v_{k,0}}}\left(\left(M'_{l,k}\right)_k - (F_{l,k})_k + \cdots + \left(M'_{l,k}\right)_1 - (F_{l,k})_1\right)\right]$$

$$\geq (1-\beta_1)\beta_1^{kn}J_1\mathbb{E}\left[\sum_{l=1}^d \mathbb{I}_1 \frac{\partial_l f(x_{1,0})}{\sqrt{v_{1,0}}}\sum_{i=0}^{n-1}\partial_l f_i(x_{1,0})\right]$$

$$+\text{Error}_1 + \text{Error}_2 + \text{Error}_3$$

$$\overset{(*)}{\geq} -\frac{\beta_1^n}{\sqrt{k-1}}(1-\beta_1)\left|J_1\mathbb{E}\left[\sum_{l=1}^d \mathbb{I}_1 \frac{\partial_l f(x_{1,0})}{\sqrt{v_{1,0}}}\sum_{i=0}^{n-1}\partial_l f_i(x_{1,0})\right]\right|$$

$$+\text{Error}_1 + \text{Error}_2 + \text{Error}_3$$

where $(*)$ holds for large $k$ such that $\beta^{(k-1)n} \leq \frac{\beta_1^n}{\sqrt{k-1}}$. We specify $\text{Error}_1, \text{Error}_2, \text{Error}_3$ as follows.

$$\text{Error}_1 = -d(1-\beta_1)|J_1|\frac{1}{1-\frac{1}{\sqrt{\beta_2^n}}}n^2\sqrt{\frac{2}{n\beta_2^n}}\left(\sum_{j=0}^\infty \beta_1^{jn}\triangle_{n(k-j)}\right)$$

$$-\sum_{l=1}^d (1-\beta_1)|J_1|\left(\left|\sum_{i=0}^{n-1}\partial_l f_i(x_{k,0})\right|\right)\left(\sqrt{\frac{2\rho_3^2}{\beta_2^n}}\frac{1}{\left(1-\frac{(1-\beta_2)4n\rho_2}{\beta_2^n}\right)}\delta_1\right)\left(\sum_{j=0}^\infty \beta_1^{jn}(j+1)\right)$$

$$-d(1-\beta_1)\triangle_{n(k-1)}|J_1|n^2\left(\sqrt{\frac{2\rho_3^2}{\beta_2^n}}\frac{1}{\left(1-\frac{(1-\beta_2)4n\rho_2}{\beta_2^n}\right)}\delta_1\right)\left(\sum_{j=0}^\infty \beta_1^{jn}(j+1)^2\right)$$

$$-2(1-\beta_1)d\sqrt{\frac{2\rho_3^2}{\beta_2^n}}|J_1|n^2\left(\sum_{j=0}^\infty \beta_1^{jn}\triangle_{n(k-1-j)}\right)$$

$$-(1-\beta_1)d\sqrt{\frac{2\rho_3^2}{\beta_2^n}}|J_1|n\left[Q_k + Q_{k-1} + \beta_1^n(Q_k + Q_{k-1} + Q_{k-2}) + \cdots\right]$$

Since $\delta_2 = \lim_{k\to\infty}\sum_{j=1}^{k-1}(\beta_1^n)^j\sqrt{\frac{k}{k-j}}$ is a finite constant, we have $\sum_{j=0}^\infty \beta_1^{jn}\triangle_{n(k-j)} = \frac{\triangle_1\delta_2}{\sqrt{nk}}$. In addition, we have

$$[Q_k + Q_{k-1} + \beta_1^n(Q_k + Q_{k-1} + Q_{k-2}) + \cdots]$$

$$\overset{(*)}{\leq} Q_k(1 + \beta_1^n + \cdots)$$
$$+Q_{k-1} + \beta_1^n(Q_{k-1} + Q_{k-2}) + \beta_1^{2n}(Q_{k-1} + Q_{k-2} + Q_{k-3}) + \cdots$$

$$\overset{(*)}{\leq} Q_k(1 + \beta_1^n + \cdots) + 2Q_{k-1} + 2\left(\beta_1^n 2Q_{k-1} + \beta_1^{2n}3Q_{k-1} + \cdots\right)$$

$$\overset{\text{Lemma F.1}}{\leq} Q_k\frac{1}{(1-\beta_1^n)} + 2Q_{k-1}\frac{1}{(1-\beta_1^n)^2}$$

$$\leq 5Q_k\frac{1}{(1-\beta_1^n)^2}$$

where $(*)$ uses the following fact: consider integers $k > J > 0$, we have $\sum_{j=1}^J \frac{1}{\sqrt{k-j}} \leq 2\frac{J}{\sqrt{k-1}}$. This inequality can be simply proved by taking the integral over $\frac{1}{\sqrt{k}}$. The final inequality is due to $Q_{k-1} < 2Q_k$. Now, we have

$$\text{Error}_1 \geq -d(1-\beta_1)|J_1|\frac{1}{1-\frac{1}{\sqrt{\beta_2^n}}}n^2\sqrt{\frac{2}{n\beta_2^n}}\frac{\triangle_1\delta_2}{\sqrt{nk}}$$

$$-\sum_{l=1}^{d}(1-\beta_1)|J_1|\left(\left|\sum_{i=0}^{n-1}\partial_l f_i(x_{k,0})\right|\right)\left(\sqrt{\frac{2\rho_3^2}{\beta_2^n}}\frac{1}{\left(1-\frac{(1-\beta_2)4n\rho_2}{\beta_2^n}\right)}\delta_1\right)\frac{1}{(1-\beta_1^n)^2}$$

$$-2d(1-\beta_1)|J_1|n^2\left(\sqrt{\frac{2\rho_3^2}{\beta_2^n}}\frac{1}{\left(1-\frac{(1-\beta_2)4n\rho_2}{\beta_2^n}\right)}\delta_1\right)\frac{1+\beta_1^n}{(1-\beta_1^n)^3}\frac{\triangle_1}{\sqrt{n(k-1)}}$$

$$-(1-\beta_1)d\sqrt{\frac{2\rho_3^2}{\beta_2^n}}|J_1|n^2\frac{\triangle_1\delta_2}{\sqrt{n(k-1)}}-(1-\beta_1)d\sqrt{\frac{2\rho_3^2}{\beta_2^n}}|J_1|n5Q_k\frac{1}{(1-\beta_1^n)^2}$$

$$\text{Error}_2 = -d\left(\frac{1}{1-\frac{1}{\sqrt{\beta_2^n}}}\sqrt{\frac{2}{n\beta_2^n}}\right)(1-\beta_1)(\beta_1+\beta_1^n)n^3\left(\sum_{j=0}^{\infty}\beta_1^{jn}\triangle_{n(k-1-j)}\right)$$

$$-\sum_{l=1}^{d}\sqrt{\frac{2\rho_3^2}{\beta_2^n}}\frac{1}{\left(1-\frac{(1-\beta_2)4n\rho_2}{\beta_2^n}\right)}\delta_1(1-\beta_1)(\beta_1+\beta_1^n)n\left(\sum_{i=0}^{n-1}|\partial_l f_i(x_{k,0})|\right)\left(\sum_{j=0}^{\infty}(j+1)\beta_1^{nj}\right)$$

$$-d\sqrt{\frac{2\rho_3^2}{\beta_2^n}}\frac{1}{\left(1-\frac{(1-\beta_2)4n\rho_2}{\beta_2^n}\right)}\delta_1(1-\beta_1)(\beta_1+\beta_1^n)n^3\triangle_{n(k-1)}\left(\sum_{j=0}^{\infty}(j+1)^2\beta_1^{nj}\right)$$

$$-d\sqrt{\frac{2\rho_3^2}{\beta_2^n}}(1-\beta_1)(\beta_1+\beta_1^n)n^2\left((Q_k+Q_{k-1})+\beta_1^n(Q_k+Q_{k-1}+Q_{k-2})+\cdots\right)$$

$$-d\sqrt{\frac{2\rho_3^2}{\beta_2^n}}(1-\beta_1)(\beta_1+\beta_1^n)n^3\left(\sum_{j=0}^{\infty}\beta_1^{jn}\triangle_{n(k-1-j)}\right)$$

Based on the calculation in Lemma F.1, we have

$$\text{Error}_2 \geq -d\left(\frac{1}{1-\frac{1}{\sqrt{\beta_2^n}}}\sqrt{\frac{2}{n\beta_2^n}}\right)(1-\beta_1)(\beta_1+\beta_1^n)n^3\frac{\triangle_1}{\sqrt{n}}\frac{\delta_2}{\sqrt{k}}$$

$$-\sum_{l=1}^{d}\sqrt{\frac{2\rho_3^2}{\beta_2^n}}\frac{1}{\left(1-\frac{(1-\beta_2)4n\rho_2}{\beta_2^n}\right)}\delta_1(1-\beta_1)(\beta_1+\beta_1^n)n\left(\sum_{i=0}^{n-1}|\partial_l f_i(x_{k,0})|\right)\frac{1}{(1-\beta_1^n)^2}$$

$$-d\sqrt{\frac{2\rho_3^2}{\beta_2^n}}\frac{1}{\left(1-\frac{(1-\beta_2)4n\rho_2}{\beta_2^n}\right)}\delta_1(1-\beta_1)(\beta_1+\beta_1^n)n^3\frac{1+\beta_1^n}{(1-\beta_1^n)^3}\frac{\triangle_1\delta_2}{\sqrt{n(k-1)}}$$

$$-d\sqrt{\frac{2\rho_3^2}{\beta_2^n}}(1-\beta_1)(\beta_1+\beta_1^n)n^2 2Q_k\frac{1}{\beta_1^n(1-\beta_1^n)^2}$$

$$-d\sqrt{\frac{2\rho_3^2}{\beta_2^n}}(1-\beta_1)(\beta_1+\beta_1^n)n^3\frac{\triangle_1}{\sqrt{n}}\frac{\delta_2}{\sqrt{k}}.$$

$$\text{Error}_3 = -d\sqrt{\frac{2\rho_3^2}{\beta_2^n}}(1-\beta_1)n^3\left(\sum_{j=0}^{\infty}\beta_1^{jn}\triangle_{n(k-j)}\right) = -\frac{d\sqrt{\frac{2\rho_3^2}{\beta_2^n}}(1-\beta_1)n^3}{\sqrt{k}}\frac{\triangle_1}{\sqrt{n}}\delta_2,$$

Since $\frac{1-\beta_1}{1-\beta_1^n} \leq 1$ and $|J_1| \leq n$, we have

$$|J_1|\left(\sqrt{\frac{2\rho_3^2}{\beta_2^n}\frac{1}{\left(1-\frac{(1-\beta_2)4n\rho_2}{\beta_2^n}\right)}}\right)\frac{1-\beta_1}{(1-\beta_1^n)^2} + \sqrt{\frac{2\rho_3^2}{\beta_2^n}\frac{1}{\left(1-\frac{(1-\beta_2)4n\rho_2}{\beta_2^n}\right)}}\frac{2n(1-\beta_1)}{(1-\beta_1^n)^2}$$

$$\leq \left(\sqrt{\frac{2\rho_3^2}{\beta_2^n}\frac{3n}{\left(1-\frac{1-\beta_2}{2}\left(-1+\frac{4\rho_2 n}{\beta_2^n}\right)\right)}}\right)\frac{1}{(1-\beta_1^n)}.$$

Using the fact that $\frac{1}{\sqrt{k-1}} \leq \frac{\sqrt{2}}{\sqrt{k}}$ (for $k \geq 2$), we have:

$$\mathbb{E}\left[\sum_{l \text{ large}}\frac{\partial_l f(x_{k,0})}{\sqrt{v_{k,0}}}\left((M'_{l,k})_k - (F_{l,k})_k + \cdots + (M'_{l,k})_1 - (F_{l,k})_1\right)\right]$$

$$\geq -\frac{\beta_1^n}{\sqrt{k-1}}(1-\beta_1)\left|J_1\mathbb{E}\left[\sum_{l=1}^d \mathbb{I}_1\frac{\partial_l f(x_{1,0})}{\sqrt{v_{1,0}}}\sum_{i=0}^{n-1}\partial_l f_i(x_{1,0})\right]\right|$$

$$+\text{Error}_1 + \text{Error}_2 + \text{Error}_3$$

$$\geq -\frac{G_6}{\sqrt{k}} - G_7\delta_1\mathbb{E}\left(\sum_{l=1}^d\sum_{i=0}^{n-1}|\partial_l f_i(x_{k,0})|\right),$$

where

$$G_6 := \beta_1^n(1-\beta_1)\sqrt{2}|J_1|\sqrt{\frac{2\rho_3^2}{\beta_2^n}}\mathbb{E}\left[\sum_{l=1}^d\left|\mathbb{I}_1\sum_{i=0}^{n-1}\partial_l f_i(x_{1,0})\right|\right] + \frac{\triangle_1}{\sqrt{n}}\delta_2 d\sqrt{\frac{2\rho_3^2}{\beta_2^n}}(1-\beta_1)n^3$$

$$+d(1-\beta_1)|J_1|\frac{1}{1-\frac{1}{\sqrt{\beta_2^n}}}n^2\sqrt{\frac{2}{n\beta_2^n}}\frac{\triangle_1\delta_2}{\sqrt{n}}$$

$$+d(1-\beta_1)|J_1|n^2\left(\sqrt{\frac{2\rho_3^2}{\beta_2^n}\frac{1}{\left(1-\frac{(1-\beta_2)4n\rho_2}{\beta_2^n}\right)}}\delta_1\right)\frac{1+\beta_1^n}{(1-\beta_1^n)^3}\frac{\triangle_1\sqrt{2}}{\sqrt{n}}$$

$$+2(1-\beta_1)d\sqrt{\frac{2\rho_3^2}{\beta_2^n}}|J_1|n^2\frac{\triangle_1\delta_2\sqrt{2}}{\sqrt{n}} - 5(1-\beta_1)d\sqrt{\frac{2\rho_3^2}{\beta_2^n}}|J_1|n\frac{1}{(1-\beta_1^n)^2}\triangle_1 n\sqrt{n}\frac{32\sqrt{2}}{(1-\beta_2)^n}\frac{1}{\beta_2^n}$$

$$+d\left(\frac{1}{1-\frac{1}{\sqrt{\beta_2^n}}}\sqrt{\frac{2}{n\beta_2^n}}\right)(1-\beta_1)(\beta_1+\beta_1^n)n^3\frac{\triangle_1}{\sqrt{n}}\delta_2$$

$$+d\sqrt{\frac{2\rho_3^2}{\beta_2^n}\frac{1}{\left(1-\frac{(1-\beta_2)4n\rho_2}{\beta_2^n}\right)}}\delta_1(1-\beta_1)(\beta_1+\beta_1^n)n^3\frac{1+\beta_1^n}{(1-\beta_1^n)^3}\frac{\triangle_1\delta_2\sqrt{2}}{\sqrt{n}}$$

$$+2d\sqrt{\frac{2\rho_3^2}{\beta_2^n}}(1-\beta_1)(\beta_1+\beta_1^n)n^2\frac{1}{\beta_1^n(1-\beta_1^n)^2}\triangle_1 n\sqrt{n}\frac{32\sqrt{2}}{(1-\beta_2)^n}\frac{1}{\beta_2^n}$$

$$+d\sqrt{\frac{2\rho_3^2}{\beta_2^n}}(1-\beta_1)(\beta_1+\beta_1^n)n^3\frac{\triangle_1}{\sqrt{n}}\delta_2,$$

$$G_7 := \left(\sqrt{\frac{2\rho_3^2}{\beta_2^n}\frac{3n}{\left(1-\frac{1-\beta_2}{2}\left(-1+\frac{4\rho_2 n}{\beta_2^n}\right)\right)}}\right)\frac{1}{(1-\beta_1^n)}$$

where $J_1 = \left(-\frac{1}{n}\beta_1 - \frac{2}{n}\beta_1^2 - \cdots - \frac{n-1}{n}\beta_1^{n-1}\right)$, $\delta_2 = \lim_{k\to\infty} \sum_{j=1}^{k-1} (\beta_1^n)^j \sqrt{\frac{k}{k-j}}$ (if needed, we can further bound $J_1$ by $n$ for simplicity). This conclude the proof.

## G.8 Proof of Lemma G.5

We now derive upper bounds for $\sum_{l\text{ large}} (a_2)$ and $\sum_{l\text{ large}} (b_2)$. The upper bound for $\sum_{l\text{ large}} (a_2)$ is very straightforward using inequality (24).

$$\sum_{l\text{ large}} (a_2) = \delta_1 \sqrt{\frac{2\rho_3^2}{\beta_2^n}} \sum_{l\text{ large}} \sum_{i=0}^{n-1} |\partial_l f_i (x_{k,0})| \overset{(24)}{\leq} \delta_1 \sqrt{\frac{2\rho_3^2}{\beta_2^n}} \sqrt{D_1} \rho_1 d \left( |\partial_\alpha f (x_{k,0})| + \sqrt{\frac{D_0}{D_1 d}} \right).$$

Now we shift gear to $\sum_{l\text{ large}} (b_2) := \delta_1 \sqrt{\frac{2\rho_3^2}{\beta_2^n}} \sum_{l\text{ large}} \sum_{i=0}^{n-1} |m_{l,k,i} - \partial_l f_i(x_{k,0})|$. To proceed, we need an upper bound for $\sum_{l\text{ large}} \sum_{i=0}^{n-1} |m_{l,k,i} - \partial_l f_i(x_{k,0})|$. For each $i$, we perform the following decomposition.

$$\sum_{l\text{ large}} |m_{l,k,i} - \partial_l f_i(x_{k,0})| \leq \underbrace{\sum_{l=1}^{d} |(1 - \beta_1) \left[\beta_1^i \partial_l f_{\tau_{k,0}}(x_{k,0}) + \cdots + \partial_l f_{\tau_{k,i}}(x_{k,i})\right] - \partial_l f_i(x_{k,0})|}_{(d_1)}$$

$$+ \underbrace{\sum_{l=1}^{d} |\beta_1^{i+1} m_{l,k-1,n-1}|}_{(d_2)}. \tag{60}$$

To start, we bound $(d_1)$.

$$(d_1) \leq \sum_{l=1}^{d} \left\{ |\partial_l f_{\tau_{k,0}}(x_{k,0})| + |\partial_l f_{\tau_{k,1}}(x_{k,1})| + \cdots + |\partial_l f_{\tau_{k,i}}(x_{k,i})| + |\partial_l f_i(x_{k,0})| \right\}$$

$$\leq \sum_{l=1}^{d} \left\{ |\partial_l f_{\tau_{k,0}}(x_{k,0})| + |\partial_l f_{\tau_{k,1}}(x_{k,1})| + \cdots + |\partial_l f_{\tau_{k,n-1}}(x_{k,n-1})| + |\partial_l f_i(x_{k,0})| \right\}$$

$$\leq \sum_{l=1}^{d} \sum_{i=0}^{n-1} \left\{ |\partial_l f_i(x_{k,0})| \right\} + \sum_{l=1}^{d} |\partial_l f_i(x_{k,0})| + d\triangle_{nk} + \cdots + nd\triangle_{nk}$$

$$\overset{(24)}{\leq} 2\sqrt{D_1}\rho_1 d \left( |\partial_\alpha f (x_{k,0})| + \sqrt{\frac{D_0}{D_1 d}} \right) + \frac{n(n+1)d}{2}\triangle_{nk}$$

$$= 2\sqrt{D_1}\rho_1 d \left( |\partial_\alpha f (x_{k,0})| + \sqrt{\frac{D_0}{D_1 d}} \right) + \frac{(n+1)\sqrt{n}d}{2\sqrt{k}}\triangle_1. \tag{61}$$

Now, we bound $(d_2)$. Recall $\sum_{i=0}^{n-1} \partial_l f_i(x_{1,0}) = \partial_l f(x_{1,0})$, we have

$$|m_{l,k-1,n-1}| \leq (1 - \beta_1) \left[ |\partial_l f_{\tau_{k-1,n-1}}(x_{k-1,n-1})| + \beta_1 |\partial_l f_{\tau_{k-1,n-2}}(x_{k-1,n-2})| + \cdots \right]$$

$$+ \beta_1^{(k-1)n} \sum_{i=0}^{n-1} |\partial_l f_i(x_{1,0})| \tag{62}$$

Note that for any $i \in [0, n-1]$, $j \in [0, n-1]$, $t \in [1, k-1]$, we have the following result.

$$\left|\partial_l f_i(x_{k-t,j})\right| \leq \left|\partial_l f_i(x_{k,0})\right| + \left|\partial_l f_i(x_{k,0}) - \partial_l f_i(x_{k-t,j})\right|$$

$$\overset{\text{Lemma } F.2}{\leq} \left|\partial_l f_i(x_{k,0})\right| + (n-j)\triangle_{(k-t)n} + n\triangle_{(k-t+1)n} + \cdots + n\triangle_{(k-1)n}$$

$$\leq \left|\partial_l f_i(x_{k,0})\right| + \triangle_{(k-1)n} + \triangle_{(k-1)n-1} + \cdots + \triangle_{(k-1)n-[(n-j)+(t-1)n-1]}$$

$$\leq \left|\partial_l f_i(x_{k,0})\right| + \triangle_{(k-1)n-1} + \triangle_{(k-1)n-2} + \cdots + \triangle_{(k-1)n-[(n-j)+(t-1)n]}$$

$$\overset{(*)}{\leq} \left|\partial_l f_i(x_{k,0})\right| + \frac{2[(n-j)+(t-1)n]\triangle_1}{\sqrt{n(k-1)-1}}, \tag{63}$$

where $(*)$ uses the following fact: consider integers $k > J > 0$, we have $\sum_{j=1}^{J} \frac{1}{\sqrt{k-j}} \leq 2\frac{J}{\sqrt{k-1}}$.

Plugging (63) into (62) and re-arranging the index, we have

$$\sum_{l=1}^{d} |m_{l,k-1,n-1}| \overset{(62)}{\leq} (1-\beta_1)\sum_{l=1}^{d}\left[\left|\partial_l f_{\tau_{k-1,n-1}}(x_{k-1,n-1})\right| + \beta_1\left|\partial_l f_{\tau_{k-1,n-2}}(x_{k-1,n-2})\right| + \cdots\right]$$

$$+\beta_1^{(k-1)n}\sum_{l=1}^{d}\sum_{i=0}^{n-1}\left|\partial_l f_i(x_{1,0})\right|$$

$$\overset{(24),(63)}{\leq} (1-\beta_1)\sum_{q=1}^{(k-1)n}\beta_1^{q-1}\left[\sqrt{D_1}\rho_1 d\left(\left|\partial_\alpha f(x_{k,0})\right| + \sqrt{\frac{D_0}{D_1 d}}\right) + \frac{2qd\triangle_1}{\sqrt{n(k-1)-1}}\right]$$

$$+\beta_1^{(k-1)n}\sum_{l=1}^{d}\sum_{i=0}^{n-1}\left|\partial_l f_i(x_{1,0})\right|$$

$$\leq (1-\beta_1)\sum_{q=1}^{\infty}\beta_1^{q-1}\left[\sqrt{D_1}\rho_1 d\left(\left|\partial_\alpha f(x_{k,0})\right| + \sqrt{\frac{D_0}{D_1 d}}\right) + \frac{2qd\triangle_1}{\sqrt{n(k-1)-1}}\right]$$

$$+\beta_1^{(k-1)n}\sum_{l=1}^{d}\sum_{i=0}^{n-1}\left|\partial_l f_i(x_{1,0})\right|$$

$$\overset{\text{Lemma } F.1}{\leq} \sqrt{D_1}\rho_1 d\left(\left|\partial_\alpha f(x_{k,0})\right| + \sqrt{\frac{D_0}{D_1 d}}\right) + \frac{1}{1-\beta_1}\frac{2d\triangle_1}{\sqrt{n(k-1)-1}}$$

$$+\beta_1^{(k-1)n}\sum_{l=1}^{d}\sum_{i=0}^{n-1}\left|\partial_l f_i(x_{1,0})\right|$$

$$\overset{\text{When } k \geq 4}{\leq} \sqrt{D_1}\rho_1 d\left(\left|\partial_\alpha f(x_{k,0})\right| + \sqrt{\frac{D_0}{D_1 d}}\right) + \frac{1}{1-\beta_1}\frac{2\sqrt{2}d\triangle_1}{\sqrt{nk}}$$

$$+\beta_1^{(k-1)n}\sum_{l=1}^{d}\sum_{i=0}^{n-1}\left|\partial_l f_i(x_{1,0})\right|$$

$$\overset{(i)}{\leq} \sqrt{D_1}\rho_1 d\left(\left|\partial_\alpha f(x_{k,0})\right| + \sqrt{\frac{D_0}{D_1 d}}\right) + \frac{1}{1-\beta_1}\frac{2\sqrt{2}d\triangle_1}{\sqrt{nk}}$$

$$+\frac{\sqrt{2}\beta_1^n}{\sqrt{k}}\sum_{l=1}^{d}\sum_{i=0}^{n-1}\left|\partial_l f_i(x_{1,0})\right| \tag{64}$$

where the second last inequality holds because: when $k \geq 4$, $\frac{1}{\sqrt{n(k-1)-1}} \leq \frac{\sqrt{2}}{\sqrt{nk}}$. $(i)$ holds when $k$ is large enough such that $\beta_1^{(k-1)n} \leq \frac{\beta_1^n}{\sqrt{k-1}}$. Further, $\frac{\beta_1^n}{\sqrt{k-1}} \leq \frac{\sqrt{2}\beta_1^n}{\sqrt{k}}$ when $k \geq 2$.

Now, we have derived upper bounds for $(d_1)$ and $(d_2)$. Plugging (64) and (61) into (60), we conclude the proof.

$$\sum_{l=1}^{d}\sum_{i=0}^{n-1}|m_{l,k,i} - \partial_l f_{\tau_{k,i}}(x_{k,0})| \overset{(61),(64)}{\leq} 3n\sqrt{D_1}\rho_1 d\left(|\partial_\alpha f(x_{k,0})| + \sqrt{\frac{D_0}{D_1 d}}\right) + \frac{(n+1)nd\sqrt{n}}{2\sqrt{k}}\Delta_1$$

$$+\frac{nd}{1-\beta_1}\frac{2\sqrt{2}\Delta_1}{\sqrt{nk}} + \frac{\sqrt{2}n\beta_1^n}{\sqrt{k}}\sum_{l=1}^{d}\sum_{i=0}^{n-1}|\partial_l f_i(x_{1,0})|$$

$$= \frac{1}{\sqrt{k}}\left[\frac{d(n+1)n^{\frac{3}{2}}}{2}\Delta_1 + \frac{d2\sqrt{2}\sqrt{n}\Delta_1}{1-\beta_1} + \sqrt{2}n\beta_1^n\sum_{i=0}^{n-1}\|\nabla f_i(x_{1,0})\|_1\right]$$

$$+3n\sqrt{D_1}\rho_1 d\left(|\partial_\alpha f(x_{k,0})| + \sqrt{\frac{D_0}{D_1 d}}\right). \tag{65}$$

$$\sum_{l \text{ large}} (b_2) = \delta_1\sqrt{\frac{2\rho_3^2}{\beta_2^n}}\sum_{l \text{ large}}\sum_{i=0}^{n-1}|m_{l,k,i} - \partial_l f_i(x_{k,0})|$$

$$\leq \delta_1\sqrt{\frac{2\rho_3^2}{\beta_2^n}}\left\{\frac{1}{\sqrt{k}}\left[\frac{d(n+1)n^{\frac{3}{2}}}{2}\Delta_1 + \frac{d2\sqrt{2}\sqrt{n}\Delta_1}{1-\beta_1} + \sqrt{2}n\beta_1^n\sum_{i=0}^{n-1}\|\nabla f_i(x_{1,0})\|_1\right]\right.$$

$$\left. +3n\sqrt{D_1}\rho_1 d\left(|\partial_\alpha f(x_{k,0})| + \sqrt{\frac{D_0}{D_1 d}}\right)\right\}.$$

The proof is concluded by adding expectation on both sides of the inequality.

### G.9 Proof of Lemma G.6

$$\mathbb{E}\left\{\sum_{l \text{ large}}(a_1) + \sum_{l \text{ large}}(b_1) - \sum_{l \text{ large}}\{(a_2)+(b_2)\}\right\}$$

$$= \mathbb{E}\left[\sum_{l \text{ large}}\frac{\partial_l f(x_{k,0})^2}{\sqrt{v_{l,k,0}}}\right] + \mathbb{E}\left[\sum_{l \text{ large}}\frac{\partial_l f(x_{k,0})}{\sqrt{v_{l,k,0}}}\sum_{i=0}^{n-1}(m_{l,k,i} - \partial_l f_i(x_{k,0}))\right]$$

$$-d\delta_1\sqrt{\frac{2\rho_3^2}{\beta_2^n}}\mathbb{E}\left[\sum_{i=0}^{n-1}|\partial_l f_i(x_{k,0})|\right] - d\delta_1\sqrt{\frac{2\rho_3^2}{\beta_2^n}}\mathbb{E}\left[\sum_{i=0}^{n-1}|m_{l,k,i} - \partial_l f_i(x_{k,0})|\right]$$

$$\overset{\text{Lemma G.4 and G.5 and F.3}}{\geq} \mathbb{E}\left[\sum_{l \text{ large}}\frac{\partial_l f(x_{k,0})^2}{\sqrt{v_{l,k,0}}}\right] - \frac{1}{\sqrt{k}}(G_4 + G_5 + G_6) - G_7\delta_1\sqrt{D_1}\rho_1 d\left(\mathbb{E}|\partial_\alpha f(x_{k,0})| + \sqrt{\frac{D_0}{D_1 d}}\right)$$

$$-\delta_1\sqrt{\frac{2\rho_3^2}{\beta_2^n}}\sqrt{D_1}\rho_1 d\left(\mathbb{E}|\partial_\alpha f(x_{k,0})| + \sqrt{\frac{D_0}{D_1 d}}\right)$$

$$-\delta_1\sqrt{\frac{2\rho_3^2}{\beta_2^n}}\left[\frac{d(n+1)n^{\frac{3}{2}}}{2}\Delta_1 + \frac{d2\sqrt{2}\sqrt{n}\Delta_1}{1-\beta_1} + \sqrt{2}n\beta_1^n\sum_{i=0}^{n-1}\mathbb{E}\|\nabla f_i(x_{1,0})\|_1\right]\frac{1}{\sqrt{k}}$$

$$-\delta_1\sqrt{\frac{2\rho_3^2}{\beta_2^n}}3n\sqrt{D_1}\rho_1 d\left(\mathbb{E}|\partial_\alpha f(x_{k,0})| + \sqrt{\frac{D_0}{D_1 d}}\right),$$

where constant terms $G_4, G_5, G_6, G_7$ can be seen at the end of Appendix G.7.

Since $\alpha = \arg\max_{l=1,2,\cdots,d}|\partial_l f(x_{k,0})|$ and $\sum_{l \text{ large}}\frac{\partial_l f(x_{k,0})^2}{\sqrt{v_{l,k,0}}} \geq \frac{\partial_\alpha f(x_{k,0})^2}{\sqrt{v_{\alpha,k,0}}}$, we have

$$\mathbb{E}\left\{\sum_{l \text{ large}} (a_1) + \sum_{l \text{ large}} (b_1) - \sum_{l \text{ large}} \{(a_2) + (b_2)\}\right\}$$

$$\geq \quad \mathbb{E}\left\{\frac{\partial_\alpha f(x_{k,0})^2}{\sqrt{v_{\alpha,k,0}}}\right\}$$

$$- \left[G_4 + G_5 + G_6 + \delta_1\sqrt{\frac{2\rho_3^2}{\beta_2^n}}\left(\frac{d(n+1)n^{\frac{3}{2}}}{2}\Delta_1 + \frac{d2\sqrt{2}\sqrt{n}\triangle_1}{1-\beta_1} + \sqrt{2}n\beta_1^n\sum_{i=0}^{n-1}\mathbb{E}\|\nabla f_i(x_{1,0})\|_1\right)\right]\frac{1}{\sqrt{k}}$$

$$- \delta_1\left(\sqrt{\frac{2\rho_3^2}{\beta_2^n}}4n + G_7\right)\sqrt{D_1}\rho_1 d\left(\mathbb{E}|\partial_\alpha f(x_{k,0})| + \sqrt{\frac{D_0}{D_1 d}}\right)$$

$$= \quad \mathbb{E}\left\{\frac{\partial_\alpha f(x_{k,0})^2}{\sqrt{v_{\alpha,k,0}}}\right\} - F_2\frac{1}{\sqrt{k}} - F_3\mathbb{E}|\partial_\alpha f(x_{k,0})| - F_4, \tag{66}$$

where $F_2 := \delta_1\sqrt{\frac{2\rho_3^2}{\beta_2^n}}\left[\frac{d(n+1)n^{\frac{3}{2}}}{2}\Delta_1 + \frac{d2\sqrt{2}\sqrt{n}\triangle_1}{1-\beta_1} + \sqrt{2}n\beta_1^n\sum_{i=0}^{n-1}\|\nabla f_i(x_{1,0})\|_1\right] + G_4 + G_5 + G_6$;

$F_3 := \delta_1\left(\sqrt{\frac{2\rho_3^2}{\beta_2^n}}4n + G_7\right)\sqrt{D_1}\rho_1 d$;

$F_4 := \delta_1\left(\sqrt{\frac{2\rho_3^2}{\beta_2^n}}4n + G_7\right)\sqrt{D_1}\rho_1 d\sqrt{\frac{D_0}{D_1 d}}$.

Now, we discuss two cases.

**Case (a): when** $|\partial_\alpha f(x_{k,0})| \geq 4\sqrt{2}\frac{\Delta_1}{(1-\beta_2)\sqrt{D_1 nkd}}$. In this case, we have the following result.

$$\frac{\partial_\alpha f(x_{k,0})^2}{\sqrt{v_{\alpha,k,0}}} \quad \geq \quad \frac{\partial_\alpha f(x_{k,0})^2}{\sqrt{(1-\beta_2)\left(|\partial_\alpha f_{\tau_{k,0}}(x_{k,0})|^2 + \beta_2|\partial_\alpha f_{\tau_{k-1,n-1}}(x_{k-1,n-1})|^2 + \cdots\right)}}$$

$$\geq \quad \frac{\partial_\alpha f(x_{k,0})^2}{\sqrt{(1-\beta_2)\left(|\partial_\alpha f_{\tau_{k,0}}(x_{k,0})|^2 + \beta_2|\partial_\alpha f_{\tau_{k-1,n-1}}(x_{k,0}) + \triangle_{n(k-1)}|^2\cdots\right)}}$$

$$\overset{\text{Lemma F.3}}{\geq} \quad \frac{\partial_\alpha f(x_{k,0})^2}{\sqrt{(1-\beta_2)\left(\sum_{j=0}^\infty\left(\sqrt{|\partial_\alpha f(x_{k,0})|^2 + \frac{D_0}{D_1 d}}\sqrt{D_1 d} + \sum_{t=1}^j \Delta_{n(k-1)-t}\right)^2\beta_2^j\right)}}$$

$$\geq \quad \frac{\partial_\alpha f(x_{k,0})^2}{\sqrt{(1-\beta_2)\left(\sum_{j=0}^\infty\beta_2^j\left(\left(|\partial_\alpha f(x_{k,0})|^2 + \frac{D_0}{D_1 d}\right)D_1 d + 4\sqrt{2}j\triangle_{nk}\sqrt{|\partial_\alpha f(x_{k,0})|^2 + \frac{D_0}{D_1 d}}\sqrt{D_1 d} + 8j^2\triangle_{nk}^2\right)\right)}}$$

$$\overset{\text{Lemma F.1}}{\geq} \quad \frac{\partial_\alpha f(x_{k,0})^2}{\sqrt{D_1 d\left(\left(|\partial_\alpha f(x_{k,0})|^2 + \frac{D_0}{D_1 d}\right) + \sqrt{|\partial_\alpha f(x_{k,0})|^2 + \frac{D_0}{D_1 d}}\frac{4\sqrt{2}\triangle_{nk}}{(1-\beta_2)\sqrt{D_1 d}} + \frac{16\triangle_{nk}^2}{D_1 d(1-\beta_2)^2}\right)}}$$

$$\overset{\text{Case (a)}}{\geq} \quad \frac{\partial_\alpha f(x_{k,0})^2}{\sqrt{\frac{5}{2}D_1 d\left(|\partial_\alpha f(x_{k,0})|^2 + \frac{D_0}{D_1 d}\right)}},$$

Now we consider the following two sub-cases.

- When $\partial_\alpha f(x_{k,0})^2 \leq \frac{D_0}{D_1 d}$:

$$\frac{\partial_\alpha f(x_{k,0})^2}{\sqrt{v_{\alpha,k,0}}} - F_2\frac{1}{\sqrt{k}} - F_3|\partial_\alpha f(x_{k,0})| - F_4$$

$$\geq \frac{\partial_\alpha f(x_{k,0})^2}{\sqrt{5D_0}} - F_2\frac{1}{\sqrt{k}} - F_3\sqrt{\frac{D_0}{D_1 d}} - F_4$$

- When $\partial_\alpha f(x_{k,0})^2 \geq \frac{D_0}{D_1 d}$:

$$\frac{\partial_\alpha f(x_{k,0})^2}{\sqrt{v_{\alpha,k,0}}} - F_2 \frac{1}{\sqrt{k}} - F_3 |\partial_\alpha f(x_{k,0})| - F_4$$

$$\geq \frac{|\partial_\alpha f(x_{k,0})|}{\sqrt{5D_1 d}} - F_2 \frac{1}{\sqrt{k}} - F_3 |\partial_\alpha f(x_{k,0})| - F_4$$

$$= |\partial_\alpha f(x_{k,0})| \left( \frac{1}{\sqrt{5D_1 d}} - F_3 \right) - F_2 \frac{1}{\sqrt{k}} - F_4$$

$$\geq |\partial_\alpha f(x_{k,0})| \left( \frac{1}{\sqrt{5D_1 d}} - F_3 \right) - F_2 \frac{1}{\sqrt{k}} - F_3 \sqrt{\frac{D_0}{D_1 d}} - F_4$$

Combining together, we have the following results for **Case (a)**.

$$\left\{ \frac{\partial_\alpha f(x_{k,0})^2}{\sqrt{v_{\alpha,k,0}}} \right\} - F_2 \frac{1}{\sqrt{k}} - F_3 |\partial_\alpha f(x_{k,0})| - F_4 \geq \min\left\{ \frac{\partial_\alpha f(x_{k,0})^2}{\sqrt{5D_0}}, |\partial_\alpha f(x_{k,0})| \left( \frac{1}{\sqrt{5D_1 d}} - F_3 \right) \right\} - F_2 \frac{1}{\sqrt{k}} - F_3 \sqrt{\frac{D_0}{D_1 d}} - F_4.$$

**Case (b):** When $|\partial_\alpha f(x_{k,0})| < 4\sqrt{2} \frac{\Delta_1}{(1-\beta_2)\sqrt{D_1 n k d}}$, we have

$$\left\{ \frac{\partial_\alpha f(x_{k,0})^2}{\sqrt{v_{\alpha,k,0}}} \right\} - F_2 \frac{1}{\sqrt{k}} - F_3 |\partial_\alpha f(x_{k,0})| - F_4 \geq -F_2 \frac{1}{\sqrt{k}} - F_3 |\partial_\alpha f(x_{k,0})| - F_4$$

$$\geq -F_2 \frac{1}{\sqrt{k}} - F_3 4\sqrt{2} \frac{\Delta_1}{(1-\beta_2)\sqrt{D_1 n k d}} - F_4$$

$$= -\frac{1}{\sqrt{k}} G_1 - F_4, \qquad (67)$$

where $G_1 := F_2 + F_3 4\sqrt{2} \frac{\Delta_1}{(1-\beta_2)\sqrt{D_1 n d}}$. Now, the following two claims are both true.

- Claim 1:

$$(67) \geq \frac{\partial_\alpha f(x_{k,0})^2}{\sqrt{5D_0}} - \frac{1}{k} \frac{\left( 4\sqrt{2} \frac{\Delta_1}{(1-\beta_2)\sqrt{D_1 n d}} \right)^2}{\sqrt{5D_0}} - \frac{1}{\sqrt{k}} G_1 - F_4.$$

- Claim 2:

$$(67) \geq |\partial_\alpha f(x_{k,0})| \left( \frac{1}{\sqrt{5D_1 d}} - F_3 \right) - \frac{1}{\sqrt{k}} 4\sqrt{2} \frac{\Delta_1}{(1-\beta_2)\sqrt{D_1 n d}} \left( \frac{1}{\sqrt{5D_1 d}} - F_3 \right)$$
$$- \frac{1}{\sqrt{k}} G_1 - F_4.$$

Combining Claim 1 and Claim 2, we have

$$(67) \geq \min\left\{ \frac{\partial_\alpha f(x_{k,0})^2}{\sqrt{5D_0}}, |\partial_\alpha f(x_{k,0})| \left( \frac{1}{\sqrt{5D_1 d}} - F_3 \right) \right\} - F_4$$

$$- \frac{1}{\sqrt{k}} \left[ \max\left\{ \frac{\left( 4\sqrt{2} \frac{\Delta_1}{(1-\beta_2)\sqrt{D_1 n d}} \right)^2}{\sqrt{5D_0}}, \frac{4\sqrt{2}\Delta_1}{(1-\beta_2)\sqrt{D_1 n d}} \left( \frac{1}{\sqrt{5D_1 d}} - F_3 \right) \right\} + G_1 \right]$$

$$:= \min\left\{ \frac{\partial_\alpha f(x_{k,0})^2}{\sqrt{5D_0}}, |\partial_\alpha f(x_{k,0})| \left( \frac{1}{\sqrt{5D_1 d}} - F_3 \right) \right\} - \frac{1}{\sqrt{k}} G_2 - F_4, \qquad (68)$$

where $G_2 := \max\left\{ \dfrac{\left(4\sqrt{2}\frac{\Delta_1}{(1-\beta_2)\sqrt{D_1 n d}}\right)^2}{\sqrt{5D_0}}, \dfrac{4\sqrt{2}\Delta_1}{(1-\beta_2)\sqrt{D_1 n d}}\left(\dfrac{1}{\sqrt{5D_1 d}} - F_3\right) \right\} + G_1$.

Combining **Case (a)** and **Case (b)** together, we have

$$
\mathbb{E}\left\{ \sum_{l\ \text{large}}(a_1) + \sum_{l\ \text{large}}(b_1) - \sum_{l\ \text{large}}\{(a_2)+(b_2)\} \right\} \geq \mathbb{E}\min\left\{ \dfrac{\partial_\alpha f\left(x_{k,0}\right)^2}{\sqrt{5D_0}}, |\partial_\alpha f\left(x_{k,0}\right)|\left(\dfrac{1}{\sqrt{5D_1 d}} - F_3\right) \right\}
$$

$$
-\dfrac{1}{\sqrt{k}}G_2 - F_4
$$

$$
\geq \mathbb{E}\min\left\{ \dfrac{\|\nabla f(x_{k,0})\|_2^2}{d\sqrt{5D_0}}, \|\nabla f(x_{k,0})\|_1\left(\dfrac{1}{d\sqrt{5D_1 d}} - \dfrac{F_3}{d}\right) \right\}
$$

$$
-\dfrac{1}{\sqrt{k}}G_2 - F_4,
$$

where the last inequality is because of $\|\nabla f(x_{k,0})\|_2^2 \leq d\partial_\alpha f\left(x_{k,0}\right)^2$, $\|\nabla f(x_{k,0})\|_1 \leq d|\partial_\alpha f\left(x_{k,0}\right)|$.

Recall $F_3 \to 0$ when $\beta_2 \to 1$, so there exists an interval $(1-\epsilon, 1]$, such that $\frac{1}{\sqrt{5D_1 d}} - F_3 \geq \frac{1}{\sqrt{10D_1 d}}$, or equivalently $F_3 \leq \frac{1}{\sqrt{10D_1 d}}$ (note that $F_3$ is the same as "$A(\beta_2)$" stated in the condition of Lemma G.6). With such a choice of $\beta_2$, we have the following results by changing all the $F_3$ into $\frac{1}{\sqrt{10D_1 d}}$:

$$
\mathbb{E}\left\{ \sum_{l\ \text{large}}(a_1) + \sum_{l\ \text{large}}(b_1) - \sum_{l\ \text{large}}\{(a_2)+(b_2)\} \right\} \geq \mathbb{E}\min\left\{ \dfrac{\|\nabla f(x_{k,0})\|_2^2}{d\sqrt{5D_0}}, \|\nabla f(x_{k,0})\|_1 \dfrac{1}{d\sqrt{10D_1 d}} \right\}
$$

$$
-\dfrac{1}{\sqrt{k}}G_2 - F_4.
$$

$$
\overset{(*)}{\geq} \dfrac{1}{d\sqrt{10D_1 d}}\mathbb{E}\min\left\{ \sqrt{\dfrac{2D_1 d}{D_0}}\|\nabla f(x_{k,0})\|_2^2, \|\nabla f(x_{k,0})\|_1 \right\}
$$

$$
-\dfrac{1}{\sqrt{k}}G_2 - F_5.
$$

In inequality $(*)$, we change $F_4$ into $F_5 := \sqrt{\frac{D_0}{D_1 d}}\frac{1}{\sqrt{10D_1 d}}$. This is because: first, $F_4 = F_3\sqrt{\frac{D_0}{D_1 d}}$; second, $F_3 \leq \frac{1}{\sqrt{10D_1 d}}$.

The proof of Lemma G.6 is completed.

We restate all the constants as follows ($G_4, G_5, G_6$ are specified in Appendix G.7):

$$
G_2 := \max\left\{ \dfrac{\left(4\sqrt{2}\frac{\Delta_1}{(1-\beta_2)\sqrt{D_1 n d}}\right)^2}{\sqrt{5D_0}}, \dfrac{4\sqrt{2}\Delta_1}{(1-\beta_2)\sqrt{D_1 n d}\sqrt{10D_1 d}} \right\} + G_1;
$$

$$
G_1 := F_2 + 4\sqrt{2}\dfrac{\Delta_1}{(1-\beta_2)\sqrt{D_1 n d}\sqrt{10D_1 d}};
$$

$$
F_2 := \delta_1 \sqrt{\dfrac{2\rho_3^2}{\beta_2^n}}\left[ \dfrac{d(n+1)n^{\frac{3}{2}}}{2}\Delta_1 + \dfrac{d2\sqrt{2}\sqrt{n}\triangle_1}{1-\beta_1} + \sqrt{2}n\beta_1^n \sum_{i=0}^{n-1}\|\nabla f_i(x_{1,0})\|_1 \right] + G_4 + G_5 + G_6;
$$

$$
F_5 := \sqrt{\dfrac{D_0}{D_1 d}}\dfrac{1}{\sqrt{10D_1 d}}.
$$

*Remark* G.14. We comment that we can always replace $F_5$ in the final result by $F_4$, i.e., we can choose not to apply the last inequality $(*)$ in the proof. The benefit of using $F_4$ is that $F_4$ monotonously decrease to 0 when increasing $\beta_2$ to 1. This monotone property is not shown in the notation of $F_5$. Nevertheless, we choose to use $F_5$ since it is a much cleaner constant.

## G.10 Proof of Lemma G.9

Based on Descent Lemma, we have

$$
\sum_{k=t_0}^{T} \frac{\eta_0}{\sqrt{nk}}(\text{r.h.s. of (35)}) \quad \leq \quad \sum_{k=t_0}^{T} \mathbb{E}\left\langle \nabla f\left(x_{k,0}\right), x_{k,0} - x_{k+1,0}\right\rangle
$$

$$
\leq \quad \mathbb{E}f\left(x_{t_0,0}\right) - \mathbb{E}f\left(x_{T+1,0}\right) + \sum_{k=t_0}^{T} \frac{L}{2}\mathbb{E}\left\|x_{k+1,0} - x_{k,0}\right\|_2^2
$$

$$
\overset{(21)}{\leq} \quad \mathbb{E}f\left(x_{t_0,0}\right) - f^* + \sum_{k=t_0}^{T} \frac{Ld}{2}\left(\frac{n\eta_0}{\sqrt{nk}}\frac{(1-\beta_1)}{\sqrt{1-\beta_2}}\frac{1}{1-\frac{\beta_1}{\sqrt{\beta_2}}}\right)^2
$$

$$
= \quad \mathbb{E}f\left(x_{t_0,0}\right) - f^* + \sum_{k=t_0}^{T} \frac{Ldn}{2k}\left(\eta_0\frac{(1-\beta_1)}{\sqrt{1-\beta_2}}\frac{1}{1-\frac{\beta_1}{\sqrt{\beta_2}}}\right)^2
$$

Plugging in the r.h.s. of (35), we have the following relation after rearranging.

$$
\sum_{k=t_0}^{T} \frac{\eta_0}{\sqrt{nk}}\left[\frac{1}{d\sqrt{10D_1d}}\mathbb{E}\min\left\{\sqrt{\frac{2D_1d}{D_0}}\|\nabla f(x_{k,0})\|_2^2, \|\nabla f(x_{k,0})\|_1\right\} - \mathbb{E}\left[\mathcal{O}(\frac{1}{\sqrt{k}})\right] - \mathcal{O}(\sqrt{D_0})\right]
$$

$$
\leq \mathbb{E}f\left(x_{t_0,0}\right) - f^* + \sum_{k=t_0}^{T} \frac{1}{k}\left[\frac{Ldn}{2}\left(\eta_0\frac{(1-\beta_1)}{\sqrt{1-\beta_2}}\frac{1}{1-\frac{\beta_1}{\sqrt{\beta_2}}}\right)^2\right].
$$

Recall we have $2\left(\sqrt{T} - \sqrt{t_0-1}\right) \leq \sum_{k=t_0}^{T}\frac{1}{\sqrt{k}}$, $\sum_{k=t_0}^{T}\frac{1}{k} \leq \log\frac{T+1}{t_0}$. Further, since $\|\cdot\|_1 \geq \|\cdot\|_2$, we get the following relation when $t_0 = 1$.

$$
\min_{k\in[1,T]}\mathbb{E}\left[\min\left\{\sqrt{\frac{2D_1d}{D_0}}\|\nabla f(x_{k,0})\|_2^2, \|\nabla f(x_{k,0})\|_2\right\}\right]
$$

$$
\leq \frac{1}{\sqrt{T}}\frac{d\sqrt{10D_1dn}}{2\eta_0}\left[\mathbb{E}f\left(x_{1,0}\right) - f^* + \log(T+1)\left(\frac{Ldn}{2}\left(\eta_0\frac{(1-\beta_1)}{\sqrt{1-\beta_2}}\frac{1}{1-\frac{\beta_1}{\sqrt{\beta_2}}}\right)^2 + H_2\right)\right] + F_5,
$$

$$
\tag{69}
$$

We specify all the constant as follows. For all the following constants, We keep the same notation as their appearance in their corresponding lemmas.

$F_5 := \sqrt{\frac{D_0}{D_1d}}\frac{1}{\sqrt{10D_1d}}$;

$H_2 := \frac{\eta_0(G_2+dF_1)}{\sqrt{n}}$;

$F_1 := \triangle_1 n^2\sqrt{n}\frac{32\sqrt{2}}{(1-\beta_2)^n\beta_2^n}\frac{1-\beta_1}{\sqrt{1-\beta_2}}\frac{1}{1-\frac{\beta_1}{\sqrt{\beta_2}}}n$;

$G_2 := \max\left\{\frac{\left(4\sqrt{2}\frac{\triangle_1}{(1-\beta_2)\sqrt{D_1nd}}\right)^2}{\sqrt{5D_0}}, \frac{4\sqrt{2}\triangle_1}{(1-\beta_2)\sqrt{D_1nd}\sqrt{10D_1d}}\right\} + G_1$;

$G_1 := F_2 + 4\sqrt{2}\frac{\triangle_1}{(1-\beta_2)\sqrt{D_1nd}\sqrt{10D_1d}}$;

$F_2 := \delta_1\sqrt{\frac{2\rho_3^2}{\beta_2^n}}\left[\frac{d(n+1)n^{\frac{3}{2}}}{2}\triangle_1 + \frac{d2\sqrt{2}\sqrt{n}\triangle_1}{1-\beta_1} + \sqrt{2}n\beta_1^n\sum_{i=0}^{n-1}\|\nabla f_i(x_{1,0})\|_1\right] + G_4 + G_5 + G_6$;

$$G_5 = d\sqrt{\frac{2\rho_3^2}{\beta_2^n}}\left(\frac{\beta_1^{2n}2n^3\triangle_1\sqrt{2}}{\sqrt{n}}\frac{1-\beta_1}{(1-\beta_1^n)^2} + n\left(1-\beta_1^{n-1}\right)\sum_{i=0}^{n-1}\mathbb{E}|\partial_\alpha f_i(x_{1,0})|\frac{(1-\beta_1)\beta_1^n\sqrt{2}}{1-\beta_1^n}\right);$$

$$G_4 = d\sqrt{\frac{2\rho_3^2}{\beta_2^n}}\beta_1^n\sqrt{2}n(n-1)\sum_{i=0}^{n-1}\mathbb{E}\left(|\partial_\alpha f_i(x_{1,0})|\right);$$

$$
\begin{aligned}
G_6 \quad := \quad & \beta_1^n(1-\beta_1)\sqrt{2}\,|J_1|\sqrt{\frac{2\rho_3^2}{\beta_2^n}}\mathbb{E}\left[\sum_{l=1}^{d}\left|\mathbb{I}_1\sum_{i=0}^{n-1}\partial_l f_i(x_{1,0})\right|\right] + \frac{\triangle_1}{\sqrt{n}}\delta_2 d\sqrt{\frac{2\rho_3^2}{\beta_2^n}}(1-\beta_1)n^3 \\[2mm]
& + d(1-\beta_1)|J_1|\frac{1}{1-\frac{1}{\sqrt{\beta_2^n}}}n^2\sqrt{\frac{2}{n\beta_2^n}}\frac{\triangle_1\delta_2}{\sqrt{n}} \\[2mm]
& + d(1-\beta_1)|J_1|n^2\left(\sqrt{\frac{2\rho_3^2}{\beta_2^n}}\frac{1}{\left(1-\frac{(1-\beta_2)4n\rho_2}{\beta_2^n}\right)}\delta_1\right)\frac{1+\beta_1^n}{(1-\beta_1^n)^3}\frac{\triangle_1\sqrt{2}}{\sqrt{n}} \\[2mm]
& + 2(1-\beta_1)d\sqrt{\frac{2\rho_3^2}{\beta_2^n}}|J_1|n^2\frac{\triangle_1\delta_2\sqrt{2}}{\sqrt{n}} - 5(1-\beta_1)d\sqrt{\frac{2\rho_3^2}{\beta_2^n}}|J_1|n\frac{1}{(1-\beta_1^n)^2}\triangle_1n\sqrt{n}\frac{32\sqrt{2}}{(1-\beta_2)^n\,\beta_2^n} \\[2mm]
& + d\left(\frac{1}{1-\frac{1}{\sqrt{\beta_2^n}}}\sqrt{\frac{2}{n\beta_2^n}}\right)(1-\beta_1)(\beta_1+\beta_1^n)n^3\frac{\triangle_1}{\sqrt{n}}\delta_2 \\[2mm]
& + d\sqrt{\frac{2\rho_3^2}{\beta_2^n}}\frac{1}{\left(1-\frac{(1-\beta_2)4n\rho_2}{\beta_2^n}\right)}\delta_1(1-\beta_1)(\beta_1+\beta_1^n)n^3\frac{1+\beta_1^n}{(1-\beta_1^n)^3}\frac{\triangle_1\delta_2\sqrt{2}}{\sqrt{n}} \\[2mm]
& + 2d\sqrt{\frac{2\rho_3^2}{\beta_2^n}}(1-\beta_1)(\beta_1+\beta_1^n)n^2\frac{1}{\beta_1^n(1-\beta_1^n)^2}\triangle_1n\sqrt{n}\frac{32\sqrt{2}}{(1-\beta_2)^n\,\beta_2^n} \\[2mm]
& + d\sqrt{\frac{2\rho_3^2}{\beta_2^n}}(1-\beta_1)\left(\beta_1+\beta_1^n\right)n^3\frac{\triangle_1}{\sqrt{n}}\delta_2,
\end{aligned}
$$

$$\Delta_1 := \eta_0\frac{L\sqrt{d}}{\sqrt{1-\beta_2}}\frac{1-\beta_1}{1-\frac{\beta_1}{\sqrt{\beta_2}}};$$

$$\delta_1 := \frac{(1-\beta_2)4n\rho_2}{\beta_2^n} + \left(\frac{1}{\sqrt{\beta_2^n}}-1\right);$$

where $J_1 = \left(-\frac{1}{n}\beta_1 - \frac{2}{n}\beta_1^2 - \cdots - \frac{n-1}{n}\beta_1^{n-1}\right)$, $\delta_2 = \lim_{k\to\infty}\sum_{j=1}^{k-1}(\beta_1^n)^j\sqrt{\frac{k}{k-j}} = \frac{\beta_1^n}{1-\beta_1^n}$. If needed, we can further bound $J_1$ by $n$ for simplicity.

Further, $\rho_1, \rho_2, \rho_3$ are constants satisfying the following conditions for $\forall l = 1, \cdots, d$. Usually, these constants can be different for different problems. In the worst case, we have $0 \le \rho_3 \le \sqrt{n}\rho_1 \le n$. $\rho_3$ is larger when $\partial_l f_i(x_{k,0})$ are more aligned.

$$\rho_1 \ge \frac{\sum_{i=1}^{n}|\partial_l f_i(x_{k,0})|}{\sqrt{\sum_{i=1}^{n}|\partial_l f_i(x_{k,0})|^2}};$$

$$\rho_2 \ge \frac{|\max_i \partial_l f_i(x_{k,0})|^2}{\frac{1}{n}\sum_{i=1}^{n}|\partial_l f_i(x_{k,0})|^2};$$

$$\rho_3 \ge \frac{\left|\sum_{i=1}^{n}\partial_l f_i(x_{k,0})\right|}{\sqrt{\frac{1}{n}\sum_{i=1}^{n}|\partial_l f_i(x_{k,0})|^2}}.$$

The proof of Lemma G.9 is now completed. This also concludes the whole proof of Theorem 3.1.

## G.11 Dissussion on Bias Correction Terms and Non-Zero $\epsilon$ (Hyperparameter for Numerical Stability)

In the above analysis, we focus on Adam without bias correction terms and we consider $\epsilon = 0$ ($\epsilon$ is the hyperparameter for numerical stability in Algorithm 1). For completeness, we now briefly discuss how to incorporate the bias correction terms and non-zero $\epsilon$ into our analysis above. Based on the current convergence proof, we only requires several additional simple changes. We list them as follows.

**Adam with bias correction terms:** This bias correction terms are introduced by (Kingma & Ba, 2014)). It can be implemented by (1) changing the stepsize $\eta_k$ into $\hat{\eta}_k = \frac{\sqrt{1-\beta_1^k}}{1-\beta_1^k}\eta_k = \frac{\sqrt{1-\beta_1^k}}{1-\beta_1^k}\frac{\eta_0}{\sqrt{k}}$; (2) change the initialization of Algorithm 1 into $m_{1,-1} = v_{1,-1} = 0$. More details can be seen in (Kingma & Ba, 2014)). We now explain how to inlcude this two changes into our analysis.

(1) Change of stepsize: We observe that the new stepsize $\hat{\eta}_k$ is well bounded around the old stepsize $\eta_k$, i.e., $\hat{\eta}_k \in [\sqrt{1-\beta_2}\eta_k, \frac{1}{1-\beta_1}\eta_k]$. Therefore, to prove the convergence of Adam with $\hat{\eta}_k$, we add the folliwing steps into the current proof.

- Whenever we need an upper bound on $\hat{\eta}_k$, we use $\hat{\eta}_k \leq \frac{1}{1-\beta_1}\eta_k$. Then we follow the original analysis with an extra constant $\frac{1}{1-\beta_1}$. This step will appear in Lemma F.2. It turns out we only need to change the constant $\triangle_{nk}$ in Lemma F.2 into $\frac{1}{1-\beta_1}\frac{\eta_0}{\sqrt{nk}}\frac{L\sqrt{d}}{\sqrt{1-\beta_2}}\frac{1-\beta_1}{1-\frac{\beta_1}{\sqrt{\beta_2}}}$. The rest of the analysis remains the same.

- Whenver we need a lower bound on $\hat{\eta}_k$, we use $\hat{\eta}_k \geq \sqrt{1-\beta_2}\eta_k$. Then we follow the original analysis with an extra constant $\sqrt{1-\beta_2}$. This step will appear in Lemma G.9 and we only need to change the constant terms in the final result. The rest of the analysis remains the same.

(2) Change of initialization: In our current analysis, we use initialization $m_{1,-1} = \nabla f(x_0)$ and $v_{1,-1} = \max_i \nabla f_i(x_0) \circ \nabla f_i(x_0)$. Now we explain how to prove convergence with initialization $m_{1,-1} = v_{1,-1} = 0$.

- We use $m_{1,-1} = \nabla f(x_0)$ at Lemma G.11, in which we bound the difference between $M_{l,k}$ and $M'_{l,k}$. The goal of this lemma is to control $M_{l,k} - M'_{l,k} = \mathcal{O}(\beta_1^{(k-1)n})$.

  As explained in the proof of Lemma G.11, after expanding $M_{l,k}$ and $M'_{l,k}$ into serieses, they only differ when "$k \leq 0$" (as shown in equation (48)). When we use $m_{1,-1} = 0$, half of the terms in equation (48) will become 0. However, it does not affect the result of Lemma G.11 since all the terms in equation (48) are (at least) weighted by $\beta_1^{(k-1)n}$. So even if half of them to be 0, the rest of the terms is still in the order of $\beta_1^{(k-1)n}$. Therefore, Lemma G.11 still holds with a few changes on the constant terms.

- We use $v_{1,-1} = \max_i \nabla f_i(x_0) \circ \nabla f_i(x_0)$ at Lemma G.1,which is mainly based on Lemma F.1 in (Shi et al., 2020). According to (Shi et al., 2020): to include bias correction terms into analysis, we just need to add one more constraint $k > \frac{8\sqrt{2}}{1-\beta_2^n} + 1$ and then we can reach the same conclusion.

**Adam with non-zero $\epsilon$:** In our current analysis, we consider $\epsilon = 0$. In practice, $\epsilon$ is often set to be a small postive number such as $10^{-8}$. Proving convergence with $\epsilon > 0$ is strictly simpler. It only requires a few simple changes based on the current proof. We explain as below.

When $\epsilon \neq 0$, the new 2nd-order momentum becomes $\hat{v}_{k,0} := \sqrt{v_{k,0}} + \epsilon$. This brings the following changes:

- Whenever we want an upper bound on $\hat{v}_{k,0}$, we can choose one of the following upper bound.
  - When $\sqrt{v_{k,0}} \geq \epsilon$, we have $\hat{v}_{k,0} \leq 2\sqrt{v_{k,0}}$. Then, we follow the same steps in the current proof with minor changes on the constant.
  - When $\sqrt{v_{k,0}} < \epsilon$, we have $\hat{v}_{k,0} \leq 2\epsilon$. This step, again, decouple the statistical dependency between $\nabla f_{\tau k,0}(x)$ and $\sqrt{v_{k,0}}$. It changes Adam into SGD and thus simplifies the proof. Many technical lemmas could be skipped in this case.

- Whenever we want a lower bound on $\hat{v}_{k,0}$, we have $\hat{v}_{k,0} \geq \epsilon$. This step decouples the statistical dependency between $\nabla f_{\tau k,0}(x)$ and $\sqrt{v_{k,0}}$. It changes Adam into SGD and thus simplifies the proof. Many technical lemmas could be skipped in this case.