# OpenReview forum: "Adam Can Converge Without Any Modification On Update Rules"
_NeurIPS.cc/2022/Conference — NeurIPS 2022 Accept_

### Official Review · Reviewer_xyuf · 2022-07-03

**Rating:** 8
**Confidence:** 4
**Soundness:** 4 excellent
**Presentation:** 4 excellent
**Contribution:** 4 excellent

**Summary:**

This paper proves that Adam with hyper-parameters $\beta_1 \leq \sqrt{\beta_2}<1$ and large $\beta_2$ converges to a neighborhood of stationary points under mild assumptions on the function class and if Strong Growth Condition is satisfied, it converges to stationary point. Specifically, they argue that analyzing Adam with large $\beta_1$ which results in large momentum is harder and previous works could not manage to handle non-zero $\beta_1$.

**Questions:**

Question:
1) in lines 123-124, can you tell me (intuitively) why SGC is reasonable in overparametrized regime?can you point out to more literature (if any) regarding SGC?

2) You say in line 98 that in your theoretical analysis you let $\epsilon$ be an arbitrary non-negative constant including 0, but in proofs I couldn't catch that. It seems to me you just set it to 0. Please explain.

3) in line 625, is it like you can $\beta_1=0$ and large $\beta_2$ can converge repeating the proof of Reddi? or is it a fundamentally different approach?

4) in lines 196-199, I can't understand why smaller batchsize needs a larger $\beta_2$. The bound $\gamma_1(n)$ increases with n, so it seems to me that smaller $n$ has smaller $\gamma_1(n)$ hence smaller $\beta_2$ is okay.

Grammatical Typos:

line 93, randomly. line 125, converge "to". line 161, require. line 205, "and". line 262, "red". line 328, "reduces". line 509, "some more". line 597 and 622, "require".  line 658, "as". line 677, "an". line 690, converge"s". line 732, this. line 732, "a" instead of "an".

Math typos:

line 225 and 699, I think for $i>0$, $f_i(x)=1/2(x+2)^2"+"3/2$.

line 291, equation 3, why there is no sign of $v_{k,0}$ in r.h.s (which is the case in equation 4) ?

line 311, definition of $m_{l,k,i}$ should have $f_\tau$ rather than $f$.

bottom of page 22, $\beta_2$s in nominators may change to $\beta_1$.This also occurs in line 734, first line.

line 725, first line, $j=0$.

line 739, I feel a missing $1\sqrt{k}$ in r.h.s. Also in line 740, there might be a missing $n$ in r.h.s. because of the summation.

just above line 789, definition of $v$ should have $\beta_2$ rather than $\beta_1$.

line 793, eq 17, $f_{\tau_{k,i+1}}(x_{k,i})$ should change to $f_{\tau_{k,i}}(x_{k,i+1})$, which is corrected in line 795.

line 815, $x_k - x_{k+1}$ instead of $x_{k+1} - x_{k}$.

line 837, 2nd line, based on eq (18) there might be a missing $\frac{1}{\sqrt{1-\beta_2}}$, which just affects $F_1$.

in the lines following line 1052, the subscript $\ell$ is missing. you can note that in the beginning of Appendix $G_3$ to prevent confusion.

equation at the bottom of page 40, based on Lemma F.3 you may have an extra factor of $n$, as Lemma F.3 deals with $f_\tau$ rather than $f$.

bottom of page 46, $-\frac{1}{\sqrt{k}}$.

in equation 45, I feel like $f_0$ should cancel out in all epochs, just like epoch 0.

in lines 1159 and 1166, Lemma G.12 gives $\beta_2^n$ in denominator rather than nominator. Please correct it.

line 1163, second equation, it is $ (F_{l,k}) _{k-1}$.

**Limitations:**

regarding negative societal impact, as authors pinpoint, training neural nets for illegal use is the main concern of this kind of work.

**Strengths And Weaknesses:**

This paper is superbly written and the proofs, although being long, are managed to be as clear as possible. The writing style is excellent and helps reader understand the main ideas, specially the long proofs. I checked all the proofs, except for pages 56-58 and 60-64. Except for some typos, which do not affect the results, I could not find any flaw. I really like how you manage to use color-ball method and finite vs infinite sum argument in your work. Also as pointed out in several places, you explain the short-comings of previous work in analyzing non-zero $\beta_1$.The removal of bounded gradient assumption and sticking to Assumption 2.1 which encompasses SGC and constant variance assumptions also looks entertaining. Example of giving convex functions in prop. 3.3 when Adam diverges in a relatively large region is also instructive: small $\beta_2$ is dangerous. examples on practical networks also corroborate the analytic results. This piece of work seems to have important contribution for analyzing methods with momentum like Adam and as pointed in conclusion, it paves the way for much more intricate convergence analysis of Adam and its effectiveness compared to SGD. I highly recommend this paper to be accepted.

---

> ### Author Response · Authors · 2022-08-02
> **Response to Reviewer xyuf**
>
> We would like to thank the reviewer for the encouraging and supportive feedback. We also thank the reviewer for the careful proof reading.
>
> > **Q1:**  "Why SGC is reasonable in overparametrized regime?  can you point out to more literature (if any) regarding SGC?"
>
> SGC  states that  $\sum_ {i=0}^{n-1}\left\|\nabla f_ {i}(x)\right\|_ {2}^{2} \leq D_ {1}\|\nabla f(x)\|_ {2}^{2}$.
>
> This condition implies:  When $\|\nabla f(x)\|=0$, we have $\left\|\nabla f_{j}(x)\right\|=0$ for all $j$.
> We say "this condition is reasonable
>  in the overparameterized regime" for the following reasons: First, overparameterized model often has strong expressivity to
> interpolate or fit the labelled training data completely. Second, when every data point reaches zero training loss,  the gradient of each point also converges to zero. As such, SGC holds true in the overparameterized regime.
>
> We will add more discussion to the revised script. For more information, we recommend  [1] (Section 2)  and the slides in [2] (page 14-18) for a simple intuitive explanation.
>
> [1] Fast and faster convergence of sgd for over-parameterized models and an accelerated perceptron.
>
> [2]  Interpolation, growth conditions, and stochastic gradient descent (Doctoral dissertation, University of British Columbia).
>
> > **Q2:**  "You say in line 98 that in your theoretical analysis you let $\epsilon$ be an arbitrary non-negative constant $\epsilon$ including 0, but in proofs I couldn't catch that. It seems to me you just set it to 0. Please explain."
>
> Our result holds true for any $\epsilon \geq 0$. We apologize that we only demonstrate the proof with $\epsilon = 0$ and miss out the rest. However, proving convergence with $\epsilon \neq 0$ is strictly simpler. It only requires a few simple changes based on the current proof. We explain as below.
>
> When $\epsilon \neq  0$, the new 2nd-order momentum becomes $ \widehat{v}_ {k,0} := \sqrt{ v_ {k,0}} + \epsilon$. This brings the following changes:
>
> 1.Whenever we want a lower bound on $\widehat{v}_ {k,0}$,  we  have $\widehat{v}_ {k,0} \geq \epsilon$. This step  decouples the statistical dependency between $\nabla f_ {\tau_ {k,0}}(x)$ and $\sqrt{v_ {k,0}}$. It changes Adam into SGD and thus simplifies the proof. Many technical lemmas could be skipped in this case.
>
> 2.Whenever we want an upper bound on $\widehat{v}_ {k,0}$, we can choose one of the following upper bound:
>
> - 2.1.When $\sqrt{ v_ {k,0}} \geq \epsilon$, we have $\widehat{v}_ {k,0} \leq 2 \sqrt{ v_ {k,0}}$. Then, we follow the same steps in the current proof with minor changes on the constant.
>
> - 2.2. When $\sqrt{ v_ {k,0}} < \epsilon$, we have $\widehat{v}_ {k,0} \leq 2 \epsilon$.
>   This step,  again, decouple the statistical dependency between $\nabla f_ {\tau_ {k,0}}(x)$ and  $\sqrt{v_ {k,0}}$. It changes Adam into SGD and thus simplifies the proof. Many technical lemmas could be skipped in this case.
>
> In the revised version, we will add more rigorous discussion on "non-zero $\epsilon$" in appendix.
>
> > **Q3**: "in line 625 , is it like you can $\beta_{1}=0$ and large $\beta_{2}$ can converge repeating the proof of Reddi? Or is it a fundamentally different approach?"
>
> The proof would be fundamentally different.
> For the toy example in Reddi et al., the proof of convergence (under large $\beta_2$) is simple because: 1. it is a 1-dimensional linear function. 2. All the $f_i$ are sampled sequentially without any randomness (i.e., f1 f2 f3 f1 f2 f3).  However, we consider a broad class of generic non-convex function class with random sampling strategy. It requires much different approach to prove convergence in the latter case.
>
> > **Q4:**  "in lines 196-199, I can't understand why smaller batchsize needs a larger $\beta_{2}$."
>
> Batchsize equals to (number of total sample)/(number of batches). In the context of finite-sum setting with $n$ summand, $n$ usually stands for the number of batches. For instance: In the extreme case when batchsize $=1$, $n$ equals to the number of total samples.  Therefore, smaller batchsize brings larger $n$ and hence larger $\beta_2$ is needed. We will add this explanation to the revised Experiment Section.
>
> > **Questions related to the typos:**
>
> > **Q5:** "line 291, equation 3, why there is no sign of $v_{k,0}$ in r.h.s?"
>
> Thanks for pointing it out, we will re-write this formula in the revised version.
>
> > **Q6:** "line 739 , I feel a missing $1 /\sqrt{k}$ in r.h.s. Also in line 740, there might be a missing $n$ in r.h.s. because of the summation."
>
> Since $1 /\sqrt{k} \leq 1$, the inequality in line 739 still holds. As for the missing terms due to the summation, they are all positive so we directly relax them to 0 in the lower bound.  We will add more explanations to these missing steps in the revised version.
>
> **Other grammatical and math typos:** We thank the reviewer again for taking the time to carefully read the whole script throughout. Thanks for pointing out these typos. They will be fixed in the revised version.

---

> > ### Comment · Reviewer_xyuf · 2022-08-08
> > **Response**
> >
> > I read the reviews and responses, and would like to thank authors for highlighting the revised manuscript and addressing the concerns of reviewers in detail.

---

### Official Review · Reviewer_V9yg · 2022-07-11

**Rating:** 4
**Confidence:** 5
**Soundness:** 3 good
**Presentation:** 2 fair
**Contribution:** 2 fair

**Summary:**

This manuscript studied the Adam algorithm in the finite-sum setting: the number of samples is fixed in advance and Adam can pick $(\beta_1,\beta_2)$ according to it. The authors showed that under the case $\beta_1<\sqrt{\beta_2}<1$ and large $\beta_2$, Adam can converge to the stationary point under strong growth conditions and without bounded gradient assumption. When $\beta_2$ is small, the authors also identified a region of $(\beta_1,\beta_2)$ such that Adam diverges. Some simple experiments on standard benchmark datasets are provided to justify the theoretical findings.

**Questions:**

1. Why this paper's message is important compared with Reddi et al. 2018? Why the knowledge of $n$ in finite-sum optimization is so important? What does the result look like in the online nonconvex optimization and finite-sum convex optimization setting?

2. The regret minimization might fail in certain regimes of $\beta_1,\beta_2$, but it might also converge to a stationary point without the knowledge of $n$ or even in an online setting. I don't know why Theorem 3.1 is so important since it is not comparable with Theorem 4 of Reddi et al. 2018. To establish the importance of the finite-sum setting, the authors need to provide a hardness result in an online learning setting in terms of non-convergence to a stationary point in polynomial time.

3. What is the relationship between $D_0=0$ in Assumption 2.2 and the bounded gradient? We know that Adam can converge if we have a bounded gradient (e.g., Guo et al. 2021).

4. How does the knowledge of $n$ affect the results theoretically and empirically?

If these concerns can be addressed, I am happy to increase my score.


======POST REBUTTAL======

Thank you for the authors's response. I have read other's reviews as well. My main concern still remains with the finite-sum structure, but I  decided to change my evaluation a bit after reading the rebuttal.

**Strengths And Weaknesses:**

Strengths:
1. The paper studied an important problem: the convergence of Adam in the finite-sum setting and the number of summands in the objective function is known in advance.
2. The paper is generally well-written.
3. The proof idea is clearly presented.

Weaknesses:
1. While the results are interesting, I doubt the significance and the logical flow of this result.

(1). The paper does not convey important messages about the convergence of Adam compared with Reddi et al. 2018. In particular, Reddi et al. 2018 studied Adam in the online convex optimization setting: they provided regret guarantees and non-convergence of average regret. However, this paper can only guarantee Adam's convergence with the strong growth condition and only to a stationary point. I don't see why the framework of finite-sum optimization and the knowledge of $n$ (i.e., the number of samples) would be so important to differentiate this paper from Reddi et al. 2018.

For example, it might be possible that in the particular case of  $\beta_1<\sqrt{\beta_2}<1$ for Adam, the average regret does not converge but Adam indeed converges to a stationary point in the online learning setting such as Reddi et al. 2018 (where $n$ is not available). To show the fundamental advantage of finite-sum optimization and the knowledge of $n$, the authors are expected to show something like this: for an online nonconvex optimization problem, Adam with $\beta_1<\sqrt{\beta_2}<1$ with large $\beta$ and any other parameter choice (i.e., learning rate) cannot even converge to a stationary point in polynomial time, but the finite-sum version of this problem can indeed converge in polynomial time.

(2). The convergence to stationary point only when $D_0=0$ in Theorem 3.1. What is the relationship between $D_0=0$ in Assumption 2.2 and the bounded gradient? We know that Adam can converge if we have a bounded gradient (e.g., Guo et al. 2021).

2. Experiments are rather weak. The authors did not perform an ablation study in terms of $n$ (i.e., the number of training samples). Since this is the main message of the paper, the authors are expected to show that without the knowledge of $n$, Adam would diverge due to inappropriate hyperparameter choice (e.g., learning rate, $\beta_1$ as in Theorem 3.1).

---

> ### Author Response · Authors · 2022-08-02
> **Response to Reviewer V9yg: Part (1/4)**
>
> We thank the reviewer for all the valuable comments and questions. We provide our response as below. We are happy to engage in any further discussions.
>
> > **Comment 1:  Why is finite-sum so important?**  "I don't see why the framework of finite-sum optimization  would be so important to differentiate this paper from Reddi et al. 2018." "To establish the importance  (To show the fundamental advantage) of the finite-sum setting, the authors need to provide a hardness result in an online learning setting in terms of non-convergence to a stationary point in polynomial time."
>
>
> We would like to emphasize that we do not intend to show the "fundamental advantage of finite-sum".
> **We believe both online and finite-sum are important settings. We did not try to argue that one outweighs another**.
> We choose finite-sum formulation for the following reasons:
>
> **(1)** Finite-sum problem corresponds to empirical risk minimization in deep learning tasks. It is widely used in practical applications such as NLP, CV,  GNN, etc.
>
>
> **(2)** Finite-sum is a common setting  for adaptive gradient methods and stochastic gradient methods. It is widely studied in including but not limited to [1,2, 3,4,5]
>
> [1] Chen, X., Liu, S., Sun, R., and Hong, M. On the convergence of a class of adam-type algorithms for non-convex optimization
>
> [2] De, S., Mukherjee, A., and Ullah, E. Convergence guarantees for rmsprop and adam in non-convex optimization and an empirical comparison to nesterov acceleration.
>
> [3] Shi, N., Li, D., Hong, M., and Sun, R. Rmsprop converges with proper hyper-parameter.
>
> [4] Liu, Y., Gao, Y., and Yin, W. An improved analysis of stochastic gradient descent with momentum.
>
> [5] Gower, R.M., Loizou, N., Qian, X., Sailanbayev, A., Shulgin, E. and Richtárik, P., 2019, May. SGD: General analysis and improved rates.
>
>
>
> We would also like to point out that "finite-sum setting" and "online setting" are NOT disjoint. These two settings actually have overlaps. Understanding this relation is important because **Reddi et al' divergence example actually belongs to both finite-sum  and online setting**. We elaborate as below (This may not be the real focus of your question. But this relation is crucial for understanding our motivation.)
>
>
>
> Reddi et al's online example has an equivalent counterpart in the finite setting. In particular,
>  the example in Reddi et al. 2018 can be translated to a finite-sum problem with sequential cyclic sampling order (f1, f2, f3, f1 ,f2, f3, etc). The "non-zero regret" can also be translated to "converge to  sub-optimal points for convex problem". Please refer to Appendix D.2 to see how the counter-example can be presented in the finite-sum setting.
>
> After realizing the relation above, we can discuss both the divergence example in Reddi et al. 2018 and our results  in the same context of finite-sum. When put in the same context, we find out Reddi et al. pick $n$ after fixing $(\beta_1,\beta_2)$, which is quite different from practical application. This finding motivates us to explore the behavior of Adam under fixed $n$ (Theorem 3.1,and Proposition 3.3). }

---

> ### Author Response · Authors · 2022-08-02
> **Response to Reviewer V9yg: Part (2/4)**
>
> > **Comment 2: What is the difference with Reddi et al. 2018?**
> "Reddi et al. 2018 has already studied Adam in the online convex optimization setting: they provided regret guarantees and non-convergence of average regret.  However, this paper consider... only to a stationary point"  "I don't see why the framework of finite-sum optimization  would be so important to differentiate this paper from Reddi et al. 2018."  "The paper does not convey important messages about the convergence of Adam compared with Reddi et al. 2018. ".
>
> Correct us if we mis-understand you:
> The reviewer seems to suggest that Reddi et al. 2018 prove both the convergence and non-convergence for Adam. However, we respectfully point out this view is NOT true: they only prove non-convergence of Adam, but do NOT provide any condition for convergence of Adam; what they did is to prove convergence of AMSGrad --a variant of Adam.
>
> **We believe we have important new messages not shown in  Reddi et al. 2018**. We clarify the difference as below.
>
> **Main differences with Reddi et al.2018**: Our message is significantly different from that in Reddi et al. 2018.  Actually, the messages are quite opposite. We explain as follows.
>
> - **Message in Reddi et al. 2018**: They point out that Adam can diverge. To fix the divergence issue, **they introduce a new method called AMSGrad which provably converges**.
>
> - **Our message:**  Despite the divergence by Reddi et al. 2018, we point out that **Adam can still converge without any modification**. So practitioners can use it confidently.
>
> - In particular, we point out a crucial gap between Reddi et al 2018's counter-example and the setting in practical application: the order of picking $n$ or picking $(\beta_1, \beta_2)$ is different. Then, in the setting of practical applications, we manage to prove convergence.
>
> We believe it is important to study vanilla Adam without any modification. We explain two reasons as follows.
>
> - 1.Firstly, though AMSGrad fix the divergence issue of Adam, it was reported to be inefficient in practice. That is, AMSGrad often slows down the training process [1 ,2].
>
>    [1] Adashift: Decorrelation and convergence of adaptive learning rate methods.
>
>    [2]  Adaptive gradient methods with dynamic bound of learning rate.
>
> - 2.Second, **vanilla Adam is still exceptionally popular**: Up to July 2022, Adam received more than 110,000 citations. **However, there is very litle understanding on the behavior of vanilla Adam** (perhaps due to the criticism of divergence). Our result shows that Adam is still theoretically justified. It could be safely deployed as long as hyperparameters are chosen properly.
>
> **Other differences with Reddi et al. 2018:**
> Here, we compare the convergence result in Reddi et al. 2018(their Theorem 4) with our convergence result (our Theorem 3.1).
>
> - 1.As discussed above, they study AMSGrad while we study Adam.
>
> - 2.They require  both bounded gradient and bounded domain condition; while we don't need any of these.
>
> - 3.They consider decreasing momentum parameter $\beta_{1t}$, while we consider any constant $\beta_1$.
>
> - 4.They study convex  setting while we study nonconvex  setting.
>
> We will add the above discussion to the revised version if necessary.
>
> > **A related question:**  "What does the result look like in the online nonconvex optimization and finite-sum convex optimization setting?"
>
> **Finite-sum convex setting**: Based on our Theorem 3.1 (convergence of gradient norm), we can easily recover "convergence to global optima" by further assuming convexity.
>
> **Online nonconvex setting**: For both Reddi et al. Theorem 4 and our current results,  it is unclear how to prove "sublinear regret" to nonconvex online setting. Proving sublinear regret requires reaching global-optima, which is not desirable in nonconvex optimization. To our best knowledge, we are not aware of any first-order algorithm that can achieve this.
>
> If necessary: we can also add the following result to complete the picture:
>
> **(1)** Divergence in finite-sum non-convex setting.
>
> **(2)** Non-zero regret in non-convex online setting.
>
> **(3)** Convergence to global-optima in finite-sum convex setting.
>
> **(4)** Sublinear regret  in convex online setting.
>
> Based on our curent divergence example (function (2)), **(1)** and **(2)** can be simply achieved by changing our current divergence example to arbitrary high-order (higher than 2) polynomials when $x< -1$.
> For this new function, without changing any proof in Appendix E, we can recover the same divergence results as Proposition 3.3. Further, when expressing this new function in the online setting (by sequentially sampling f1 f2 f3 f1 f2 f3..).  The average regret explodes to infinity under the same condition as the current version.  This can be shown without changing any proof in Appendix E.
>
> Based on Theorem 3.1 (convergence of gradient norm),  **(3)** and **(4)** can be shown with a few extra standard steps in optimization textbook.

---

> ### Author Response · Authors · 2022-08-02
> **Response to Reviewer V9yg: Part (3/4)**
>
> > **Comment 3: Convergence only to stationary points is not satisfactory.** "it is possible that the average regret does not converge but Adam indeed converges to a stationary point in the online learning setting such as Reddi et al. 2018."  "The regret minimization might fail in certain regimes of $\beta_1$ and $\beta_2$, but it might also converge to a stationary point without the knowledge of  or even in an online setting.  I don't know why Theorem 3.1 is so important since it is not comparable with Theorem 4 of Reddi et al. 2018. " "the authors need to provide a hardness result in an online learning setting in terms of non-convergence to a stationary point in polynomial time."
>
> Correct me if we mis-understand your comment: Reviewer suggests that ”In the counter-example by Reddi et al. 2018, although Adam suffers non-zero regret, it still converges to a stationary point”. We respectfully disagree with this point, since in that example Adam does NOT converge to the stationary point of the problem. This comment may have confused the notion of "stationary point". We will elabaroate below.
>
> Reddi et al. 2018 consider convex online constrained problem where $x \in [-1,1]$. In a constrained problem, **the stationary point often refers to the KKT point** . Since the problem is convex, the only stationary point (KKT point) is the global minimizer $x =-1$. Reddi et al. show that Adam can converge to a point $x =1$, which is sub-optimal and NOT a stationary point (KKT point).
>      The reviewer might think that the "stationary point" is defined as "the point where the algorithm does not move" (correct us if wrong). If using this (non-common) definition, then Adam converges to a "stationary point" (of the algorithm Adam).  However, we respectfully point out that in optimization area, such a point is often called "fixed point" of the algorithm, NOT the "stationary point" of the algorithm. Stationary point often refers to the ”stationary point” or KKT point of the problem.
>
> Suggested modification: We do find that many researchers are not familiar with the notion of "stationary point". We will modify "stationary point" to "critical point" in the revised version. Hopefully this will minimize such misunderstanding.
>
>
>
> For completeness, we  summarize our results and Reddi et al's result under refined expression. Hope it will provide a clearer comparison.
>
> **Reddi et al 2018's non-convergence result:** In convex problem, the fixed point of Adam might be sub-optimal. This is the standard description of non-convergence.
>
> **Our convergence result:** In nonconvex problem, the fixed point of Adam is critical point of the problem. This is the standard description of convergence.
>
>
> As introduced in Section 1, our result does not contradict with Reddi et al. 2018.  This is NOT due to "convex vs nonconvex" or "online vs finite-sum". Rather, this is because Reddi et al. consider "picking $n$ after picking $(\beta_1,\beta_2)$ " while we consider the reversed ordering.

---

> ### Author Response · Authors · 2022-08-02
> **Response to Reviewer V9yg: Part (4/4)**
>
> > **Other questions:**
>
> > **Question 1:**
> "The convergence to stationary point only when  in Theorem 3.1. What is the relationship between  in Assumption 2.2 and the bounded gradient? We know that Adam can converge if we have a bounded gradient (e.g., Guo et al. 2021) "
>
>
> When $D_0 = 0$, we have  $\sum_ {i=0}^{n-1}\left\|\nabla f_ {i}(x)\right\|_ {2}^{2} \leq D_ {1}\|\nabla f(x)\|_ {2}^{2}.$
>
> This condition (a.k.a. SGC) does  NOT turn into bounded gradient assumption since the r.h.s. is still unbounded. Instead, this condition implies that:
> When $\|\nabla f(x)\|=0$, we have $\left\|\nabla f_{j}(x)\right\|=0$ for all $j$.
> SGC holds when the model is able to interpolate or fit the labelled training data completely:
> When every data point reaches zero training loss,  the gradient of each point also converges to zero.
> Interpolation often happens for expressive models such as over-parameterized deep neural networks.  Please refer to [1] for more info. We will also add more explanation to the revised script.
>
> We emphasize that $D_0$=0  is a **necessary condition for convergence to stationary points**. We have numerical results  in Appendix B1 showing that when $D_0 \neq 0$, Adam CANNOT reach 0 gradient. Instead, it only converge to a neighborhood of stationary point (Figure 15).
> As a result, "convergence to neighborhood" is also proved in  [2,3,4,5,6].
> We classify "convergence to neighborhood" as a "good case" because it also happens in SGD. We distinguish it from the "bad case" of diverge to infinity (as summarized in our Table 1).
>
>
> [1] Vaswani, S., Bach, F. and Schmidt, M., 2019, April. Fast and faster convergence of sgd for over-parameterized models and an accelerated perceptron.
>
> [2] Zaheer, M., Reddi, S., Sachan, D., Kale, S., and Kumar, S. Adaptive methods for nonconvex optimization.
>
> [3] Shi, N., Li, D., Hong, M., and Sun, R. Rmsprop converges with proper hyper-parameter.
>
> [4] Yan, Y., Yang, T., Li, Z., Lin, Q., and Yang, Y. A unified analysis of stochastic momentum methods for deep learning.
>
> [5] Yu, H., Jin, R., and Yang, S. On the linear speedup analysis of communication efficient momentum sgd for distributed non-convex optimization.
>
> [6] Liu, Y., Gao, Y., and Yin, W. An improved analysis of stochastic gradient descent with momentum.
>
>
>
>
> > **Question 2:**  "Experiments are rather weak. The authors did not perform an ablation study in terms of $n$. "
>
> These experiments are demonstrated in Appendix B1, Figure 11, 12. As shown in these Figures, the divergence region of $(\beta_1, \beta_2)$ expends with $n$. Adam would diverge if $(\beta_1, \beta_2)$ are chosen inappropriately  independent of $n$.
>
>
>
> > **Question 3:**  How does the knowledge of $n$ affect the results theoretically and empirically?
>
> Theoretically:  it affect the threshold of minimum $\beta_2$ to ensure convergence (i.e., the threshold $\gamma(n)$ in Theorem 3.1). For larger $n$, we need larger $\beta_2$ to ensure convergence.
>
> Empirically: We observe numerically that the divergence region of $(\beta_1, \beta_2)$ expends with $n$ (Appendix B1, Figure 11 12).  These phenomena align with the above statement in Theorem 3.1,
>
>
>  Thanks again for reviewing our paper. We are happy to engage in any further discussion.

---

### Official Review · Reviewer_UR9H · 2022-07-11

**Rating:** 6
**Confidence:** 3
**Soundness:** 3 good
**Presentation:** 3 good
**Contribution:** 3 good

**Summary:**

This paper studies the Adam algorithm and proves that randomly shuffled Adam converges to the neighborhood of stationary points when 1st and 2nd-order momentum parameters satisfy that $\beta_1 < \sqrt{\beta_2} < 1$ and $\beta_2$ is sufficiently large. The analysis does not rely on the bounded gradient assumption. The divergence behavior of Adam is also studied, where it is shown that when $\beta_2$ is small, Adam can diverge to infinity for a large region of $(\beta_1, \beta_2)$.

**Questions:**

Some comments and questions are as follows.

1. The font of some figure legends is too small, e.g., Figure 1(a, b, c), Figure 2, and Figure 4. The axis labels like beta1 and beta2 should be replaced with $\beta_1$ and $\beta_2$.
2. In Theorem 3.1, is $T$ fixed? What is the dependence of $\beta_2$ on $T$?
3. Theorem 3.1 only shows that the loss gradient will become smaller in expectation, and there is still a gap between the convergence of $\mathbb{E}[||\nabla f(x_{k,0})||]$ and the actual convergence of $x_{k,0}$. How to resolve this?
4. As shown in Figure 4(c), the training loss does not seem to converge even for large $\beta_2$?
5. In the statement of Lemma G.6, $\alpha$ is a random variable, but the left-hand-side of the equations after line 996 are expectations, so why can they be bounded by a random variable?
6. In Appendix G.3 for the proof of Lemma G.3,  $\frac{\partial_\ell f(x_{k,0})}{\sqrt{v_{k,0}}}$ should be $\frac{\partial_\ell f(x_{k,0})}{\sqrt{v_{\ell,k,0}}}$ instead? Similarly for the subsequent proofs.
7. When controlling term (a) in the equation after line 1156, in the first inequality where Lemma G.12 is applied, there is a problem with the second term. Due to the indicator $\tilde I_{k, k-1}$, we have $\max_i |\partial_\ell f_i(x_{k,0})| \geq Q_k$ which is a lower bound on the absolute value, and this does not lead to an upper bound on the absolute value of the second term, so the inequality is wrong here. Did I miss anything? If not, how to fix this?


---
Minor issues:

1. In the statement of Lemma F.3, it should be $|\partial_\ell f_{\tau_k, i}(x_{k, i+1}) - \partial_\ell f_{\tau_k, i}(x_k,i)|$?
2. Unify terms like $1+\beta_1+\beta_1^2 + \cdots$ and $1 + \beta_1 + \cdots + \beta^\infty$ (define this properly).


**Limitations:**

The authors have adequately addressed the limitations and potential negative societal impact of their work.

**Strengths And Weaknesses:**

The convergence analysis of Adam is an important topic, and the claim made in this paper advances the understanding in this regard. The paper is well-written and easy to follow, and the graphic illustrations are nice. Some discussions on the comparison with existing works look redundant, especially those repeatedly emphasizing the removal of the bounded gradient assumption and the difference between the current work and the result in Reddi et al. (2018). I think these should be more condensed.

The theoretical analysis seems to be novel, though there might be some issues as will be discussed below. I'm willing to adjust the score if the questions are addressed.

---

> ### Author Response · Authors · 2022-08-02
> **Response to Reviewer UR9H: Part (1/2)**
>
> We thank the reviewer for the thoughtful review and feedback. We also thank the reviewer for the careful proof reading.
> > **Question:** In Theorem 3.1, is $T$ fixed? What is the dependence of $\beta_{2}$ on $T$ ?
>
> We respectfully point out that $T$ is not fixed here. Theorem 3.1 holds for any $T$ larger than the threshold $k_m$ ($k_m$ usually = 15 as we stated in footnote 4. ).
> Here, we analyze vanilla Adam with constant $\beta_2$.  So $\beta_2$  does not depend on T. We will modify the statement of Theorem 3.1 to minimize the mis-understanding.
>
> > **Question:** There is a  gap  between $\mathbb{E}\left[\left\|\nabla f\left(x_{k, 0}\right)\right\|\right]$ and the actual convergence of $x_{k, 0}$. How to resolve this?
>
> We totally agree that "converges of gradient norm in expectation" does not imply the "convergence of all possible trajectories".
> In the script, we claim "convergence" for the ease of presentation. If necessary, we will explain that our convergence results imply "the sequence of Adam converges to the set of stationary points with high probability"; but it does not imply ``the sequence of Adam converges to a stationary point almost surely".
>
> Meanwhile, it is often difficult to prove almost-sure convergence. For Adaptive gradient methods, most well-known results (e.g., including but not limited to [1,2,3,4,5]) refer to "convergence of gradient norm in expectation".
> Proving almost-sure convergence for Adam will be an interesting theoretical direction.
>
> [1] Zou, F., Shen, L., Jie, Z., Zhang, W., and Liu, W. A sufficient condition for convergences of
> adam and rmsprop.
>
> [2] Zaheer, M., Reddi, S., Sachan, D., Kale, S., and Kumar, S. Adaptive methods for nonconvex  optimization.
>
> [3] Chen, X., Liu, S., Sun, R., and Hong, M. On the convergence of a class of adam-type algorithms for non-convex optimization
>
> [4] Luo, L., Xiong, Y., and Liu, Y. Adaptive gradient methods with dynamic bound of learning rate.
>
> [5] Zhou, D., Chen, J., Cao, Y., Tang, Y., Yang, Z., and Gu, Q. On the convergence of adaptive gradient methods for nonconvex optimization.
>
> > **Question:** in Figure 4(c), the training loss does not seem to converge even for large $\beta_{2}$?
>
>  When training Transformer XL (Dai et al., 2019) on the WikiText-103 dataset, the training loss of around 2 and 3  is already satisfactory.  This is a bit different from CV tasks where training loss is often close to 0. In the revised version, we will plot the training loss trajectory to show the convergence.
>
> > **Question:** In the statement of Lemma G.6, $\alpha$ is a random variable, but the left-hand-side of the equations after line 996 are expectations, so why can they be bounded by a random variable?
>
> We apologize for having a typo here:  the right-hand-side should be $E(\left|\partial_{\alpha} f\left(x_{k, 0}\right)\right|)$ instead of $\left|\partial_{\alpha} f\left(x_{k, 0}\right)\right|$. We already corrected this typo in the revised version (in Lemma G.5, line 1007), please feel free to check.

---

> ### Author Response · Authors · 2022-08-02
> **Response to Reviewer UR9H: Part (2/2)**
>
> > **Question:**
> When controlling term (a) in the equation after line 1156, in the first inequality where Lemma G. 12 is applied, there is a problem with the second term. Due to the indicator $\tilde{I}_ {k, k-1}$, we have $\max _ {i}\left|\partial_ {\ell} f_ {i}\left(x_ {k, 0}\right)\right| \geq Q_ {k}$ which is a lower bound on the absolute value, and this does not lead to an upper bound on the absolute value of the second term, so the inequality is wrong here. Did I miss anything? If not, how to fix this?
>
> Thanks for pointing it out. We apologize for this careless flaw. However, this inequality can be fixed with two simple extra steps.  We explain as follows.
>
> Line 1156 claims the following  upper bound(we omit all the irrelevant multiplied constant terms):
>
> $$
> \tilde{\mathbb{I}}_ {k,k-1} \left|\sum_ {i=0}^{n-1} \partial_ l f_ i(x_ {k,0})\right| \leq  n Q_k,
> $$
>
> where  $\tilde{\mathbb{I}}_ {k,k-1}:=\mathbb{I}\left( \max_ i |\partial_ l f_ i(x_ {k,0})| \geq Q_ k \text{ and } \max_ i |\partial_ l f_ i(x_ {k-1,0})|  \leq Q_ {k}+Q_ {k-1}  \right)$.
>
> As you mentioned, according to the definition of the indicator, we only have a lower bound for $|\partial_ l f_ i(x_ {k,0})|$. So we cannot directly get $ |\partial_ l f_ i(x_ {k,0})| \leq Q_ k$ (as we did in the script). To get the correct upper bound, we perform the following extra two steps.
>
> **Step 1:**
> We change $|\partial_l f_i(x_{k,0})|$ into $|\partial_l f_i(x_{k-1,0})|$ using Lipschitz property (Lemma  F.3 in the appendix  or Lemma F.2 in the revised version):
>
> $$\tilde{\mathbb{I}}_ {k,k-1} \left|\sum_ {i=0}^{n-1} \partial_ l f_ i(x_ {k,0})\right|
> \leq \tilde{\mathbb{I}}_ {k,k-1} \left|\sum_{i=0}^{n-1} \partial_ l f_ i(x_ {k-1,0})\right| +  n^2 \triangle_ {n(k-1)}$$.
>
> **Step 2:**
> By the definition of $\tilde{\mathbb{I}}_ {k,k-1}$,
> we can get an upper bound for $|\sum_{i=0}^{n-1} \partial_l f_i(x_{k-1,0})|$. We are led to the following upper bound.
>
> $$\tilde{\mathbb{I}}_ {k,k-1} \left|\sum_ {i=0}^{n-1} \partial_ l f_ i(x_ {k,0})\right|
> \leq  n (Q_ k + Q_ {k-1}) +  n^2 \triangle_ {n(k-1)}$$.
>
>
>
> We can further relax $(Q_k + Q_{k-1}) \leq 3Q_k$ using the relation  $Q_{k-1} \leq 2 Q_k$.
>
> With the above two steps, we can recover the original inequality  with the following minor changes: (1) Change  the constant terms before $Q_k$.  (2) Add an additional error term $n^2\triangle_{n(k-1)}$, which is still in the order of $\mathcal{O}(1/\sqrt{k})$ and can be absorbed into other existing terms in line 1156.
>
> In the revised script (already submitted to OpenReivew), we have already added Step 1 to 2 to the revised appendix using **GREEN** color (line 1166, 1167 in the revised version). All the changes of constant terms due to the extra Step 1 and Step 2 are also marked in **GREEN** color. Please feel free to check.
>
>
> > **Other comments:** The font of some figure legends is too small, e.g., Figure 1(a, b, c), Figure 2, and Figure 4. The axis labels like beta1 and beta2 should be replaced with $\beta_1$ and $\beta_2$.
> Some discussions on the comparison with existing works look redundant.
> There is typo in the statement of Lemma F.3 \& Please unify the expression of infinite sum.
>
> Thanks for pointing out these issues. We will revise them accordingly.

---

> > ### Comment · Reviewer_UR9H · 2022-08-06
> > **Thanks for the clarifications**
> >
> > I thank the authors for their efforts. Overall I'm satisfied with the responses. I've also read other reviewers' reviews and the corresponding rebuttal. I have adjusted the rating accordingly.
> >
> >
> > IMHO, for the proofs in the appendix, it seems that it might be clearer to write out those sums using explicit indices instead of '$+ \cdots$' because sometimes it is difficult to figure out what is omitted.

---

> > > ### Author Response · Authors · 2022-08-07
> > > **Thanks for your positive feedback**
> > >
> > > We'd like to thank the reviewer for the positive feedback. We will polish our writing based on your suggestions.

---

### Official Review · Reviewer_tmNN · 2022-07-19

**Rating:** 6
**Confidence:** 4
**Soundness:** 4 excellent
**Presentation:** 4 excellent
**Contribution:** 3 good

**Summary:**

In this paper, the authors provide a solid convergence analysis which suggest Adam can converge to stationary points by picking proper $\beta_1$ and $\beta_2$ after fixing $n$. The conclusion is proved without assumption of bounded gradient.  The analysis can provide new insights for tuning Adam.

**Questions:**

In the unconvergence counterexample provided by Reddi et al, 2018, the gradient variance is coupled with the number of target functions, i.e. $n$. Can authors explain more about how to decouple $n$ and gradient variance in your analysis?

**Limitations:**

Some prior works have provided similar conclusion that tuning large $\beta_1$ and $\beta_2$ can make Adam converge. Even so, this paper provides a more solid and through analysis on generalized function class.

**Strengths And Weaknesses:**

Strengths:
+ The authors provide a through convergence analysis for the well-known unconvergence issus of Adam.  By picking large $\beta_1$ and $\beta_2$ after fixing $n$, the authors prove Adam can converge to stationary point of the specific function class given by Assumption 2.1 and Assumption 2.2.
+ Compared to the divergence counterexamples provdied by [1], the target function is generalized to a larger class.

Weaknesses:
+ However, some prior works [2] have provided the similiar conclusion that, after fixing the unconvergen counterexamples in [1], tuning larger $\beta_1$ and $\beta_2$ can make Adam converge. Meanhwile, they provide an equation to descibe the relation between $\beta_1$, $\beta_2$ and $C$ (the divergence cirtical point in [1]). Even so, their analysis is contrained on the convex finite-sum problem in [1] and not generalized. The analysis in paper would be more general. Can authors explain more comparison and strengths over them?
+ Theorem 3.1 depends on Assuption 2.2 which contrain the variance of stochastic gradients. However, the unconvergence of Adam emerges only when variance is large [1]. Would Theorem 3.1 mean that Adam converge when gradient variance is within the constrains in Assumption 2.2? Can author analyze the behavior of Adam and how to tune $\beta_1$ and $\beta_2$ when gradient variance is large enought to diverge Adam?

[1] On the Convergence of Adam and Beyond. ICLR 2018.
[2] AdaShift: Decorrelation and Convergence of Adaptive Learning Rate Methods. ICLR 2019.
[2] Adaptive Gradient Methods with Dynamic Bound of Learning Rate. ICLR 2019.

---

> ### Author Response · Authors · 2022-08-02
> **Response to tmNN: Part (1/2)**
>
> We thank the reviewer for the valuable feedback and comments.
>
> > **Question 1:**  Can authors explain more comparison and strengths over  Theorem 1 in [2]?
>
> Thanks for pointing out this result in [2]. Let us briefly summarize the difference between Theorem 1 in [2] (convergence on Reddi's example in [1]) and our Theorem 3.1 (convergence on generic problems).
>
>
> **Strength 1:** As pointed out by the reviewer,  the convergence result for Adam in [2] is constrained in the setting of counter-example in [1]. What they find is that  "not all $\beta_1$ and $\beta_2$ would fail on the counter-example".
>  However, they do NOT provide any condition for convergence of Adam on other general nonconvex functions. In contrast, we prove the convergence on a broad class of generic non-convex problems.
>
>
> **Strength 2:** They require bounded gradient assumption; while we do not need it.
>
> **Strength 3:** They require extra boundedness condition on $v_t \in [C_1, C_2]$; while we do not need it.
>
> We will include the comparison in the revised Section 2.2.
> For completeness, we still want to emphasize that our work delivers **significantly different messages** from that in  [2]. We summarize as below.  (This may not be the real focus of your question. But for completeness, let us elaborate the difference here.)
>
> **Messages in [2]:**
> They revisited the counter-example in [1] and analyze the reason for Adam's divergence. **Then they propose a new method AdaShift to fix the convergence issue**.
>
>
>
> **Our messages:**  Despite the divergence by Reddi et al. 2018, we point out
> that: For generic nonconvex problems, **Adam can still converge without any modification. So practitioners can
> use it confidently**.
>
>
>
> We believe it is important to study vanilla Adam (without any modification) for generic nonconvex problem.
> This is because vanilla Adam is still exceptionally popular in deep learning tasks: up to July 2022, vanilla Adam
> received more than 110,000 citations. However, there is very litle understanding
> on the behavior of vanilla Adam (perhaps due to the criticism of divergence).
> Our result shows that Adam is still theoretically justified. It could be safely
> deployed as long as hyperparameters are chosen properly.
>
>
>
> [1] On the Convergence of Adam and Beyond. ICLR 2018.
>
> [2] AdaShift: Decorrelation and Convergence of Adaptive Learning Rate Methods. ICLR 2019.
>
>
>
>
>
> > **Question 2:** Theorem 3.1 depends on Assumption 2.2 which contrain the variance of stochastic gradients. However, the unconvergence of Adam emerges only when variance is large [1]. Would Theorem 3.1 mean that Adam converge when gradient variance is within the constrains in Assumption 2.2?
>
>
>
> Reviewer comments that "the unconvergence of Adam emerges only when variance is large". If we understand you correctly, this comment implicitly suggests that Assumption 2.2 is the key factor of "avoiding divergence a priori". If this is the comment in your mind (please correct us if wrong), then we provide the response as follows.
>
> We respectfully point out this view is NOT correct.  Reddi et al's counter-example (1-dimensional linear functions) actually satisfies Assumption 2.2 with $D_1= n^2 + n -1$ and $D_0 =0$.   Therefore, Assumption 2.2 is not the key factor of "avoiding divergence". Instead, as we point out in the Introduction, the key factor lies in the order of picking the problem and $(\beta_1, \beta_2)$. Recall Reddi et al. fix $(\beta_1, \beta_2)$ before constructing the divergence example.
> What Reddi et al. suggest is that:  *for specific $(\beta_1, \beta_2)$*, we can construct divergence example with large enough variance. While we point out the other side of the coin: When the problem is fixed (even when the variance is large), there is still certain  $(\beta_1, \beta_2)$ that brings convergence.
>
>
> In the revised version of Appendix D.2, we will add more relevant discussion and emphasize that Reddi et al.'s example indeed satisfies Assumption 2.2. We are happy to engage in any further discussion.

---

> > ### Comment · Reviewer_tmNN · 2022-08-10
> > **Thanks for the response.**
> >
> > Thanks for the authors' effort to reply my confusion and questions. I am statisfied with the authors's clarification as well as the revision of paper. I would adjust the rating accordingly.

---

> > > ### Author Response · Authors · 2022-08-10
> > > **Thanks for your positive feedback**
> > >
> > > We would like to thank the reviewer for the positive feedback. Thanks again for your effort and time in reviewing our paper.

---

> ### Author Response · Authors · 2022-08-02
> **Response to tmNN: Part (2/2)**
>
>
> > **Question 3:** Can author analyze the behavior of Adam and how to tune $\beta_{1}$ and $\beta_{2}$ when gradient variance is large enough to diverge Adam?
>
> 1.  *Analyze the behavior*: In the setting of [1], variance of gradient  $= \frac{1}{n}(n^2 + n-1) -\frac{1}{n^2}$. So large variance happens when $n$ is large. When $n$ is large, our analysis can still be applied. As long as $n$ is fixed  (no matter large or small), we point out Adam converges when $\beta_2$ is larger than a threshold dependent on $n$ (Theorem 3.1). On the other hand, Adam can diverge when $\beta_2$ is small.
>
>
>
> 2. *How to tune $\beta_1$ and $\beta_2$*: Based on the behavior of Adam pointed out above. We provide the following simple solution:  When variance (or $n$) is large enough to diverge Adam under certain $\beta_1$ and $\beta_2$, we can just simply **tune up $\beta_2$ to obtain convergence**. When $\beta_2$ is larger than the threshold $\gamma(n)$,  Adam will converge for all $\beta_1 <\sqrt{\beta_2}$. This tuning suggestion is also introduced in Appendix C for practitioners.
>
>
> We will add more relevant discussion to the revised script.
>
>
> **Question 4:** In the unconvergence counterexample provided by Reddi et al, 2018 , the gradient variance is coupled with the number of target functions, i.e. $n$. Can authors explain more about how to decouple $n$ and gradient variance in your analysis?
>
>
> We did not try to  decouple $n$ and variance in our analysis. The key property is that $n$ is fixed in our analysis. When $n$ is fixed (no matter large or small), we can always tune up $\beta_2$ to be larger than an $n$-dependent threshold $\gamma_1(n)$ such that the variance of gradient is under control. As such, Adam will lie near the gradient direction. Note that "fixing $n$" or "fixing samplesize" is  common in practice. So we believe our setting is reasonable.
>
>
> Thanks again for reviewing our paper. We are happy to engage in any further discussion.

---

### Author Response · Authors · 2022-08-04
**Dear AC and reviewers: A revised script has be uploaded**

Dear AC and reviewers:

Thanks a lot for reviewing our paper. We have revised the paper as follows.

Firstly, we modify the presentation and emphasize that **we focus on vanilla
Adam instead of any of its variant algorithm**. During the rebuttal phase,
we do find that this message is overlooked in the submitted version. However,
this message is crucial to distinguish our work and most literature on "adaptive
gradient methods & Adam family”: Most existing related works analyze new
variants of algorithms to fix the divergence issue, while we take a different perspective. We find out that, on generic nonconvex functions, vanilla Adam can converge without any modification! Compared with Reddi et al. 2018, we deliver
an opposite (but still reasonable) message that **"Adam is still good, please use it
confidently”**. We hope the newly added discussions can better express the main focus of the paper.

We believe this positive result for Adam is important: Despite the divergence
issue by Reddi et al., vanilla Adam is still exceptionally popular: Adam has
received more than 110,000 citations. However, we have very little understanding
of why it works well. Up to now, the convergence of vanilla Adam is far less
studied than its variants (perhaps due to the criticism of divergence).

The lack of theory might cause confusion for practitioners. Imagine I was a
practitioner and I am using Adam to train deep neural nets. However, Adam
with default hyperparameter fails in my tasks. Shall I keep tuning hyperparameter
to make it work? Or shall I just give up and switch to other algorithms? Due
to the existence of divergence example and the absence of convergence theory,
I am afraid I would easily give up on Adam since I don’t see any hope out of
it. However, our paper points out that ”Adam is still theoretically justified.
Please tune the hyperparameters confidently”. Further, we provide some tuning
suggestions to save the cost of grid-searching.

We believe our positive result is
meaningful to the community. For practitioners, our result can refresh our understanding on Adam. It can also help better train deep networks. For theorists, our analysis does not require bounded gradient assumption (which is a common assumption for Adaptive gradient methods). Our proof consists of a new method to handle unbounded momentum in a stochastic non-linear dynamics system.

Besides the above changes, we also revise our script based on most comments by the
reviewers. Some modifications are highlighted in different colors. Due to the
limitation of time, a few discussions in the rebuttal haven’t been updated in the
script yet. We will further add these discussions in the future version very soon.

Thanks again for the time and effort on reviewing our paper.

Authors

---

### Public Comment · ~Kyunghun_Nam1 · 2023-08-09
**Some question about Appendix G.**

Dear authors

I hope this message finds you well. Currently, I am a graduate student with a focus on optimization, and I have been deeply engaged in studying your contributions to this field.

While analyzing your recent work, specifically your thesis, I found myself intrigued by certain aspects of the concepts you've presented. However, some parts remain unclear to me. I would be immensely grateful if you could provide clarifications on the following points:

1. In Appendix G.1, the derivation of the lower bound of $\mathbb{E} \langle \nabla f(x_{k, 0}), x_{k, 0} - x_{k+1, 0} \rangle$ and the upper bound of $\mathbb{E} \lVert x_{k+1, 0} - x_{k, 0} \rVert_2^2$ are [resemted as the starting points for convergence analysis. Could you please elaborate on the reasoning behind this choice and the significance of deriving the bound for these terms? While I follow the inequality (3) in Section 5 $\mathbb{E} \langle \nabla f(x_{k, 0}), \sum_{i=0}^{n-1} \frac{m_{k, i}}{\sqrt{v_{k, i}}} \rangle > 0$, I am particularly interested in understanding why demonstrating a lower bound, not greater than 0, was essential to initiate the proof.

2. The derivation of the upper bound $O(\eta_k^2 m_{l, k, i}/v_{l, k, i}) = O(\frac{1}{k})$ in accordance with Adam's update rule has caught my attention. I find myself struggling to fully grasp how this relationship was formulated. Your insights into this part of the derivation would be greatly appreciated.

I understand that these inquiries might touch on foundational elements of the subject, but your expert perspective would be invaluable to my ongoing research. Thank you very much for considering my request, and I look forward to hearing from you soon.

Best regards,

---

> ### Public Comment · Authors · 2023-08-11
> **Response to your question**
>
> Dear Kyunghun:
>
> Thanks for your interest in our work! We address your questions as follows.
>
> **Q1:** Why choosing the lower bound of  $\mathbb{E}\left\langle\nabla f\left(x_{k, 0}\right), x_{k, 0}-x_{k+1,0}\right\rangle$ and the upper bound of $\mathbb{E}\left\|x_{k+1,0}-x_{k, 0}\right\|_2^2$  as the starting points for convergence analysis. Elaborate on the reasoning behind this choice and the significance of deriving the bound for these terms?
>
> **A1:** In all the following answers, we will refer $\mathbb{E}\left\langle\nabla f\left(x_{k, 0}\right), x_{k, 0}-x_{k+1,0}\right\rangle$  as Term 1 and $\mathbb{E}\left\|x_{k+1,0}-x_{k, 0}\right\|_2^2$   as Term 2.
>
> In convergence analysis. Term 1 characterizes “how close you are to the negative gradient direction”. As the negative gradient direction is the steepest descent direction, usually Term 1 contributes to the “improvement along iteration”. As such, we need a lower bound for Term 1 to show that the “improvement per epoch” is not too bad.
>
> In contrast, Term 2 is usually from the 2nd-order Talyor expansion and could be understood as an “error term” that does not contribute to the convergence. As such, we would like to derive an upper bound for this error term and show that it is negligible compared with Term 1.
>
> **Q2:** Why demonstrating a lower bound, not greater than 0?
>
> **A2:** Proving Term 1>0 is a necessary condition for convergence to stationary points. It tells you that your algorithm is moving in the “correct” direction. However, it does not tell us how fast the algorithm would converge to the solution set you want (e.g., the set of stationary points).
>
> Proving a lower bound of “Term 1 > constant * gradient norm^2” or “Term 1> constant * gradient norm” would further characterize the rate of convergence.
>
> **Q3:** About the derivation of the upper bound of Term 2.
>
> **A3:** The proof is based on the following fact: the update of Adam is bounded by a constant * stepsize. Therefore, when stepsize $\eta_k = 1/\sqrt{k}$, we have Term 2 $\leq \eta_k^2 * constant^2  = O (1/k)$. The detailed proof can be seen in Lemma F.2.
>
> Please let us know if there is any further question. Any form of further discussion is welcomed.
>
> Authors

---

> > ### Public Comment · ~Kyunghun_Nam1 · 2023-08-22
> > **Some additional question about Appendix G.**
> >
> > Thank you for your prompt and informative response.
> > Your explanations have greatly clarified my understanding. However, I'd like to delve deeper into a couple of points.
> >
> > In Appendix G.9, precisely Case (b), there's a section following Inequality (67) that begins with "Now the following two both claims are true". This leads to Inequality (68). Can you shed some light on this transition?
> >
> > Furthermore, in Appendix G.8, concerning the upper bounds of $(a_2)$ and $(b_2)$, the desired upper bound was for the sum of $(a_2)$ and the sum of $(b_2)$ when "$l$ is large". However, the developed expression transitions to a sum from "$l=1$" to "$d$". This seems to suggest a looser bound than initially desired (=not a tight bound).
> > And the idea of an upper bound on "large l" becomes obsolete, since we have an upper bound on the sum of all $\sum_{l=1}^d$.
> > Was this an unavoidable choice?
> >
> > Thank you in advance for your insights!

---

> > > ### Public Comment · Authors · 2023-08-22
> > > **Response to your question**
> > >
> > > Thanks for your questions. We address your concern as follow.
> > >
> > > **Q1:** About Eq (67) and (68).
> > >
> > > **A1:** We explain the core idea of the proof using simplified notation as follows. Hope it would help.
> > >
> > > Let us denote Eq. (67)  $= -c$, where $c>0$ is a certain positive constant. Let us denote $a, b_1, b_2> 0$  as certain positive constants, too.
> > >
> > > If $a<A$ for some positive constant $A$, then the following two claims hold:
> > >
> > > Claim 1: $ -c = b_1a^2 - b_1a^2 -c \geq b_1a^2 - b_1A^2 - c$
> > >
> > > Claim 2: $-c  = b_2  a - b_2 a - c \geq  b_2 a - b_2A - c$
> > >
> > > Combining together, we have:
> > >
> > > $-c \geq min [b_1a^2 - b_1 A^2 - c, b_2 a - b_2A - c ] \geq  min [b_1a^2, b_2 a] - max [b_1 A^2, b_2A] -c$
> > >
> > > If you replace the above constant with the notations and variables in the paper (e.g. $a$ corresponds to the gradient norm and $b_1, b_2$ are its coefficients), you will find the above inequality is actually Eq. (68), up to some simple re-arrangement on the constant terms.
> > >
> > >
> > > **Q2:** About the upper bounds  in Appendix G.8
> > >
> > > **A2:** No, we believe it is avoidable.  We use the current derivation because it is the simplest way to prove Lemma G.5, but it is not necessarily the cleverest way.  We believe there is room for further improvement.
> > >
> > >
> > > Hope the above explanation would help. Please feel free the raise any further question if there is any.
> > >
> > > Authors

---

> > > > ### Public Comment · ~Kyunghun_Nam1 · 2023-08-23
> > > > **Last question**
> > > >
> > > > Thank you for your quick and kind response.
> > > > I thought the above question would have been my last, but I wanted to follow up with what I think is a slight error.
> > > > It's the part on page 65 of Appendix G.9.
> > > >
> > > > Starting from "Recall $F_3 \rightarrow 0$ when $\beta_2 \rightarrow 1$," to the part where it says, "we have the following result by changing all the $F_3$  into $\frac{1}{\sqrt{10D_1d}}$", there is something I don't understand.
> > > >
> > > > 1. such that $\frac{1}{\sqrt{5 D_1d}} - F_3 \ge \frac{1}{\sqrt{10D_1d}}$ or equivalently $F_3 \le \frac{1}{\sqrt{10D_1d}}$. However, these two conditions are not the same at all. Because $\sqrt{2} - 1 \le 1$.
> > > > Please double-check.
> > > >
> > > > 2. And then there's $\epsilon$, $(1 - \epsilon, 1]$ which I can't find where it's used or appears.
> > > >
> > > > Thanks.

---

> > > > > ### Public Comment · Authors · 2023-08-23
> > > > > **Response to your question**
> > > > >
> > > > > Hi Kyunghun:
> > > > >
> > > > > We believe this is not an error. When $\beta_2$ approaches 1, $F_3$ monotonically shrinks to 0. Therefore, there must exist a threshold of $\beta_2$, s.t., whenever $\beta_2$ is larger than this threshold,  $F_3$ will be small enough, say $F_3 \leq 1/\sqrt{10D_1d}$. Under this condition, we naturally have $\frac{1}{\sqrt{5 D_1 d}}-F_3 \geq \frac{1}{\sqrt{10 D_1 d}}$.
> > > > >
> > > > > Yes,  the notation $\epsilon$  does not appear anywhere in the later proof.  Here in the sentence "Recall $F_3 \rightarrow 0$..." in Appendix G.9, we temporally use the notation "$\epsilon$" to emphasize the existence of the threshold of $\beta_2$ such that $F_3$ is small enough. This notation "$\epsilon$" could be abandoned if we make this statement using plain language, like " ...there must exist a threshold of $\beta_2$, s.t., ..." .
> > > > >
> > > > > Hope the explanation would help.
> > > > >
> > > > > Authors

---

> > > > > > ### Public Comment · ~Kyunghun_Nam1 · 2023-08-24
> > > > > > **Thank you**
> > > > > >
> > > > > > Thank you for your kind and detailed explanation of many questions.

---

### Meta-Review · Area_Chair_mXWh · 2022-08-26

**Recommendation:** Accept
**Confidence:** Certain

**Metareview:**

This paper proves that the vanilla Adam algorithm can converge to a stationary point with properly chosen $\beta_1$ and $\beta_2$ if the number of samples $n$ is fixed, in contrast to the well-known non-convergence example by Reddi et al. This result has a clear conceptual message, which justifies the use of vanilla Adam, and it's worth sharing the result with the ML community. The reviewers find the presentation and the analysis to be of high quality as well.

**Award:**

No

---

### Decision · Program_Chairs · 2022-09-14

Accept